# Incentivizing Truthfulness in Fully Decentralized Learning with Guaranteed Accurate Convergence

## Abstract

Decentralized learning has gained significant attention due to its advantages in scalability, privacy, and fault tolerance. In this paradigm, multiple agents collaboratively train a global model by exchanging parameters only with their neighbors, without the assistance of a centralized server. However, a key vulnerability of existing decentralized learning approaches is their implicit assumption that all agents behave honestly during gradient updates and information sharing. In real-world scenarios, this assumption often breaks down, as selfish or strategic agents may be incentivized to manipulate gradients or share false information for personal gain, ultimately compromising the final learning outcome. In this work, we propose a fully decentralized payment mechanism that, for the first time, guarantees both truthful behaviors and accurate convergence in decentralized stochastic gradient descent algorithms. This represents a significant advancement, as it addresses two major limitations of existing truthfulness mechanisms for collaborative learning: 1) reliance on a centralized server for payment collection, and 2) the tradeoff between ensuring truthfulness and maintaining convergence accuracy. In addition to characterizing the convergence rate under convex or strongly convex conditions, we also prove that our approach guarantees the cumulative gain that an agent can obtain through strategic behavior remains finite, even as the number of iterations approaches infinity—a property unattainable by most existing truthfulness mechanisms. Experimental results on several machine learning applications confirm the effectiveness of our approach.

## 1 Introduction

Recent years have witnessed significant advances in decentralized methods for collaborative learning and optimization (Nedic & Ozdaglar, 2009; Lian et al., 2017; Yang et al., 2019). By distributing both data and computational resources across multiple agents, decentralized methods leverage the combined computing power of multiple devices to collaboratively train a global model without the need for a centralized server. Compared with server-assisted collaborative learning[1], decentralized learning avoids monopolistic control and single points of failure (Warnat-Herresthal et al., 2021), and hence, is widely applied in areas such as decentralized machine learning (Verbraeken et al., 2020), multi-robot coordination (Shorinwa et al., 2024), and wireless networks (Du et al., 2024).

However, almost all existing decentralized learning approaches implicitly assume that all participating agents act truthfully, which is essential for their successful execution. This premise becomes untenable in practical scenarios where participating agents are strategic and self-interested. In such cases, participating agents may manipulate gradient updates or share false information to maximize their own utilities, ultimately undermining the performance of collaborative learning. For example, in decentralized learning with heterogeneous data distributions among agents, an agent may inflate its local gradient updates to skew the final model in favor of its own data distribution (Chakarov et al., 2024a;b). Similarly, in a shared market, a firm may inject noise into its shared data samples to degrade the quality of other firms' predictive model training, thereby maintaining competitive

---

[1]We use "server-assisted collaborative learning" to refer to collaborative learning involving a centralized server or aggregator, with federated learning as a representative example.

Table 1: Comparison of our approach with existing truthfulness results for collaborative learning.

| Approach | Fully decentralized? | $\varepsilon$-Incentive compatible?[a] | Budget balanced?[b] | Accurate convergence? |
|---|---|---|---|---|
| Angeli & Manfredi (2023) | ✗ | ✓ | ✗ | ✓ |
| Dorner et al. (2023) | ✗ | ✗[c] | ✓ | ✗ |
| Chakarov et al. (2024a;b) | ✗ | ✗ | ✓ | ✗ |
| Chen et al. (2025) | ✓ | ✓ | ✓ | ✗ |
| Our approach | ✓ | ✓ | ✓ | ✓ |

[a] We use "$\varepsilon$-Incentive compatible" to describe whether an approach can guarantee that the cumulative gain that an agent obtains from persistent strategic behaviors remains finite (bounded by some finite value $\varepsilon$), even as the number of iterations approaches infinity.

[b] We use "Budget balanced" to mean total payments collected equal total payments made, requiring no external subsidies or surplus. This ensures that the mechanism is financially sustainable and scalable in practice.

[c] Dorner et al. (2023) does not provide a clear result on $\varepsilon$-incentive compatibility.

advantage (Dorner et al., 2023). More motivating examples are provided in Appendix B.1. Such untruthful behaviors pose a significant threat to the performance of existing decentralized learning and optimization algorithms (as demonstrated by our experimental results in Fig. 2).

To mitigate strategic behaviors of participating agents in collaborative learning, several approaches have been proposed, which can be broadly categorized into incentive-based approaches (Zhan et al., 2021; Lyu et al., 2022; Angeli & Manfredi, 2023; Alon et al., 2024; Clinton et al., 2025) and joint-differential-privacy (JDP)-based approaches (Pai & Roth, 2013; Kearns et al., 2014; Zhu et al., 2020; 2024). However, all existing approaches rely on a centralized server to collect information from all agents and then execute a truthfulness mechanism. For example, the Vickrey–Clarke–Groves (VCG) mechanism, a well-known incentive-based approach, requires a centralized server to aggregate true gradients/functions from all agents to compute the corresponding monetary transfers (Jain & Walrand, 2010; Dave et al., 2021; Qian et al., 2021; Angeli & Manfredi, 2023; Dorner et al., 2023; Chakarov et al., 2024a;b). Similarly, JDP-based approaches require a centralized server to collect iteration variables from all agents in order to compute the necessary noise (Han et al., 2015; Hale & Egerstedt, 2015; Zhang et al., 2022)[2]. To the best of our knowledge, no existing approaches can effectively incentivize truthful behaviors in fully decentralized learning and optimization.

Our contributions are summarized as follows:

1. We propose a fully decentralized payment mechanism that incentivizes truthful behaviors among interacting strategic agents in decentralized learning and optimization. This represents a substantial breakthrough, as existing truthfulness mechanisms (in, e.g., Dave et al. (2021); Qian et al. (2021); Lyu et al. (2022); Zhang et al. (2022); Dorner et al. (2023); Angeli & Manfredi (2023); Chakarov et al. (2024a;b); Alon et al. (2024); Clinton et al. (2025)) all rely on a centralized server to aggregate local information from agents. To the best of our knowledge, this is the **first** payment mechanism implemented in a **fully decentralized** setting, without the assistance of any centralized server or aggregator.

2. Our payment mechanism guarantees that the incentive for a strategic agent to deviate from truthful behaviors diminishes to zero over time (see Lemma 1). Building on this, we further prove that the cumulative gain that an agent can obtain from its strategic behaviors remains finite, even when the number of iterations **tends to infinity** (see Theorem 2). This stands in sharp contrast to existing incentive-based approaches for federated learning in Chakarov et al. (2024a;b), which cannot eliminate agents' incentives to behave untruthfully—resulting in a cumulative gain that grows unbounded when the number of iterations tends to infinity.

3. In addition to ensuring diminishing incentives for untruthful behavior in decentralized learning, our payment mechanism also guarantees accurate convergence of decentralized learning, even in the presence of persistent gradient manipulation by agents (see Theorem 1). This is in stark contrast to existing JDP-based truthfulness results in Han et al. (2015); Zhang et al.

---

[2]Although the recent work by Chen et al. (2025) proposes a JDP-based truthfulness approach for decentralized aggregative optimization, it is limited to scenarios where each agent's objective function depends on an aggregative term of others' optimization variables.

(2022); Chen et al. (2025) and incentive-based truthfulness results in Dorner et al. (2023); Chakarov et al. (2024a;b), all of which are subject to an optimization error. We analyze the convergence rates of decentralized learning under our payment mechanism for general convex or strongly convex global objective functions. This is more comprehensive than existing truthfulness results in Dave et al. (2021); Angeli & Manfredi (2023); Dorner et al. (2023); Chakarov et al. (2024a;b) that focus solely on the strongly convex case.

4. Different from most existing VCG-based approaches (in, e.g., Angeli & Manfredi (2023); Alon et al. (2024)), which cannot ensure budget-balance (the total payments from all agents sum to zero, a property essential for the financial sustainability and practical scalability of the mechanism), our payment mechanism is budget-balanced. This is significant in a fully decentralized setting since no centralized server is used to manage subsidies or surplus.

5. We evaluate the performance of our truthful mechanism using multiple representative decentralized machine learning tasks, including image classification on the FeMNIST dataset, sentiment analysis on the Sent140 dataset, and next-character prediction on the Shakespeare dataset. The experimental results confirm the effectiveness of our approach.

## 2 RELATED WORK

Due to space limitations, we leave the comparison between our work and relevant studies on decentralized learning/optimization, game theory, robustness in decentralized learning/optimization (e.g., Byzantine attacks), and personalized learning to Appendix B.3. Furthermore, we highlight the key differences between our approach and state-of-the-art truthfulness results in Table 1.

**Incentive-based truthfulness approaches.** Truthfulness in statistical (mean) estimation has been addressed using incentive/payment mechanisms (Cai et al., 2015; Chen et al., 2023; Clinton et al., 2025). These mechanisms are typically one-shot, where agents choose their strategies once and a centralized server broadcasts payments accordingly, which renders them inapplicable to multi-round, gradient-based decentralized learning algorithms (a detailed discussion on the difference between truthfulness in server-assisted and decentralized mean estimation is provided in Appendix B.2). Based on the well-known VCG mechanism (Vickrey, 1961; Clarke, 1971; Groves, 1973), truthfulness results have also been reported for federated learning (see, e.g., Dave et al. (2021); Qian et al. (2021); Lyu et al. (2022); Angeli & Manfredi (2023); Dorner et al. (2023); Chakarov et al. (2024a;b)). However, the VCG mechanism relies on a server to calculate and collect monetary payments, which makes it inapplicable in a fully decentralized setting. It is worth noting that many results have discussed incentive mechanisms for encouraging agents' contributions of data/resources in collaborative learning (Chen et al., 2020; Sim et al., 2020; Blum et al., 2021; Fraboni et al., 2021; Karimireddy et al., 2022; Wang et al., 2023; Yu et al., 2023; Tsoy et al., 2024). However, those results do not consider agents' strategic manipulation for personal gains.

**JDP-based truthfulness approaches.** JDP-based approaches incentivize truthful behavior by injecting noise into algorithmic outputs, thereby masking the impact of any single agent's misreporting on the final model and promoting truthfulness (Han et al., 2015; Zhang et al., 2022). However, these approaches require a centralized server to collect local optimization variables from all agents to determine the needed noise amplitude, which makes it infeasible in a fully decentralized setting. Moreover, JDP-based approaches have to compromise convergence accuracy to ensure truthfulness (Chen et al., 2025), which is undesirable in accuracy-sensitive applications.

*Notations:* We use $\mathbb{R}^n$ to denote the set of $n$-dimensional real Euclidean space. We denote $\nabla F(\theta)$ as the gradient of $F(\theta)$ and $\mathbb{E}[\theta]$ as the expected value of a random variable $\theta$. We denote the set of $N$ agents by $[N]$, the neighboring set of agent $i$ as $\mathcal{N}_i$. We denote the coupling matrix by $W = \{w_{ij}\} \in \mathbb{R}^{N \times N}$, where $w_{ij} > 0$ if agent $j$ interacts with agent $i$, and $w_{ij} = 0$ otherwise. We define $w_{ii} = 1 - \sum_{j \in \mathcal{N}_i} w_{ij}$. We abbreviate "with respect to" as *w.r.t.* throughout the paper.

## 3 PROBLEM FORMULATION AND PRELIMINARIES

### 3.1 DECENTRALIZED LEARNING AND OPTIMIZATION

We consider $N \geq 2$ agents participating in decentralized learning and optimization, each possessing a private dataset whose distribution can be heterogeneous across the agents. The goal is for all agents

to cooperatively find a solution $\theta^*$ to the following stochastic optimization problem:

$$\min_{\theta \in \mathbb{R}^n} F(\theta) = \frac{1}{N} \sum_{i=1}^{N} f_i(\theta), \quad f_i(\theta) = \mathbb{E}_{\zeta_i \sim \mathcal{P}_i}[l(\theta; \zeta_i)], \tag{1}$$

where $\theta \in \mathbb{R}^n$ denotes a global model parameter and $\zeta_i$ is a random data sample of agent $i$ drawn from its local data distribution $\mathcal{P}_i$. The loss function $l(\theta; \zeta_i) : \mathbb{R}^n \times \mathbb{R}^n \mapsto \mathbb{R}$ is assumed to be differentiable in $\theta$ for every $\zeta_i$ and the local objective function $f_i(\theta)$ of agent $i$ is general convex.

In real-world applications, the data distribution $\mathcal{P}_i$ is typically unknown. Hence, each agent $i$ can only access a noisy estimate of the gradient $\nabla f_i(\theta_{i,t})$, computed at its current local model parameter $\theta_{i,t}$ using the available local data. For example, at each iteration $t$, agent $i$ samples a batch of $B \geq 1$ data points and computes a gradient estimate as $g_i(\theta_{i,t}) = \frac{1}{B} \sum_{j=1}^{B} \nabla l(\theta_{i,t}; \zeta_{ij})$. Using this gradient estimate $g_i(\theta_{i,t})$, along with the model parameters $\{\theta_{j,t}\}_{j \in \mathcal{N}_i}$ received from its neighbors, agent $i$ updates its local parameter according to a decentralized learning/optimization algorithm.

Existing decentralized learning/optimization algorithms (see, e.g., Lian et al. (2017); Li et al. (2019); Liu et al. (2019); Assran et al. (2019); Koloskova et al. (2020b); Amiri & Gündüz (2020); Kim et al. (2020); Pu et al. (2021); Bullins et al. (2021); Lin et al. (2021); Castiglia et al. (2021); Huang et al. (2022); Wang & Yang (2023); Ding et al. (2023); Bars et al. (2024); Huang et al. (2025); da Silva et al. (2025)) universally assume that participating agents are honest and behave truthfully. However, this assumption may be unrealistic in real-world scenarios, where agents can behave selfishly or strategically. For example, a strategic agent may amplify its gradient estimates to bias the final model parameter in favor of its own data distribution, or inject noise into its shared information to degrade the performance of other agents' models for competitive advantage. (We provide additional motivating examples to illustrate how agents can benefit from gradient manipulation in decentralized least squares and consensus-based decentralized mean estimation in Appendix B.1 and Appendix B.2, respectively.) Such strategic gradient manipulation can significantly degrade the learning performance of existing decentralized learning algorithms (as evidenced by our experimental results in Fig. 2).

Next, we discuss the classical decentralized stochastic gradient descent (SGD) in the presence of gradient manipulation by a strategic agent $i \in [N]$.

**Decentralized SGD in the presence of strategic behavior.** At each iteration $t$, each agent $i$ strategically chooses a manipulated gradient $m_{i,t}$, which, in general, is a function of agent $i$'s true gradient $g_i(\theta_{i,t})$. Using the manipulated gradient $m_{i,t}$ and the model parameters $\{\theta_{j,t}\}_{j \in \mathcal{N}_i}$ received from its neighbors, each agent $i$ updates its local model parameter according to the following Algorithm 1:

---

**Algorithm 1** Decentralized SGD in the presence of strategic behavior (from agent $i$'s perspective)

1: **Input:** Random initialization $\theta_{i,0} \in \mathbb{R}^n$; stepsize $\lambda_t > 0$.
2: Send $\theta_{i,0}$ to neighbors $j \in \mathcal{N}_i$ and receive $\theta_{j,0}$ from neighbors $j \in \mathcal{N}_i$.
3: **for** $t = 0, \dots, T$ **do**
4: $\quad \theta_{i,t+1} = \sum_{j \in \mathcal{N}_i \cup \{i\}} w_{ij} \theta_{j,t} - \lambda_t m_{i,t}$;
5: $\quad$ Send $\theta_{i,t+1}$ to neighbors $j \in \mathcal{N}_i$ and receive $\theta_{j,t+1}$ from neighbors $j \in \mathcal{N}_i$.
6: **end for**
7: **Output:** $x_{i,T}$ on agent $i$.

---

It is worth noting that we focus on gradient manipulation rather than model-parameter manipulation for two reasons. First, any manipulation of model parameters shared among agents effectively corresponds to some form of alteration in the gradient estimates, as proven in Corollary 1 in Appendix B.1. Second, gradient manipulation is the most direct and practically effective strategy for a strategic agent to increase its personal gain. Specifically, by upscaling its own gradient estimates, an agent can increase the influence of its local data on the cooperative learning process, thereby pulling the final model parameter closer to the minimizer of its local objective function and reducing its own cost. On the other hand, by injecting noise into its gradient estimates, an agent can reduce the usefulness of its data to its neighbors, degrading their model performance and gaining competitive advantage. In comparison, manipulating model parameters does not provide a clear strategic benefit. For these reasons, we focus on gradient manipulation as the primary form of untruthful or strategic behavior by participating agents.

To quantitatively analyze strategic interactions among agents, we adopt a game-theoretic framework that explicitly defines each agent's strategic behaviors, rewards, payments, and net utilities.

### 3.2 GAME-THEORETIC FRAMEWORK

**Strategic behaviors and action space.** A self-interested agent can enhance its individual outcome through two strategic behaviors: amplifying its gradient estimates to bias the final model parameter in favor of its own data distribution, and injecting noise to degrade the performance of other agents' model parameters for competitive advantage. Both strategic behaviors can be modeled as an agent's action at each iteration. Formally, we define the action space for each strategic agent $i \in [N]$ in decentralized learning/optimization as follows:

$$\mathcal{A}_i = \left\{ \alpha_i | \alpha_i(g_i(\theta)) = a_i g_i(\theta) + b_i \xi_i, \text{ with } a_i \geq 1 \text{ and } b_i \in \mathbb{R} \right\}, \tag{2}$$

where $g_i(\theta)$ represents agent $i$'s true gradient estimate and $\xi_i$ is a zero-mean noise vector with bounded variance. The scaling factor $a_i$ quantifies the degree of gradient amplification, while the noise factor $b_i$ specifies the magnitude of noise injection, both of which can be strategically chosen by agent $i$ at each iteration. For any action space $\mathcal{A}_i$, we assume that it includes the identity mapping, which maps $g_i$ to itself with probability one. Hence, truthfulness is always a feasible action.

The action space defined in equation 2 is designed to capture the strategic behavior of agents in decentralized learning rather than arbitrary malicious attacks. To this end, we consider $a_i \geq 1$ and ignore $a_i < 1$, because the latter typically reduces the influence of agent $i$'s local data on cooperative learning, making it a non-utility-improving choice for a rational agent. This focus on strategic, utility-driven behavior excludes general malicious attacks, which are typically disruptive and do not align with an agent's goal of improving its outcome within the learning framework. In addition, compared with the action spaces defined in existing results on federated learning, such as Dorner et al. (2023) that only considers noise injection (i.e., fixing $a_i = 1$) and Chakarov et al. (2024a) that only considers gradient amplification (i.e., fixing $b_i = 0$), our formulation accounts for a broader range of strategic manipulations.

**Rewards.** We denote a decentralized learning algorithm as $\mathcal{M}$. At each iteration $t$, agent $i$ chooses an action $\alpha_{i,t} \in \mathcal{A}_i$. This action produces a (manipulated) gradient $m_{i,t} = \alpha_{i,t}(g_i(\theta_{i,t}))$, which is then used by agent $i$ in the update step of $\mathcal{M}$. Considering $T + 1$ iterations of $\mathcal{M}$, we let $\boldsymbol{\alpha}_i = \{\alpha_{i,t}\}_{t=0}^T$ denote the action trajectory of agent $i$ from iteration 0 to iteration $T$, $\boldsymbol{\theta}_j = \{\theta_{j,t}\}_{t=0}^T$ denote the model-parameter trajectory of agent $j$, and $\boldsymbol{\theta}_{-i} = \{\boldsymbol{\theta}_j\}_{j \in \mathcal{N}_i}$ denote the collection of model-parameter trajectories received by agent $i$ from all its neighbors. Given an initial model parameter $\theta_{i,0}$, an action trajectory $\boldsymbol{\alpha}_i$, and a collection of model-parameter trajectories $\boldsymbol{\theta}_{-i}$, an implementation of $\mathcal{M}$ generates a final model parameter $\theta_{i,T+1} = \mathcal{M}(\theta_{i,0}, \boldsymbol{\alpha}_i, \boldsymbol{\theta}_{-i})$ for agent $i$. We denote the reward that agent $i$ obtains from its final objective (cost) function value as $R_i(f_i(\theta_{i,T+1}))$.

In a minimization problem (see equation 1), the reward function $R_i(f_i(\theta))$ increases as the objective function $f_i(\theta)$ decreases. Therefore, a self-interested agent can boost its reward by biasing the final solution toward the minimizer of its local objective function. Common choices for $R_i(f_i(\theta))$ include the linear function $R_i(f_i(\theta)) = -f_i(\theta)$ and the sigmoid-like function $R_i(f_i(\theta)) = (1 + e^{-1/f_i(\theta)})^{-1}$ (Chakarov et al., 2024b). We allow different agents to have different reward functions.

**Payments and net utilities.** To mitigate gradient manipulation by participating agents, we augment the decentralized learning protocol by introducing a payment mechanism (which can be efficiently computed and implemented in a fully decentralized manner between any pair of interacting agents, see details in Section 4). Furthermore, to incentivize agents to actively participate in decentralized learning (e.g., to discourage local-only learning), we introduce a regularization term that penalizes large deviations of each agent's learned model parameter from the global optimal solution $\theta^*$. We denote the augmented decentralized learning protocol (with a payment mechanism) by $\mathcal{M}_p$. The net utility of agent $i$ from executing $\mathcal{M}_p$ over $T + 1$ iterations is defined as follows:

$$U_{i,0 \to T}^{\mathcal{M}_p}(\boldsymbol{\alpha}_i, \boldsymbol{\alpha}_{-i}) = R_i(f_i(\theta_{i,T+1})) - K\|\theta_{i,T+1} - \theta^*\|^2 - \sum_{t=0}^T P_{i,t}, \tag{3}$$

where $K$ is an arbitrary positive penalty coefficient, $P_{i,t}$ is the total payment made by agent $i$ to all its neighbors, and $\boldsymbol{\alpha}_{-i} = \{\boldsymbol{\alpha}_j\}_{j \neq i}$ denotes the action trajectories of all agents except agent $i$.

We note that all existing incentive-based approaches for collaborative learning ensure truthfulness by incorporating a payment or penalty term into each agent's net utility. Without such payments, a self-interested agent can freely manipulate its gradients to reduce its own loss and increase its rewards, thereby distorting the collaborative learning process. Therefore, equation 3 includes the cumulative payments of each agent in its net utility. Accordingly, a rational agent must consider both its rewards and payments when maximizing its net utility.

Next, we introduce two truthfulness-related concepts in our game-theoretic framework.

**Definition 1** ($\delta$-truthful action (Chakarov et al., 2024a;b)). *For any given $\delta \geq 0$ and any $i \in [N]$, an action $\alpha_i \in \mathcal{A}_i$ (with $\mathcal{A}_i$ defined in equation 2) of agent $i$ is $\delta$-truthful if it satisfies $\mathbb{E}[\|\alpha_i(g_i(\theta)) - g_i(\theta)\|] \leq \delta$ for any $\theta \in \mathbb{R}^n$. In particular, the action $\alpha_i$ is fully truthful when $\delta = 0$.*

Definition 1 quantifies the truthfulness of an agent's action in collaborative learning/optimization. It can be seen that a smaller $\delta$ corresponds to a higher level of truthfulness in the agent's action.

**Definition 2** ($\varepsilon$-incentive compatibility (Nisan et al., 2007)). *For any given $\varepsilon \geq 0$, a decentralized learning protocol $\mathcal{M}_p$ is $\varepsilon$-incentive compatible if for all $i \in [N]$, $\mathbb{E}[U_{i,0 \to T}^{\mathcal{M}_p}(\boldsymbol{h}_i, \boldsymbol{h}_{-i})] \geq \mathbb{E}[U_{i,0 \to T}^{\mathcal{M}_p}(\boldsymbol{\alpha}_i, \boldsymbol{h}_{-i})] - \varepsilon$ holds for any arbitrary action trajectory $\boldsymbol{\alpha}_i$ of agent $i$, where $\boldsymbol{h}_i$ is the truthful action trajectory of agent $i$ and $\boldsymbol{h}_{-i}$ is truthful action trajectories of all agents except agent $i$.*

Definition 2 (also called $\varepsilon$-Bayesian-incentive compatibility) is a standard and commonly used notion in the existing incentive-compatibility literature (Deng et al., 2020; Yin et al., 2022; Chakarov et al., 2024a;b). It implies that if a decentralized learning protocol is $\varepsilon$-incentive compatible, then the expected net utility that an agent can gain from any (possibly untruthful) action trajectory is at most $\varepsilon$ greater than that obtained by being truthful in all iterations. Clearly, a smaller $\varepsilon$ corresponds to a lower gain that an untruthful agent can obtain. In addition, according to the definition of $\varepsilon$-Nash equilibrium in Huang et al. (2007) (also provided in Definition 6 in Appendix C.1), if a decentralized learning protocol is $\varepsilon$-incentive compatible, then the truthful action trajectory profile of all agents $\boldsymbol{h} = (\boldsymbol{h}_i, \boldsymbol{h}_{-i})$ forms an $\varepsilon$-Nash equilibrium (see Lemma 7 in Appendix C.2).

## 4 PAYMENT MECHANISM DESIGN FOR DECENTRALIZED LEARNING

In this section, we propose a fully decentralized payment mechanism (see Mechanism 1) to incentivize truthful behaviors of participating agents in decentralized learning and optimization.

---

**Mechanism 1** Fully decentralized payment mechanism (for interacting agents $i$ and $j$ at iteration $t$)

1: **Input:** $\theta_{\iota,t-1}, \theta_{\iota,t}, \theta_{\iota,t+1}$ for $\iota \in \{i,j\}$ available to both agents $i$ and $j$ under Algorithm 1 (note $\theta_{\iota,t+1}$ has been shared at the end of iteration $t$); initialization $\theta_{i,-1} = \theta_{j,-1} = \mathbf{0}_n$; $C_t > 0$.
2: Agents $i$ and $j$ simultaneously compute both $\Delta_{\theta_{i,t}} \triangleq \|\theta_{i,t+1} - 2\theta_{i,t} + \theta_{i,t-1}\|^2$ and $\Delta_{\theta_{j,t}} \triangleq \|\theta_{j,t+1} - 2\theta_{j,t} + \theta_{j,t-1}\|^2$.
3: **if** $\Delta_{\theta_{i,t}} \geq \Delta_{\theta_{j,t}}$ **then**
4:   Agent $i$ transfers $P_{i,t}^j = C_t(\Delta_{\theta_{i,t}} - \Delta_{\theta_{j,t}})$ to agent $j$.
5: **else**
6:   Agent $i$ receives $P_{j,t}^i = C_t(\Delta_{\theta_{j,t}} - \Delta_{\theta_{i,t}})$ from agent $j$.
7: **end if**

---

Mechanism 1 is implementable in a fully decentralized manner without the assistance of any server or aggregator. At each iteration $t$, if agent $i$ manipulates its gradient estimates such that its model-parameter increment $\|\theta_{i,t+1} - 2\theta_{i,t} + \theta_{i,t-1}\|$ exceeds that of its neighbor agent $j$, agent $i$ pays an amount $P_{i,t}^j > 0$ to agent $j$ (in this case, we denote the payment of agent $j$ as $P_{j,t}^i = -P_{i,t}^j$). Conversely, if agent $i$'s model-parameter increment $\|\theta_{i,t+1} - 2\theta_{i,t} + \theta_{i,t-1}\|$ is no greater than that of its neighbor agent $j$, it receives a payment of amount $P_{j,t}^i \geq 0$ from agent $j$ (in this case, we denote the payment of agent $i$ as $P_{i,t}^j = -P_{j,t}^i$). We emphasize that our payment mechanism can be readily applied to any first-order (gradient-based) decentralized learning and optimization algorithm. For an agent $i$, its total payment to all its neighbors at iteration $t$ is given by $P_{i,t} = \sum_{j \in \mathcal{N}_i} P_{i,t}^j$.

In Mechanism 1, with both agents $i$ and $j$ having access to $\theta_{\iota,t-1}$, $\theta_{\iota,t}$, and $\theta_{\iota,t+1}$ for $\iota \in \{i,j\}$ from the update of Algorithm 1 (note that $\theta_{\iota,t+1}$ has been shared at the end of iteration $t$), the two agents can cross-verify the computed payment value, making the mechanism robust to unilateral manipulation. This represents a significant advance compared with the payment mechanism in Angeli & Manfredi (2023) for server-assisted collaborative optimization, which requires all agents to truthfully report their local objective-function values for payment calculation—thereby creating a risk that strategic agents may manipulate the algorithmic update and the payment mechanism separately.

Our payment mechanism can effectively discourage agents from free-riding. Specifically, the model-parameter increment $\|\theta_{i,t+1} - 2\theta_{i,t} + \theta_{i,t-1}\|$ of agent $i$ depends on both the consensus errors (see the dynamics of $\Delta_{\theta_{i,t}}$ in equation 54) and the (sign-indefinite) local gradients. Consequently, even if agent $i$ uses a zero (or low) gradient, there is no guarantee that $\|\theta_{i,t+1} - 2\theta_{i,t} + \theta_{i,t-1}\|$ will be smaller than that of its neighbor $j$, meaning that leveraging zero gradients does not reliably increase agent $i$'s payment gains. In contrast, free-riding behavior (or using low gradients) invariably degrades agent $i$'s own reward $R_i(f_i(\theta_{i,T}))$, as it weakens the influence of agent $i$'s data in collaborative learning and ultimately leads to a worse final model for the agent itself. Hence, free-riding is not a utility-improving choice for a rational agent.

Our payment mechanism is conceptually inspired by the classical VCG mechanism (Vickrey, 1961; Clarke, 1971; Groves, 1973) but has several fundamental differences: 1) conventional VCG mechanisms require a central server to calculate and collect payments (Dave et al., 2021; Qian et al., 2021; Lyu et al., 2022; Angeli & Manfredi, 2023; Alon et al., 2024), whereas our mechanism operates in a pairwise fashion without reliance on any third party. As a result, it can be implemented in a fully decentralized manner; 2) conventional VCG mechanisms are typically designed for one-shot games. In contrast, our payment mechanism is naturally compatible with iterative algorithms, encompassing a wide range of decentralized learning methods. In fact, an iterative algorithm setting inherently forms a multi-stage game, where agents repeatedly adjust actions, posing challenges for both truthfulness and convergence analysis; and 3) conventional VCG mechanisms are not budget-balanced (see, e.g., Angeli & Manfredi (2023); Alon et al. (2024)), whereas our payment mechanism is budget-balanced, i.e., $\sum_{i=1}^{N} P_{i,t} = 0$, making it financially sustainable and scalable in practice.

The existing approaches most closely related to ours are the payment mechanisms proposed by Chakarov et al. (2024a;b). However, there are several fundamental differences: 1) our payment mechanism is implementable in a fully decentralized manner, and hence, is applicable to arbitrary connected communication graphs, whereas the mechanisms in Chakarov et al. (2024a;b) rely on a centralized server to aggregate gradients from all agents, and thus operate only under a centralized communication structure; and 2) our payment mechanism achieves $\varepsilon$-incentive compatibility with a finite $\varepsilon$ even in an infinite time horizon (see Theorem 2), whereas the incentive $\varepsilon$ in Chakarov et al. (2024a;b) becomes unbounded as the number of iterations tends to infinity (see Claim 23 in Chakarov et al. (2024a) or Theorem 5.1 in Chakarov et al. (2024b)), leading to a vanishing incentive compatibility guarantee over iterations.

## 5 THEORETICAL RESULTS

**Assumption 1.** *For any $i \in [N]$, $R_i(f_i(\theta))$ is $L_{R,i}$-Lipschitz continuous w.r.t. $\theta$ and $g_i(\theta)$ is $H_i$-Lipschitz continuous. Moreover, the stochastic gradient estimate $g_i(\theta)$ is unbiased and has bounded variance $\sigma_i^2$, i.e., $\mathbb{E}[g_i(\theta)] = \nabla f_i(\theta)$ and $\mathbb{E}[\|g_i(\theta) - \nabla f_i(\theta)\|^2] \leq \sigma_i^2$. In addition, for a general convex $f_i(\theta)$, we assume that $f_i(\theta)$ is $L_{f,i}$-Lipschitz continuous. However, this assumption is not required for a strongly convex $f_i(\theta)$. For the sake of notational simplicity, we denote $L_R = \max_{i \in [N]}\{L_{R,i}\}$, $H = \max_{i \in [N]}\{H_i\}$, $L_f = \max_{i \in [N]}\{L_{f,i}\}$, and $\sigma = \max_{i \in [N]}\{\sigma_i\}$.*

**Assumption 2.** *The weight matrix $W$ is symmetric and satisfies $W\mathbf{1}_N = \mathbf{1}_N$ and $\mathbf{1}_N^\top W = \mathbf{1}_N^\top$. We assume $\rho = \max\{|\pi_2|, |\pi_N|\} < 1$, where $\pi_N \leq \cdots \leq \pi_2 < \pi_1 = 1$ denote the eigenvalues of $W$.*

Assumption 1 is standard and commonly used in the decentralized stochastic optimization/learning literature (Ma et al., 2015; Chen et al., 2017; Yin et al., 2018; Koloskova et al., 2020a; Sun et al., 2022; Beznosikov et al., 2022). It is worth noting that we allow $f_i(\theta)$ to be convex, which is more general than the strongly convex assumption used in existing truthfulness results in Angeli & Manfredi (2023); Dorner et al. (2023); Chakarov et al. (2024a). Assumption 2 ensures that the communication graph is connected (Pu & Nedić, 2021).

## 5.1 CONVERGENCE RATE ANALYSIS

We first prove that, under Mechanism 1, the incentive for a strategic agent in Algorithm 1 to deviate from truthful behavior diminishes to zero. We then analyze the convergence rates of Algorithm 1 in the presence of strategic behaviors, for strongly convex and general convex global objective functions, respectively.

**Lemma 1.** *Under Assumptions 1 and 2, for any $\delta > 0$, if we set $\lambda_t = \frac{\lambda_0}{(t+1)^v}$, $C_t = \frac{C_0 \kappa_t^2}{\delta^2 (t+1)^{-2v}}$ with $\kappa_t = \frac{1}{(t+1)^r}$, $v \in (\frac{1}{2}, \frac{2}{3})$, $r \in (\frac{1}{2}, v)$, and $C_0$ given in equation 72, then under Mechanism 1, the optimal action for any agent $i$ in Algorithm 1 is $\kappa_t \delta$-truthful with respect to its neighboring agents' actual gradients. Moreover, as the number of iterations tends to infinity, the optimal action for agent $i$ is fully truthful, i.e., an agent will have zero incentive to deviate from truthful behavior.*

Lemma 1 proves that when the neighbors of agent $i$ are truthful, the incentive for agent $i$ to deviate from truthful behavior converges to zero. This stands in sharp contrast to existing payment mechanisms in Chakarov et al. (2024a;b) for federated learning, which guarantees only a bounded—yet non-diminishing—incentive for untruthful behavior at each iteration, thus leaving agents with a persistent motive to act untruthfully. Furthermore, Lemma 1 implies that by choosing an arbitrarily small $\delta > 0$, we can ensure that the optimal action of each agent $i$ is arbitrarily close to being fully truthful at every iteration. It is worth noting that since the differences in model-parameter increments in Mechanism 1 diminish to zero, we can ensure $\lim_{t \to \infty} \mathbb{E}[P_{i,t}] = 0$ (as shown in Corollary 3 in Appendix E.4). This guarantees that no payment is required from agent $i$ when it behaves truthfully as the number of iterations tends to infinity.

**Theorem 1** (Convergence rate). *We denote $\theta^*$ as a solution to the problem in equation 1. Under our Mechanism 1 and the conditions in Lemma 1, for any $i \in [N]$, $\delta > 0$, and $T \geq 0$, the following results hold for Algorithm 1 in the presence of strategic behaviors:*

*(i) if $F(\theta)$ is $\mu$-strongly convex (not necessarily Lipschitz continuous), then we have*

$$\mathbb{E}[\|\theta_{i,T} - \theta^*\|^2] \leq \mathcal{O}\left(\frac{H^2(\sigma^2 + \delta^2)}{\mu(1-\rho)^2(T+1)^v}\right); \tag{4}$$

*(ii) if $F(\theta)$ is general convex, then we have*

$$\frac{1}{T+1}\sum_{t=0}^{T}\mathbb{E}[F(\theta_{i,t}) - F(\theta^*)] \leq \mathcal{O}\left(\frac{H^2(\sigma^2 + L_f^2 + \delta^2)}{(1-\rho)^2(T+1)^{1-v}}\right), \tag{5}$$

*where $H$, $\sigma$, and $L_f$ are from Assumption 1 and $\rho = \max\{|\pi_2|, |\pi_N|\} < 1$ is from Assumption 2.*

Theorem 1 proves that, even in the presence of strategic behaviors, our proposed Mechanism 1 ensures convergence to an exact optimal solution $\theta^*$ to the problem in equation 1 at rates $\mathcal{O}(T^{-v})$ and $\mathcal{O}(T^{-(1-v)})$ for strongly convex and general convex $F(\theta)$, respectively. It is broader than existing truthfulness results in Dave et al. (2021); Angeli & Manfredi (2023); Dorner et al. (2023); Chakarov et al. (2024a;b) that focus solely on the strongly convex case. Moreover, our results are in stark contrast to existing JDP-based truthfulness results in Han et al. (2015); Hale & Egerstedt (2015); Zhang et al. (2022); Chen et al. (2025) and incentive-based truthfulness results in Dorner et al. (2023); Chakarov et al. (2024a;b), all of which are subject to optimization errors. In addition, we provide a computational-complexity analysis of Algorithm 1 under Mechanism 1 in Appendix E.3.

## 5.2 INCENTIVE-COMPATIBILITY ANALYSIS

In addition to achieving accurate convergence, our fully decentralized payment mechanism also simultaneously ensures that Algorithm 1 is $\varepsilon$-incentive compatible.

**Theorem 2** (Incentive compatibility). *Under our fully decentralized payment mechanism and the conditions in Lemma 1, Algorithm 1 is $\varepsilon$-incentive compatible, regardless of whether $F(\theta)$ is general convex or strongly convex. Namely, for any $i \in [N]$, $\delta > 0$, and $T \geq 0$ (which includes the case of $T = \infty$), the following inequality always holds:*

$$\mathbb{E}[U_{i,0 \to T}^{\mathcal{M}_p}(\boldsymbol{\alpha}_i, \boldsymbol{h}_{-i}) - U_{i,0 \to T}^{\mathcal{M}_p}(\boldsymbol{h}_i, \boldsymbol{h}_{-i})] \leq \varepsilon, \tag{6}$$

*with $U_{i,0 \to T}^{\mathcal{M}_p}(\boldsymbol{\alpha}_i, \boldsymbol{h}_{-i})$ and $U_{i,0 \to T}^{\mathcal{M}_p}(\boldsymbol{h}_i, \boldsymbol{h}_{-i})$ defined in equation 3 and $\varepsilon$ given by $\varepsilon = \mathcal{O}\left(\frac{L_R \delta}{v+r-1}\right)$.*

Theorem 2 ensures that the cumulative gain from an agent $i$'s untruthful behaviors in Algorithm 1 remains finite, even when $T \to \infty$. This contrasts with existing truthfulness results for federated learning (e.g., Chakarov et al. (2024a;b)), where $\varepsilon$ explodes as the iteration proceeds, implying that truthfulness/incentive compatibility will eventually be lost.

Existing incentive-based truthfulness results for server-assisted federated learning in, e.g., Dorner et al. (2023); Chakarov et al. (2024a;b), do not provide **simultaneous** guarantees for both $\varepsilon$-incentive compatibility and accurate convergence. Specifically, the convergence analysis in Dorner et al. (2023) requires two conditions: (i) $P(\exists t \leq T : \Pi_W(\theta_t^s - \gamma_t \bar{m}_t) \neq \theta_t^s - \gamma_t \bar{m}_t) \in \mathcal{O}(\frac{1}{NT})$ and (ii) the boundedness of $W$ (see Theorem 6.1 in Dorner et al. (2023)), where $\theta_t^s$ is the model parameter computed by the centralized server, $W$ is a projection set, $\gamma_t$ is the stepsize, and $\bar{m}_t$ is the average (manipulated) gradients reported by all agents. Moreover, in the Appendix section "Discussion on the projection assumptions", they state that Condition (i) can be guaranteed when $W$ grows at a rate of $\Omega(T)$ for general strongly convex functions, which is at odds with the boundedness requirement on $W$ in Condition (ii) when $T$ tends to infinity (their convergence error $\mathcal{O}(\frac{1+M+\varepsilon^2}{NT}) + \mathcal{O}(\frac{1}{T^2})$ in Theorem 6.1 is strictly larger than 0 unless $T$ is allowed to approach infinity). Therefore, they did not provide a method for ensuring that both conditions hold simultaneously under general strongly convex objectives. A similar issue also exists in Chakarov et al. (2024a) (see Theorem 9 and footnote 2 therein). Although the convergence analysis in Chakarov et al. (2024b) removes these two conditions, its definition $G = \sum_{t=1}^T \gamma_t \sqrt{\mathcal{C}_t}$ in Theorem 5.1 implies that $\varepsilon = \mathcal{O}(G)$ is finite only when $T$ is finite, indicating that both its $\varepsilon$-incentive compatibility and convergence statements fail to hold in an infinite time horizon.

## 6 EXPERIMENTS

We evaluate the effectiveness of our truthful mechanism using three representative decentralized machine learning tasks: image classification on the FeMNIST dataset, sentiment analysis on the Sent140 dataset, and next-character prediction on the Shakespeare dataset. The used datasets are from LEAF (Caldas et al., 2018). All these tasks involve nonconvex objective functions. We considered five agents connected in a circle, where each agent communicates only with its two immediate neighbors. For the coupling matrix $W$, we set $w_{ij} = 0.3$ if agents $i$ and $j$ are neighbors, and $w_{ij} = 0$ otherwise. For each experiment, we distributed the data across agents using a Dirichlet distribution to ensure heterogeneity. Visualizations of the heterogeneous data distributions can be found in Appendix A.1. In each experiment, we randomly divided the agents into two groups, A and B, containing two and three agents, respectively. Each agent $i$ in group A participates in Algorithm 1 using manipulated gradients $m_{i,t}^A = a_{i,t}^A g_i(\theta_{i,t}) + b_{i,t}^A \xi_{i,t}$, where each element of $\xi_{i,t}$ is drawn from a Laplace distribution with zero mean and unit variance. Agents in group B act truthfully. It is clear that when $a_{i,t}^A = 1$ and $b_{i,t}^A = 0$ hold for all $i \in [N]$ and $t \geq 0$, all agents act truthfully. In all experiments, the optimal model parameter $\theta^*$ in equation 3 was obtained by executing centralized SGD over $10,000$ iterations and the penalty coefficient was set to $K = 10^{-5}$. For each experiment, we first compared the average net utility of agents in group A under varying scaling factors $a_A$ for different payment coefficients $C$. Then, we evaluated the test accuracies in three cases: 1) Algorithm 1 without manipulation (called DSGD (no strategic behaviors)), 2) Algorithm 1 with manipulation under Mechanism 1 (called Algorithm 1 with payment), and 3) Algorithm 1 with manipulation without Mechanism 1 (called Algorithm 1 without payment). In this comparison, each agent in group A strategically selects $(a_{i,t}^A, b_{i,t}^A)$ to empirically maximize its net utility under a preset $C_t = \frac{10^{-6} \kappa_t^2}{\delta^2 (t+1)^{-2v}}$, where $\kappa_t$ and $\delta$ are specified in each experiment below. Detailed experimental setups are given in Appendix A.1 and additional results on noise factors $b_A$ and training loss are given in Appendix A.2. The code for all experiments is available online[3].

**Image classification on the FeMNIST dataset.** We train a convolutional neural network (CNN) using the architecture provided by Caldas et al. (2018). Training is conducted on $81,785$ data samples from $3,597$ writers, which are partitioned among agents using a Dirichlet distribution with parameter $\beta = 0.5$. We use a batch size of 32. We set the learning rate (stepsize) as $\lambda_t = 0.1(t+1)^{-0.55}$ and the parameters in payment coefficient $C_t$ as $\kappa_t = (t+1)^{-0.51}$ and $\delta = 10^{-4}$. These parameter choices satisfy all the conditions in Lemma 1, Theorem 1, and Theorem 2.

---

[3]https://anonymous.4open.science/r/Truthfulness-in-D-Learning/README.md

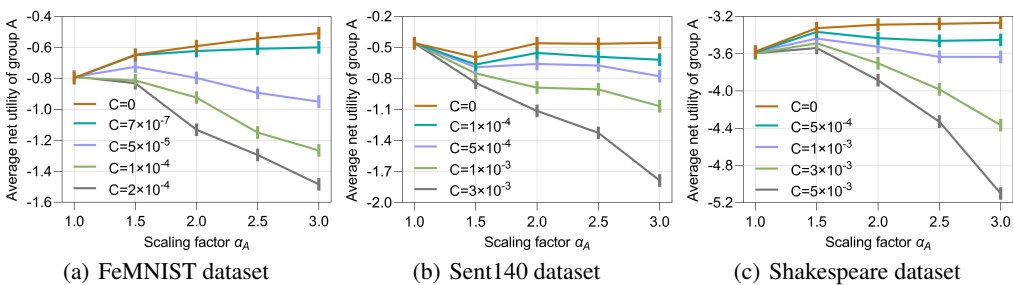

Figure 1: Average net utilities of group-A agents under varying scaling factors $a_A$ (with $b_A = 0$) for different payment coefficient $C$. The error bars represent standard errors over 10 runs.

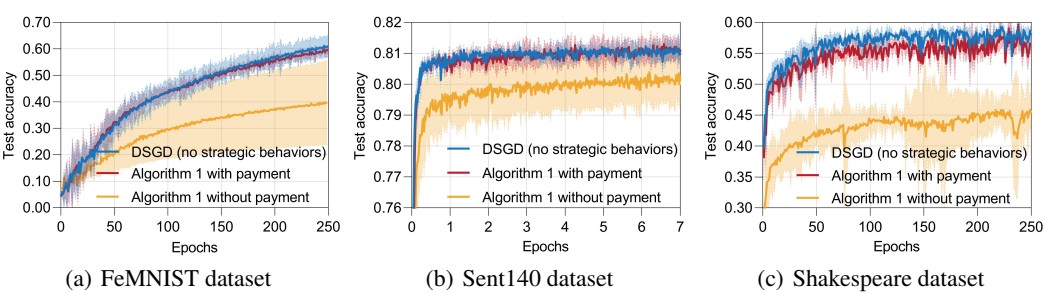

Figure 2: Comparison of test accuracies over epochs. The $95\%$ confidence intervals were computed from three independent runs with random seeds 42, 126, and 1010.

**Sentiment analysis on the Sent140 dataset.** We train a sentiment classifier based on a multi-layer perceptron (Go et al., 2009). The input (BERT embedding) to this classifier is precomputed using a frozen BERT-base model (Devlin et al., 2019). Training is conducted on $160,049$ tweets from $660,120$ users, which are partitioned among agents using a Dirichlet distribution with parameter $\beta = 10$. We use a batch size of $64$. We set the learning rate (stepsize) as $\lambda_t = 0.1(t + 1)^{-0.6}$ and the parameters in payment coefficient $C_t$ as $\kappa_t = (t + 1)^{-0.51}$ and $\delta = 10^{-4}$.

**Next-character prediction on the Shakespeare dataset.** We train a long short-term memory network (Wang et al., 2020) using the Shakespeare dataset with $398,202$ sequences from $1,080$ users. The experimental setups are the same as those used in the previous FeMNIST experiment.

In Fig. 1, the top orange line ($C = 0$) shows that without our payment mechanism, increasing $a_A$ raises the average net utility of group-A agents. However, introducing payments ($C > 0$) effectively reduces the gains from such strategic behaviors by group-A agents. Moreover, a large $C$ makes truthful participation the optimal action for group-A agents. Fig. 2 shows that the strategic behavior of agents decreases the test accuracy in conventional decentralized learning algorithms, whereas our payment mechanism mitigates this degradation. This demonstrates the effectiveness of our approach in preserving the learning accuracy of Algorithm 1 despite strategic manipulation.

## 7 CONCLUSION

We have proposed the first fully decentralized payment mechanism that incentivizes truthful behaviors of strategic agents during decentralized learning and optimization, without relying on any centralized server or aggregator. This represents a significant advance since all existing truthfulness approaches require the assistance of a centralized server in computation or execution. Our payment mechanism ensures that the cumulative gain from strategic manipulation remains finite, even over an infinite time horizon—a property unattainable in most existing truthfulness results. Moreover, unlike most existing truthfulness results, our payment mechanism is budget-balanced and guarantees accurate convergence of Algorithm 1 under strategic manipulation. The results apply to general convex and strongly convex global objective functions, making them more general than existing work, which focuses only on the strongly convex cases. Experimental results on three decentralized machine learning applications confirm the effectiveness of our approach.

**Ethics statement.**   All authors declare no conflicts of interest and no ethical issues in this work.

**Reproducibility statement.**   All authors confirm the reproducibility of both the theoretical and experimental results. The code for all experiments is available online at `https://anonymous.4open.science/r/Truthfulness-in-D-Learning/README.md` . Detailed descriptions of the experimental settings are provided in the main text and Appendix. Theoretical assumptions are clearly stated, and complete proofs of all results are included in the Appendix.

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

APPENDIX

# A    EXPERIMENTAL SETUPS AND ADDITIONAL EXPERIMENTAL RESULTS

## A.1    EXPERIMENTAL SETUPS

**Heterogeneous data distribution.** The heatmaps in Fig. 3 visualize the heterogeneous data distributions generated by the Dirichlet partitioning used in our experiments, where $\beta$ controls the degree of data heterogeneity. We set $\beta = 0.5$ for the FeMNIST and Shakespeare datasets, and $\beta = 10$ for the Sent140 dataset. The horizontal axis denotes agent IDs, the vertical axis denotes class IDs, and the color intensity indicates the proportion of samples of each class assigned to each agent.

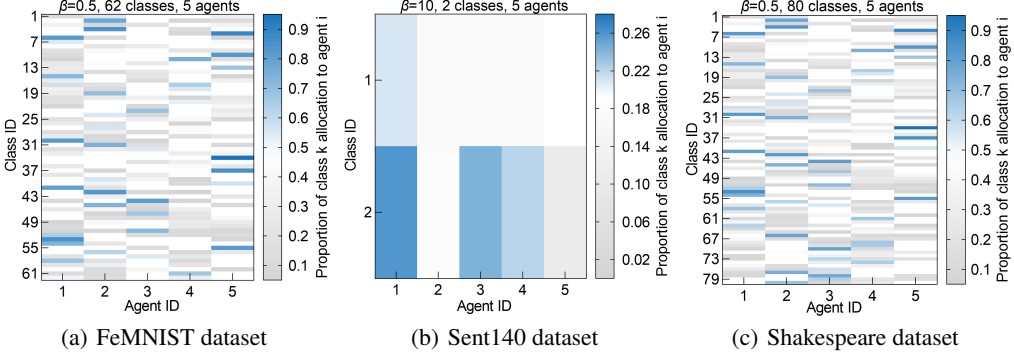

(a) FeMNIST dataset      (b) Sent140 dataset      (c) Shakespeare dataset

Figure 3: Visualization of heterogeneous data distributions among the five agents for the FeMNIST, Sent140, and Shakespeare datasets, respectively.

For all experiments, we used FedLab (Zeng et al., 2023) to migrate the dataset pipeline from LEAF's TensorFlow workflow (Caldas et al., 2018) to a PyTorch workflow (Paszke et al., 2019). In addition, each agent's local dataset was split into $90\%$ for training and $10\%$ for testing.

**FeMNIST.** The FeMNIST dataset is a variant of EMNIST, which consists of $817,851$ grayscale handwritten characters of size $28 \times 28$ across 62 classes (digits 0–9, uppercase letters A–Z, lowercase letters a–z). We performed training using $10\%$ of the entire dataset while preserving all 62 classes. In this experiment, we trained a two-layer convolutional neural network (CNN), which consists of two convolutional layers with 32 and 64 filters, respectively (both using $5 \times 5$ kernels), each followed by ReLU activation and $2 \times 2$ max pooling. The extracted features are then flattened and fed into a fully connected layer with 2048 hidden units and a final output layer of 62 classes.

**Sent140.** The Sent140 dataset consists of $1,600,498$ sentiment-labeled tweets across two classes (positive and negative). We performed training using $10\%$ of the entire dataset. For neural network training, we first employed a frozen BERT encoder (Devlin et al., 2019) to extract 768-dimensional sentence embeddings by mean-pooling the last hidden states under the attention mask (with a maximum sequence length of 128). These precomputed embeddings are stored for each agent and then used as input to a two-layer multi-layer perceptron (MLP) classifier (Go et al., 2009). Specifically, the classifier comprises a linear layer mapping the 768-dimensional input to a hidden layer of size 384, followed by a ReLU activation and a dropout layer with rate 0.1. The output of the hidden layer is then passed through a final linear layer to produce logits for classification. This architecture enables efficient sentiment classification based on fixed BERT representations.

**Shakespeare.** The Shakespeare dataset consists of lines from plays written by William Shakespeare, formulated as a next-character prediction task over a vocabulary of 80 characters. In this experiment, we performed training with a two-layer long short-term memory (LSTM) network (Wang et al., 2020) using $10\%$ of the entire dataset, which contains $3,982,028$ sequences. Each input character is first mapped to an 8-dimensional embedding vector. The embedded sequence is then processed by a two-layer LSTM, with each layer comprising 256 hidden units and a dropout rate of 0.5 between layers. The output from the final LSTM layer at the last time step is passed through a fully connected layer to produce logits over the 80-character vocabulary. This model setup enables the network to capture the sequential dependencies in Shakespearean text for effective character-level prediction.

**Hardware and computing resources.** All experiments were conducted on both Windows 11 and Ubuntu 22.04 LTS operating systems, using a workstation equipped with a 32-core CPU, 32 GB RAM, and a single NVIDIA GeForce RTX 4090 GPU with 24 GB VRAM.

## A.2    ADDITIONAL EXPERIMENTS

**Experimental results on training loss.** We evaluated the training loss in three cases: 1) Algorithm 1 without manipulation (called DSGD (no strategic behaviors)), 2) Algorithm 1 with manipulation under our payment mechanism (called Algorithm 1 with payment), and 3) Algorithm 1 with manipulation without our payment mechanism (called Algorithm 1 without payment). All parameter setups in this comparison are the same as those used for the test accuracy results in Fig. 2 of the main text.

Fig. 4 shows that the strategic behavior of agents increases the (global) training loss in conventional decentralized learning algorithms, whereas our payment mechanism mitigates this increase, yielding performance that is closer to the baseline (i.e., DSGD (no strategic behaviors)). This demonstrates the effectiveness of our approach in preserving the training accuracy of Algorithm 1 under strategic manipulation.

**Experimental results on varying noise factors $b_A$.** To evaluate the performance of our payment mechanism in scenarios where strategic agents inject noise into their gradient estimates in Algorithm 1, we conducted additional experiments to compare the average net utility of agents in group A under varying noise factors $b_A$ for different scaling factors $a_A$ and payment coefficients $C$.

Figs.5(a)–5(c) and Figs.5(g)–5(i) depict the average net utility of agents in group A on the FeMNIST and Shakespeare datasets, respectively (note that the FeMNIST and Shakespeare datasets are partitioned using a small Dirichlet parameter $\beta = 0.5$, resulting in high data heterogeneity among agents). These results show that in the absence of our payment mechanism ($C = 0$), increasing the noise factor $b_A$ raises the average net utility of group-A agents. However, introducing the payment

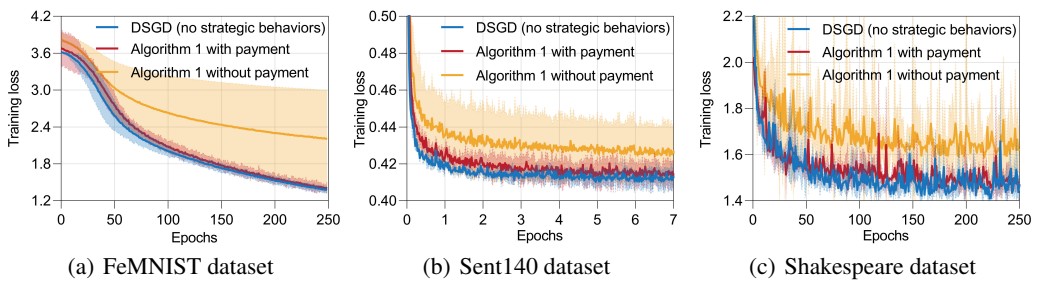

(a) FeMNIST dataset      (b) Sent140 dataset      (c) Shakespeare dataset

Figure 4: Comparison of training losses over epochs. The $95\%$ confidence intervals were computed from three independent runs with random seeds 42, 126, and 1010.

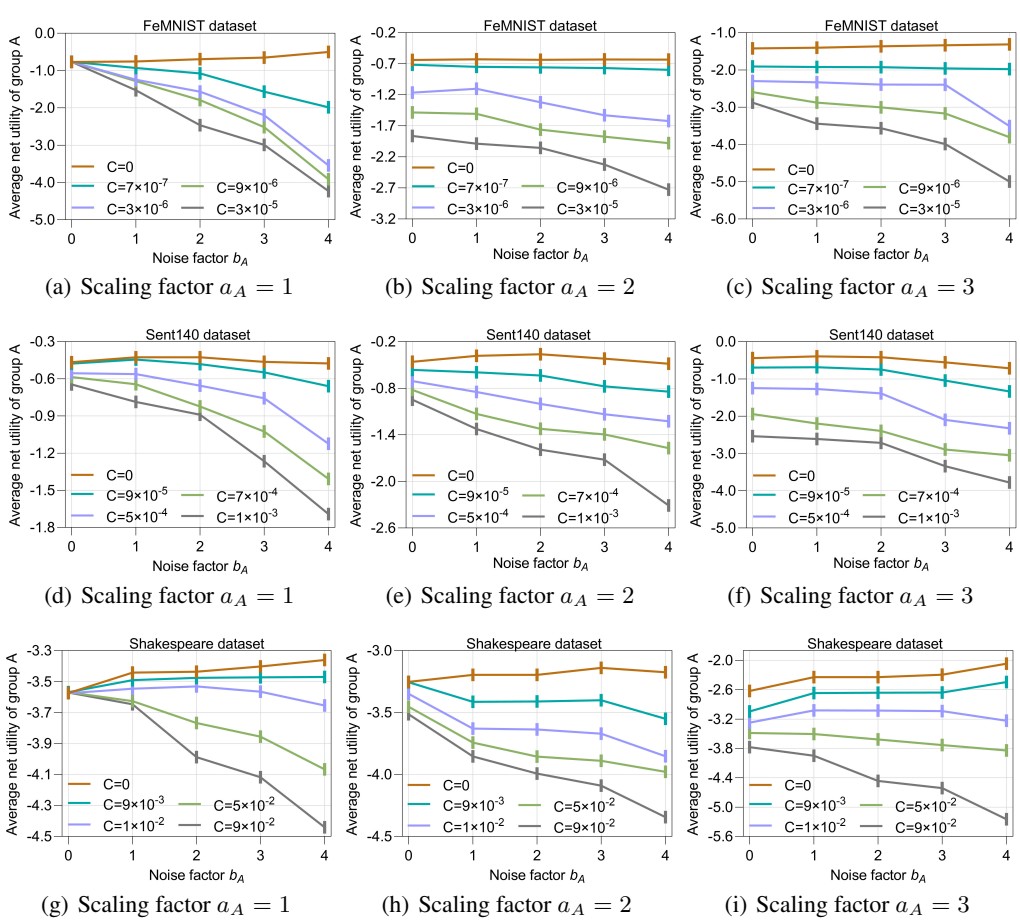

(a) Scaling factor $a_A = 1$    (b) Scaling factor $a_A = 2$    (c) Scaling factor $a_A = 3$

(d) Scaling factor $a_A = 1$    (e) Scaling factor $a_A = 2$    (f) Scaling factor $a_A = 3$

(g) Scaling factor $a_A = 1$    (h) Scaling factor $a_A = 2$    (i) Scaling factor $a_A = 3$

Figure 5: Average net utilities of group-A agents under varying noise factors $b_A$ for different scaling factors $a_A$ and payment coefficients $C$. The error bars represent standard errors over 10 runs.

mechanism ($C > 0$) effectively reduces the gains from such strategic behaviors by group-A agents. Figs. 5(d)-5(f) show the average net utility of agents in group A on the Sent140 dataset (which is partitioned using a large Dirichlet parameter $\beta = 10$, leading to nearly homogeneous data distributions among agents). In this case, increasing $b_A$ actually leads to a decrease in the net utility of group-A agents. This is because when data heterogeneity is low, the local optima of group-A agents' objective functions are close to the global optimum, and hence, injecting noise degrades the global model's performance and also reduces the rewards obtained by group-A agents.

Fig. 5 illustrates that data heterogeneity in decentralized machine learning can significantly influence agents' incentives to manipulate their gradient updates. Specifically, high data heterogeneity among agents creates strong incentives for gradient manipulation, as agents stand to gain more by deviating from truthful behavior. In contrast, in homogeneous settings, injecting noise provides no advantage and can even be detrimental to both individual and global learning performance.

## B MOTIVATING EXAMPLES AND ADDITIONAL LITERATURE COMPARISON

### B.1 MOTIVATING EXAMPLES

In this subsection, we first present a simple numerical example to illustrate how an agent's gradient manipulation can benefit itself while worsening the performance of other agents and increasing the global cost. Next, we consider a linear regression problem with stochastic least-squares objective functions to demonstrate that by amplifying its local gradient estimates, an agent can bias the global optimum toward its own local optimum, thereby reducing its local objective function value. Furthermore, we provide Corollary 1, which proves that any manipulation of model parameters shared among agents essentially corresponds to some form of modification in the gradient estimates.

**Numerical example.** We consider three agents with local objective functions $f_1(\theta) = \theta^2$, $f_2(\theta) = (\theta-2)^2$, and $f_3(\theta) = (\theta-4)^2$. The corresponding global objective function is $F(\theta) = \theta^2 - 4\theta + \frac{20}{3}$ with gradient $g(\theta) = 2\theta - 4$. The global optimum is achieved at $\theta^* = 2$. To illustrate the impact of gradient manipulation, we assume that agent 1 modifies its gradient as $m_1(\theta) = 2g_1(\theta) + 1 = 4\theta + 1$ while the other agents act truthfully. The resulting global gradient becomes $g'(\theta) = \frac{8}{3}\theta - \frac{11}{3}$, which yields a new global optimum at $\theta'^* = \frac{11}{8} = 1.375$. In this case, we have that agent 1 benefits from manipulation due to $f_1(\theta'^*) = (1.375)^2 \approx 1.89 < f_1(\theta^*) = 2^2 = 4$. In contrast, agent 2 and agent 3 experience worse outcomes due to $f_2(\theta'^*) = (1.375 - 2)^2 \approx 0.39 > f_2(\theta^*) = (2-2)^2 = 0$ and $f_3(\theta'^*) = (1.375 - 4)^2 \approx 6.89 > f_3(\theta^*) = (2-4)^2 = 4$. Moreover, the gradient manipulation by agent 1 increases the global cost from $F(\theta^*) \approx 2.67$ to $F(\theta'^*) \approx 3.06$.

**Least squares problem.** We consider a decentralized least squares problem where $N$ agents cooperatively find an optimal solution $\theta^*$ to the following stochastic optimization problem:

$$\min_{\theta \in \mathbb{R}^n} \quad F(\theta) = \frac{1}{N}\sum_{i=1}^{N} f_i(\theta), \quad f_i(\theta) = \mathbb{E}\left[(u_i^\top \theta - v_i)^2\right], \tag{7}$$

where $u_i \in \mathbb{R}^n$ denotes a feature vector that is independently and identically drawn from an unknown distribution with zero mean and a positive definite covariance matrix, i.e., $\mathbb{E}[u_i] = 0$ and $\mathbb{E}[u_i u_i^\top] = \Sigma \succ 0$. The label $v_i \in \mathbb{R}$ is generated according to the linear model $v_i = u_i^\top z_i + \xi_i$, where $z_i$ represents agent $i$'s predefined local target and $\xi_i$ denotes zero-mean noise independent of $u_i$ with variance $\sigma_\xi^2$.

The gradient of the global objective function satisfies

$$g(\theta) = \frac{1}{N}\sum_{i=1}^{N} 2\mathbb{E}[u_i u_i^\top](\theta - z_i) = 2\Sigma\left(\theta - \frac{1}{N}\sum_{i=1}^{N} z_i\right),$$

which implies that the optimal solution satisfies

$$\theta^* = \frac{1}{N}\sum_{i=1}^{N} z_i \triangleq \bar{z}. \tag{8}$$

To study the effect of gradient manipulation, we assume that agent $i$ deviates from truthful behavior by scaling its gradient estimate $g_i(\theta) = 2\Sigma(\theta - z_i)$ by some scalar $a_i > 1$, while all other agents act truthfully. Then, the gradient of the global objective function becomes

$$g'(\theta) = 2\Sigma\left(\frac{(a_i + N - 1)\theta}{N} - \frac{a_i z_i + \sum_{j \neq i} z_j}{N}\right),$$

which leads to the optimal solution becoming

$$\theta'^* = \frac{a_i - 1}{a_i + N - 1}z_i + \frac{N}{a_i + N - 1}\bar{z}, \quad \text{with } a_i > 1. \tag{9}$$

From equation 8 and equation 9, we have

$$\|\theta'^* - z_i\| = \frac{N}{a_i + N - 1}\|\bar{z} - z_i\| < \|\bar{z} - z_i\| = \|\theta^* - z_i\|,$$

for any $a_i > 1$, which implies that the optimal solution $\theta'^*$ is closer to agent $i$'s local target $z_i$ (and is also closer to agent $i$'s local optimum $\theta_i^*$ due to $\theta_i^* = z_i$) than the original optimal solution $\theta^*$. Moreover, a larger $a_i$ makes $\theta'^*$ closer to $z_i$.

Furthermore, since $\theta'^*$ is closer to $z_i$ than $\theta^*$ and $\Sigma$ is positive definite, we have

$$f_i(\theta'^*) = (\theta'^* - z_i)^\top \Sigma(\theta'^* - z_i) + \sigma^2 < f_i(\theta^*) = (\theta^* - z_i)^\top \Sigma(\theta^* - z_i) + \sigma^2,$$

which demonstrates that by amplifying its local gradient estimates, agent $i$ can bias the global optimum toward its own local optimum, thereby reducing its local objective function value.

In addition to decentralized least squares problems, similar truthfulness issues also emerge in decentralized mean estimation problems (see the next Section B.2 for details), decentralized ridge regression problems (Pu & Nedić, 2021), and many other decentralized learning scenarios.

Next, we present Corollary 1 to demonstrate that any manipulation of model parameters shared among agents in Algorithm 1 corresponds to some form of alteration in the gradient estimates.

**Corollary 1.** *For any agent $i \in [N]$, any manipulation of the model parameters that it shares with its neighbors in Algorithm 1 corresponds to some form of alteration in the gradient estimates.*

*Proof.* We first consider the conventional decentralized SGD algorithm as follows:

$$\theta_{i,t+1} = \sum_{j \in \mathcal{N}_i \cup \{i\}} w_{ij}\theta_{j,t} - \lambda_t g_i(\theta_{i,t}). \tag{10}$$

We assume that agent $i$ does not share its true model parameter $\theta_{i,t}$, but instead shares a manipulated model parameter $\tilde{\theta}_{i,t} = \hat{\alpha}_{i,t}(\theta_{i,t})$ with its neighbors, where $\hat{\alpha}_{i,t}$ represents an arbitrary action chosen by agent $i$ at iteration $t$. Then, for any neighbor $j \in \mathcal{N}_i$ of agent $i$, its update rule from equation 10 becomes

$$\begin{aligned}
\theta_{j,t+1} &= \sum_{l \in \mathcal{N}_j \setminus \{i\}} w_{jl}\theta_{l,t} + w_{ji}\tilde{\theta}_{i,t} - \lambda_t g_j(\theta_{j,t}) \\
&= \sum_{l \in \mathcal{N}_j} w_{jl}\theta_{l,t} - \lambda_t g_j(\theta_{j,t}) + w_{ji}(\hat{\alpha}_{i,t}(\theta_{i,t}) - \theta_{i,t}),
\end{aligned} \tag{11}$$

which implies that an additional term $w_{ji}(\hat{\alpha}_{i,t}(\theta_{i,t}) - \theta_{i,t})$ arises in the decentralized SGD update rule implemented by the neighbor $j \in \mathcal{N}_i$.

Substituting equation 11 into equation 10, we obtain

$$\begin{aligned}
\theta_{i,t+1} &= \sum_{j \in \mathcal{N}_i \cup \{i\}} w_{ij}\theta_{j,t} - \lambda_t g_i(\theta_{i,t}) + \sum_{j \in \mathcal{N}_i \cup \{i\}} (w_{ji}(\hat{\alpha}_{i,t-1}(\theta_{i,t-1}) - \theta_{i,t-1})) \\
&= \sum_{j \in \mathcal{N}_i \cup \{i\}} w_{ij}\theta_{j,t} - \lambda_t \alpha_{i,t}(g_i(\theta_{i,t})),
\end{aligned} \tag{12}$$

with $\alpha_{i,t}(g_i(\theta_{i,t})) = g_i(\theta_{i,t}) - \frac{\sum_{j \in \mathcal{N}_i \cup \{i\}}(w_{ji}(\hat{\alpha}_{i,t-1}(\theta_{i,t-1}) - \theta_{i,t-1}))}{\lambda_{t-1}}$. Equation 12 proves Corollary 1. $\square$

### B.2 TRUTHFULNESS IN CONSENSUS-BASED DECENTRALIZED MEAN ESTIMATION

We first clarify the fundamental difference between server-assisted and consensus-based decentralized mean estimations. We then rigorously prove that a strategic agent can benefit from manipulating its gradient estimates in consensus-based decentralized mean estimation.

**Fundamental differences between server-assisted and consensus-based decentralized mean estimations.** Existing truthfulness results for server-assisted mean estimation (see, e.g., Chen et al. (2023); Clinton et al. (2025); Dorner et al. (2023); Chakarov et al. (2024a;b)) consider the setting

with $N$ agents collaboratively estimating a mean $\mu = \frac{1}{N} \sum_{i=1}^{N} \mu_i$, where $\mu_i \in \mathbb{R}^n$ denotes the mean of agent $i$'s local data distribution. Given that $\mu_i$ is generally unknown to each agent $i$ in real-world applications, agent $i$ sends its local mean estimate $\theta_i$ (computed from its available local data samples) to a central server. The centralized server then aggregates mean estimates from all agents to compute a global mean estimate $\bar{\theta} = \frac{1}{N} \sum_{i=1}^{N} \theta_i$ and broadcasts the result to each agent. In this process, a strategic agent $i$ may report a false mean estimate $\theta_i'$ to the centralized server such that the returned global mean estimate $\bar{\theta}'$ minimizes its own mean squared error $\mathbb{E}[\|\bar{\theta}' - \mu_i\|^2]$. To mitigate such untruthful reporting, some incentive-based mechanisms have been proposed in the existing literature (e.g., Chen et al. (2023); Dorner et al. (2023); Chakarov et al. (2024a;b); Clinton et al. (2025)). Clearly, these mechanisms are typically one-shot, where agents select their strategies only once and the centralized server distributes payments a single time based on these reports.

However, in a fully decentralized setting, there is no centralized server/aggregator to collect local mean estimates from all agents and compute the global mean estimate. Instead, each agent $i$ can only exchange its local mean estimate with its neighbors. As a result, multiple rounds of communication and distributed averaging are required for all agents to reach consensus on the global mean estimate, i.e., $\lim_{T \to \infty} \theta_{i,T} = \bar{\theta}$ has to be achieved for all $i \in [N]$. This leads to a fundamental difference between consensus-based decentralized mean estimation and server-assisted mean estimation: in the consensus-based setting, information is exchanged locally over a communication network, and convergence to the global mean estimate is achieved through iterative updates rather than a single aggregation step. Consequently, strategic agents in consensus-based decentralized mean estimate algorithms can persistently adjust their actions over iterations, which poses challenges for both payment design and truthfulness analysis compared with the server-assisted scenario.

**Agents can benefit from gradient manipulation in consensus-based decentralized mean estimation.** We consider a setting with $N$ agents cooperatively estimating a global mean $\mu = \frac{1}{N} \sum_{i=1}^{N} \mu_i \in \mathbb{R}^n$. At each iteration $t$, agent $i$ obtains a data sample $x_{i,t}$, where each coordinate $x_{i,t}^p$, $p \in [n]$, is independently and identically drawn from the univariate normal distribution $\mathcal{N}(\mu_i^p, \sigma^2)$. Agent $i$ then computes its local mean estimate $\theta_{i,t}$ based on the sampled data $x_{i,t}$ and shares $\theta_{i,t}$ with its neighbors according to a given communication graph. The goal of a self-interested agent $i$ is to minimize its own mean squared error, i.e., $\mathbb{E}[\|\bar{\theta}_\infty - \mu_i\|^2]$, where $\bar{\theta}_\infty = \lim_{T \to \infty} \frac{1}{N} \sum_{i=1}^{N} \theta_{i,T}$ denotes the average of all agents' mean estimates when the number of iterations tends to infinity.

From the above discussion, we have that the local objective function of agent $i$ is $f_i(\theta) = \mathbb{E}[\|\theta - \mu_i\|^2]$, and its stochastic gradient estimate at iteration $t$ satisfies $g_i(\theta_{i,t}) = 2(\theta_{i,t} - x_{i,t})$. Following a conventional decentralized SGD in equation 10, each agent $i$ updates it mean estimate following

$$\theta_{i,t+1} = \sum_{j \in \mathcal{N}_i \cup \{i\}} w_{ij} \theta_{j,t} - 2\lambda_t (\theta_{i,t} - x_{i,t}), \quad \text{where } \lambda_t \text{ is the stepsize.} \tag{13}$$

We present the following Lemma 2 to quantitatively characterize the benefit that an agent $i$ can obtain from manipulating its gradient estimate in the consensus-based decentralized mean estimation algorithm in equation 13.

**Lemma 2.** *We consider $N$ agents cooperatively estimating a global mean vector $\mu = \frac{1}{N} \sum_{i=1}^{N} \mu_i$. If all agents truthfully participate in the consensus-based decentralized mean estimation algorithm in equation 13, then the mean squared error of any agent $i$ satisfies $\mathbb{E}[\|\bar{\theta}_\infty - \mu_i\|^2] = \mathcal{O}(\|\mu - \mu_i\|^2)$. However, if a strategic agent $i$ upscales its gradient estimate as $\alpha_i(g_i(\theta_{i,t})) = 2a_i(\theta_{i,t} - x_{i,t})$ with some constant $a_i > 1$ at every iteration $t$, then its mean squared error reduces to $\mathbb{E}[\|\bar{\theta}_\infty - \mu_i\|^2] = \mathcal{O}\left( \left( \frac{N}{a_i + N - 1} \right)^2 \|\mu - \mu_i\|^2 \right)$.*

*Proof.* 1) We first characterize the mean squared error of agent $i$ when all agents are truthful.

According to equation 13, we have $\bar{\theta}_{t+1} = \bar{\theta}_t - 2\lambda_t (\bar{\theta}_t - \bar{x}_t)$ with $\bar{\theta}_t \triangleq \frac{1}{N} \sum_{i=1}^{N} \theta_{i,t}$ and $\bar{x}_t \triangleq \frac{1}{N} \sum_{i=1}^{N} x_{i,t}$. By defining $\tilde{\theta}_t \triangleq \bar{\theta}_t - \mu$ and $\tilde{x}_t \triangleq \bar{x}_t - \mu$, we have

$$\tilde{\theta}_{t+1} = (1 - 2\lambda_t)\tilde{\theta}_t + 2\lambda_t \tilde{x}_t.$$

Since $\tilde{x}_t$ has zero mean and variance $\frac{\sigma^2}{N}$, and is independent of $\tilde{\theta}_t$, we have

$$\mathbb{E}[\|\tilde{\theta}_{t+1}\|^2] = (1 - 2\lambda_t)^2 \mathbb{E}[\|\tilde{\theta}_t\|^2] + 4\lambda_t^2 \mathbb{E}[\|\tilde{x}_t\|^2] = (1 - 2\lambda_t)^2 \mathbb{E}[\|\tilde{\theta}_t\|^2] + \frac{4\lambda_t^2 \sigma^2}{N}. \tag{14}$$

By setting the stepsize as $\lambda_t = \frac{\lambda_0}{(t+1)^v}$ with $0 < v < 1$, we have $(1 - 2\lambda_t)^2 \leq 1 - 2\lambda_t$ for all $t \geq 0$. Furthermore, by using Lemma 4 and the definition $\tilde{\theta}_t = \bar{\theta}_t - \mu$, we obtain

$$\mathbb{E}[\|\bar{\theta}_t - \mu\|^2] \leq \frac{c_1 \lambda_t \sigma^2}{N}, \tag{15}$$

where the constant $c_1$ is given by $c_1 = \frac{N}{\lambda_0 \sigma^2} \max\left\{ \mathbb{E}[\|\bar{\theta}_0 - \mu\|^2], \frac{2\lambda_0^2 \sigma^2}{N(2\lambda_0 - v)} \right\}$.

According to equation 15 and the relationship $\mathbb{E}[\bar{\theta}_t] = \mu$, we have

$$\mathbb{E}[\|\bar{\theta}_t - \mu_i\|^2] = \mathbb{E}[\|\bar{\theta}_t - \mu + (\mu - \mu_i)\|^2] \leq \frac{c_1 \lambda_t \sigma^2}{N} + \|\mu - \mu_i\|^2,$$

which implies

$$\lim_{T \to \infty} \mathbb{E}[\|\bar{\theta}_T - \mu_i\|^2] \leq \|\mu - \mu_i\|^2. \tag{16}$$

In addition, combining the relationship $(1 - \lambda_t)^2 \geq 1 - 2\lambda_t$ and equation 14, we have

$$\mathbb{E}[\|\tilde{\theta}_{t+1}\|^2] \geq (1 - 2\lambda_t)\mathbb{E}[\|\tilde{\theta}_t\|^2] + \frac{4\lambda_t^2 \sigma^2}{N}.$$

By using Lemma 5 and the definition $\tilde{\theta}_t = \bar{\theta}_t - \mu$, we obtain

$$\mathbb{E}[\|\bar{\theta}_t - \mu\|^2] \geq \frac{c_2 \lambda_t \sigma^2}{N}, \tag{17}$$

where the constant $c_2$ is given by $c_2 = \frac{N}{\lambda_0 \sigma^2} \min\left\{ \mathbb{E}[\|\bar{\theta}_0 - \mu\|^2], \frac{2\lambda_0 \sigma^2}{N} \right\}$.

According to equation 17 and the relationship $\mathbb{E}[\bar{\theta}_t] = \mu$, we have

$$\mathbb{E}[\|\bar{\theta}_t - \mu_i\|^2] \geq \frac{c_2 \lambda_t \sigma^2}{N} + \|\mu - \mu_i\|^2,$$

which implies

$$\lim_{T \to \infty} \mathbb{E}[\|\bar{\theta}_T - \mu_i\|^2] \geq \|\mu - \mu_i\|^2. \tag{18}$$

By combining equation 16 and equation 18, we have that when all agents truthfully participate in equation 13, the mean squared error of agent $i$ satisfies

$$\lim_{T \to \infty} \mathbb{E}[\|\bar{\theta}_T - \mu_i\|^2] = \mathbb{E}[\|\bar{\theta}_\infty - \mu_i\|^2] = \|\mu - \mu_i\|^2. \tag{19}$$

2) Next, we characterize the mean squared error of agent $i$ when it modifies its gradient estimate as $\alpha_i(g_i(\theta_{i,t})) = 2a_i(\theta_{i,t} - x_{i,t})$ with some constant $a_i > 1$ while all other agents act truthfully. By defining $\hat{\theta}_{i,t} \triangleq \theta_{i,t} - \bar{\theta}_t$, we have

$$\begin{aligned}
\bar{\theta}_{t+1} &= \bar{\theta}_t - 2\lambda_t \left( \bar{\theta}_t - \bar{x}_t \right) - \frac{2(a_i - 1)}{N} \lambda_t \left( \theta_{i,t} - x_{i,t} \right) \\
&= \bar{\theta}_t - 2\lambda_t \left( \bar{\theta}_t - \bar{x}_t \right) - \frac{2(a_i - 1)}{N} \lambda_t \left( \theta_{i,t} - \bar{\theta}_t + \bar{\theta}_t - x_{i,t} \right) \\
&= \bar{\theta}_t - 2\lambda_t \left( \left(1 + \frac{a_i - 1}{N}\right) \bar{\theta}_t - \left( \bar{x}_t + \frac{a_i - 1}{N} x_{i,t} \right) \right) - \frac{2(a_i - 1)}{N} \lambda_t \hat{\theta}_{i,t} \\
&= \bar{\theta}_t - \frac{2(N + a_i - 1)}{N} \lambda_t \left( \bar{\theta}_t - \left( \frac{a_i x_{i,t} + \sum_{j \neq i} x_{j,t}}{N + a_i - 1} \right) \right) - \frac{2(a_i - 1)}{N} \lambda_t \hat{\theta}_{i,t}.
\end{aligned} \tag{20}$$

For the sake of notational simplicity, we define $b \triangleq \frac{a_i + N - 1}{N}$ and $\hat{\mu} \triangleq \frac{a_i \mu_i + \sum_{j \neq i} \mu_j}{a_i + N - 1}$. Then, equation 20 can be rewritten as follows:

$$\bar{\theta}_{t+1} - \hat{\mu} = (1 - 2b\lambda_t) \left( \bar{\theta}_t - \hat{\mu} \right) + 2b\lambda_t \left( \frac{a_i x_{i,t} + \sum_{j \neq i} x_{j,t}}{a_i + N - 1} - \hat{\mu} \right) - \frac{2(a_i - 1)}{N} \lambda_t \hat{\theta}_{i,t}. \tag{21}$$

By taking the squared norm and expectations on both sides of equation 21, we obtain

$$\mathbb{E}\left[\left\|\bar{\theta}_{t+1}-\hat{\mu}\right\|^2\right] = (1+\tau_1\lambda_t)\,\mathbb{E}\left[\left\|(1-2b\lambda_t)\left(\bar{\theta}_t-\hat{\mu}\right)\right\|^2\right]$$
$$+ \left(1+\frac{1}{\tau_1\lambda_t}\right)\mathbb{E}\left[\left\|\frac{2(a_i-1)}{N}\lambda_t\hat{\theta}_{i,t}\right\|^2\right] + \frac{4b^2\lambda_t^2(a_i^2+N-1)\sigma^2}{(a_i+N-1)^2}, \tag{22}$$

for any $\tau_1 > 0$, where we have used the Young's inequality and the following relations:

$$\mathbb{E}\left[\frac{a_ix_{i,t}+\sum_{j\neq i}x_{j,t}}{a_i+N-1}-\hat{\mu}\right] = 0 \quad \text{and} \quad \mathbb{E}\left[\left\|\frac{a_ix_{i,t}+\sum_{j\neq i}x_{j,t}}{a_i+N-1}-\hat{\mu}\right\|^2\right] = \frac{(a_i^2+N-1)\sigma^2}{(a_i+N-1)^2}.$$

By letting $\tau_1 = b$, equation 22 can be rewritten as follows:

$$\mathbb{E}\left[\left\|\bar{\theta}_{t+1}-\hat{\mu}\right\|^2\right] \leq (1-b\lambda_t)\,\mathbb{E}\left[\left\|\bar{\theta}_t-\hat{\mu}\right\|^2\right]$$
$$+ \left(\lambda_0+\frac{1}{b}\right)\frac{4(a_i-1)^2}{N^2}\lambda_t\mathbb{E}\left[\left\|\hat{\theta}_{i,t}\right\|^2\right] + \frac{4b^2(a_i^2+N-1)\sigma^2}{(a_i+N-1)^2}\lambda_t^2. \tag{23}$$

According to $\alpha_i(g_i(\theta_{i,t})) = 2a_i(\theta_{i,t}-x_{i,t})$, and the definitions $\boldsymbol{\theta}_t \triangleq \text{col}(\theta_{1,t},\cdots,\theta_{N,t})$ and $\boldsymbol{x}_t \triangleq \text{col}(x_{1,t},\cdots,x_{N,t})$, the dynamics in equation 13 implies

$$\boldsymbol{\theta}_{t+1}-\mathbf{1}_N\otimes\bar{\theta}_{t+1} = (W\otimes I_n)\left(\boldsymbol{\theta}_t-\mathbf{1}_N\otimes\bar{\theta}_t\right) - 2\lambda_t\left(\boldsymbol{\theta}_t-\mathbf{1}_N\otimes\bar{\theta}_t-(\boldsymbol{x}_t-\mathbf{1}_N\otimes\bar{x}_t)\right)$$
$$+ 2\lambda_t\underbrace{\begin{bmatrix} \mathbf{1}_N\otimes\frac{a_i-1}{N}I_n \\ \vdots \\ \mathbf{1}_N\otimes(1-a_i)I_n+\mathbf{1}_N\otimes\frac{a_i-1}{N}I_n \\ \vdots \\ \mathbf{1}_N\otimes\frac{a_i-1}{N}I_n \end{bmatrix}}_{M_i}(\theta_{i,t}-x_{i,t}). \tag{24}$$

By taking the squared norm and expectations on both sides of equation 24, we obtain

$$\mathbb{E}\left[\left\|\boldsymbol{\theta}_{t+1}-\mathbf{1}_N\otimes\bar{\theta}_{t+1}\right\|^2\right] \leq (1+\tau_2)\,\mathbb{E}\left[\left\|(W\otimes I_n-2\lambda_tI_{Nn})\left(\boldsymbol{\theta}_t-\mathbf{1}_N\otimes\bar{\theta}_t\right)\right\|^2\right]$$
$$+ \left(1+\frac{1}{\tau_2}\right)\mathbb{E}\left[8\lambda_t^2\left\|\boldsymbol{x}_t-\mathbf{1}_N\otimes\bar{x}_t\right\|^2 + 8\lambda_t^2\|M_i\|^2\|\theta_{i,t}-x_{i,t}\|^2\right], \tag{25}$$

for any $\tau_2 > 0$, where we have used the Young's inequality.

The last term on the right hand side of equation 25 satisfies

$$\mathbb{E}\left[\|\theta_{i,t}-x_{i,t}\|^2\right] = \mathbb{E}\left[\|\theta_{i,t}-\bar{\theta}_t+\bar{\theta}_t-\hat{\mu}+\hat{\mu}-x_{i,t}\|^2\right]$$
$$\leq 3\mathbb{E}\left[\|\theta_{i,t}-\bar{\theta}_t\|^2\right] + 3\mathbb{E}\left[\|\bar{\theta}_t-\hat{\mu}\|^2\right] + 3\mathbb{E}\left[\|\hat{\mu}-x_{i,t}\|^2\right]. \tag{26}$$

Substituting equation 26 into equation 25 and setting $\tau_2 = 1-\rho$, we obtain

$$\mathbb{E}\left[\left\|\boldsymbol{\theta}_{t+1}-\mathbf{1}_N\otimes\bar{\theta}_{t+1}\right\|^2\right] \leq (1+(1-\rho))\left(\rho+2\lambda_t\right)^2\mathbb{E}\left[\left\|\boldsymbol{\theta}_t-\mathbf{1}_N\otimes\bar{\theta}_t\right\|^2\right]$$
$$+ 8\left(1+\frac{1}{1-\rho}\right)\lambda_t^2\mathbb{E}\left[\left\|\boldsymbol{x}_t-\mathbf{1}_N\otimes\bar{x}_t\right\|^2\right]$$
$$+ 24\left(1+\frac{1}{1-\rho}\right)\lambda_t^2\|M_i\|^2\mathbb{E}\left[\left\|\boldsymbol{\theta}_t-\mathbf{1}_N\otimes\bar{\theta}_t\right\|^2 + \|\bar{\theta}_t-\hat{\mu}\|^2 + \|\hat{\mu}-x_{i,t}\|^2\right]$$
$$\leq \left(\frac{(\rho+2\lambda_t)^2}{\rho}+\frac{24(2-\rho)\|M_i\|^2}{1-\rho}\lambda_t^2\right)\mathbb{E}\left[\left\|\boldsymbol{\theta}_t-\mathbf{1}_N\otimes\bar{\theta}_t\right\|^2\right]$$
$$+ \frac{24\|M_i\|^2(2-\rho)}{1-\rho}\lambda_t^2\mathbb{E}\left[\|\bar{\theta}_t-\hat{\mu}\|^2\right]$$
$$+ \frac{8(2-\rho)}{1-\rho}\lambda_t^2\left(\mathbb{E}\left[\|\boldsymbol{x}_t-\mathbf{1}_N\otimes\bar{x}_t\|^2\right] + 3\|M_i\|^2\mathbb{E}\left[\|x_{i,t}-\hat{\mu}\|^2\right]\right), \tag{27}$$

where we have used the relationship $(1 + (1 - \rho))(\rho + 2\lambda_t)^2 = \frac{(\rho+2\lambda_t)^2(1-(1-\rho)^2)}{\rho} \leq \frac{(\rho+2\lambda_t)^2}{\rho}$ in the second inequality.

Summing both sides of equation 23 and equation 27, we arrive at

$$\mathbb{E}\left[\left\|\bar{\theta}_{t+1} - \hat{\mu}\right\|^2\right] + \mathbb{E}\left[\left\|\boldsymbol{\theta}_{t+1} - \mathbf{1}_N \otimes \bar{\theta}_{t+1}\right\|^2\right]$$

$$\leq \left(1 - b\lambda_t + \frac{24\|M_i\|^2(2-\rho)}{1-\rho}\lambda_t^2\right)\mathbb{E}\left[\left\|\bar{\theta}_t - \hat{\mu}\right\|^2\right]$$

$$+ \left(\frac{4(\lambda_0 b + 1)(a_i - 1)^2}{bN^2}\lambda_t + \frac{(\rho+2\lambda_t)^2}{\rho} + \frac{24(2-\rho)\|M_i\|^2}{1-\rho}\lambda_t^2\right)\mathbb{E}\left[\left\|\boldsymbol{\theta}_t - \mathbf{1}_N \otimes \bar{\theta}_t\right\|^2\right]$$

$$+ \left(\frac{4b^2(a_i^2 + N - 1)\sigma^2}{(a_i + N - 1)^2}\right)\lambda_t^2 + \frac{8(2-\rho)}{1-\rho}\left(4Nd_x^2 + \frac{3\|M_i\|^2(N-1)\sum_{j\neq i}\|\mu_j - \mu_i\|^2}{(a_i + N - 1)^2}\right)\lambda_t^2,$$

where we have used $\mathbb{E}[\|x_{i,t}\|] \leq d_x$ for some constant $d_x > 0$, which follows from the boundedness of any data sample.

Since $\lambda_t = \frac{\lambda_0}{(t+1)^v}$ is a decaying sequence, we have

$$\mathbb{E}\left[\left\|\bar{\theta}_{t+1} - \hat{\mu}\right\|^2\right] + \mathbb{E}\left[\left\|\boldsymbol{\theta}_{t+1} - \mathbf{1}_N \otimes \bar{\theta}_{t+1}\right\|^2\right]$$
$$\leq \left(1 - \frac{b\lambda_t}{2}\right)\left(\mathbb{E}\left[\left\|\bar{\theta}_t - \hat{\mu}\right\|^2\right] + \mathbb{E}\left[\left\|\boldsymbol{\theta}_t - \mathbf{1}_N \otimes \bar{\theta}_t\right\|^2\right]\right) + c_3\lambda_t^2, \tag{28}$$

with $c_3 = \left(\frac{4b^2(a_i^2 + N - 1)\sigma^2}{(a_i + N - 1)^2}\right) + \frac{8(2-\rho)}{1-\rho}\left(4Nd_x^2 + \frac{3\|M_i\|^2(N-1)\sum_{j\neq i}\|\mu_j - \mu_i\|^2}{(a_i + N - 1)^2}\right)$.

Further combining Lemma 4 with equation 28, we arrive at

$$\mathbb{E}\left[\left\|\bar{\theta}_{t+1} - \hat{\mu}\right\|^2\right] + \mathbb{E}\left[\left\|\boldsymbol{\theta}_{t+1} - \mathbf{1}_N \otimes \bar{\theta}_{t+1}\right\|^2\right] \leq c_4\lambda_t, \tag{29}$$

where the constant $c_4$ is given by $c_4 = \max\{\mathbb{E}[\|\bar{\theta}_0 - \hat{\mu}\|^2] + \mathbb{E}[\|\boldsymbol{\theta}_0 - \mathbf{1}_N \otimes \bar{\theta}_0\|^2], \frac{2c_3\lambda_0^2}{b\lambda_0 - 2v}\}$.

By using $\mathbb{E}[\|a + b\|^2] \leq (\sqrt{\mathbb{E}[\|a\|^2]} + \sqrt{\mathbb{E}[\|b\|^2]})^2$ for any random variables $a$ and $b$, we have

$$\mathbb{E}\left[\|\bar{\theta}_{t+1} - \mu_i\|^2\right] \leq \left(\sqrt{\mathbb{E}\left[\|\bar{\theta}_{t+1} - \hat{\mu}\|^2\right]} + \sqrt{\mathbb{E}\left[\|\hat{\mu} - \mu_i\|^2\right]}\right)^2$$
$$\leq \left(\sqrt{c_4\lambda_t} + \left(\frac{N}{a_i + N - 1}\right)\|\mu - \mu_i\|\right)^2,$$

which implies that when agent $i$ upscales its gradient estimate as $a_i g_i(\theta_{i,t})$ with some $a_i > 1$ while all other agents act truthfully, the mean squared error of agent $i$ satisfies

$$\lim_{T\to\infty}\mathbb{E}[\|\bar{\theta}_T - \mu_i\|^2] = \mathbb{E}[\|\bar{\theta}_\infty - \mu_i\|^2] = \left(\frac{N}{a_i + N - 1}\right)^2\|\mu - \mu_i\|^2. \tag{30}$$

According to equation 19 and equation 30, we have that the mean squared error of agent $i$ is reduced from $\mathcal{O}(\|\mu - \mu_i\|^2)$ to $\mathcal{O}\left(\left(\frac{N}{a_i+N-1}\right)^2\|\mu - \mu_i\|^2\right)$ when it scales its gradient estimates by a factor $a_i > 1$. This demonstrates that a self-interested agent can benefit from its strategic gradient manipulation in consensus-based decentralized mean estimation algorithms. $\square$

In addition, it is worth noting that the consensus-based decentralized mean estimation problem is a special case of our formulated decentralized optimization/learning problem in equation 1. Therefore, our fully decentralized payment mechanism in Mechanism 1 is also applicable for incentivizing agents' truthful behavior in consensus-based decentralized mean estimation. Moreover, all theoretical results presented in Lemma 1, Theorem 1, and Theorem 2 apply directly to this special case.

### B.3 ADDITIONAL LITERATURE COMPARISON

**Decentralized learning and optimization.** In the past decade, various approaches have been proposed for decentralized learning/optimization (Shi et al., 2015; Yang et al., 2019; Verbraeken et al., 2020). Among those approaches, decentralized SGD receives the most attention due to its simplicity and effectiveness in neural network training (Lian et al., 2017; Assran et al., 2019; Amiri & Gündüz, 2020; Lin et al., 2021; Bars et al., 2024). However, decentralized SGD often suffers from error or bias terms caused by data heterogeneity among agents (Yuan et al., 2016). To address this issue, existing approaches employ decaying stepsizes in decentralized SGD to ensure accurate convergence, achieving a rate of $\mathcal{O}(1/\sqrt{T})$ for nonconvex objective functions (Assran et al., 2019; Koloskova et al., 2020b; Lin et al., 2021; Castiglia et al., 2021; Huang et al., 2025), and corresponding results for convex and strongly convex objectives are provided in Koloskova et al. (2020b); Pu et al. (2021); Huang et al. (2022); Bars et al. (2024). Nevertheless, all existing approaches rely on an implicit assumption that all agents behave truthfully. In practice, however, strategic agents may manipulate gradient updates and share false information for personal gains, which can significantly degrade leaning accuracy of the global model (see our experimental results in Fig. 2 and the motivating examples in Section B.1 and Section B.2 for details).

**Game theory.** Different from prior work on coalition games in collaborative learning (Zhou et al., 2022; Meng & Li, 2023; Wang et al., 2024), we model strategic interactions among agents in decentralized learning/optimization as a non-cooperative game. In addition, Mazumdar et al. (2020) analyzes gradient-based dynamics in continuous games, which is related to our study of gradient-based dynamics in decentralized learning/optimization. However, it does not consider mechanism design in games or the strategic manipulation of agents during algorithm execution.

**Robustness in decentralized learning and optimization.** Many works have studied the robustness of decentralized learning/optimization algorithms under corrupted gradients or noisy models, considering sources such as non-malicious or differential-privacy noise (Wang & Başar, 2022; Wang & Nedić, 2024), adversarial or Byzantine attacks (Yang et al., 2020; Turan et al., 2022; Han et al., 2025), quantization (Wang et al., 2022), and inexact gradient estimates (Hallak & Levy, 2024). These studies typically treat corruption as either exogenous noise or malicious perturbation, and focus on designing algorithms that remain effective under such disturbances. In contrast, our paper addresses gradient/data corruption that arises endogenously from strategic manipulation by selfish agents. We model such manipulation as a natural outcome of individual incentives and analyze agents' behaviors through a game-theoretic framework. Our objective is not merely to enhance algorithmic resilience, but to develop mechanism-level solutions that align incentive with truthful gradient updates, thereby preventing manipulative behaviors from occurring in the first place.

**Personalized learning.** Personalized learning methods in Zhalechian et al. (2022); Even et al. (2022); Tian et al. (2025) aim to train a global model while simultaneously providing per-agent personalization to maximize individual learning accuracy. These approaches primarily address data heterogeneity from the perspective of enhancing each agent's performance. In contrast, we address the problem from a mechanism design perspective, where the main objective is to incentivize agents to truthfully participate in decentralized learning and optimization, thereby safeguarding the performance of the learned global model.

## C NOTATIONS, DEFINITIONS, AND AUXILIARY LEMMAS

### C.1 ADDITIONAL NOTATIONS AND DEFINITIONS

Throughout this paper, we use an overbar to denote the average of all agents, e.g., $\bar{\theta}_t = \frac{1}{N}\sum_{i=1}^{N}\theta_{i,t}$, and use bold font with iteration subscripts to denote the stacked vector of all $N$ agents, e.g., $\boldsymbol{\theta}_t = \mathrm{col}(\theta_{1,t}, \cdots, \theta_{N,t})$. Moreover, we use $\theta_{i,t+1}$ to denote the model parameter of agent $i$ updated by Algorithm 1 with gradient $m_{i,t} = g_i(\theta_{i,t})$ at iteration $t$ and $\theta'_{i,t+1}$ to denote the model parameter of agent $i$ updated by Algorithm 1 with a (manipulated) gradient $m_{i,t} = \alpha_{i,t}(g_i(\theta_{i,t}))$ under action $\alpha_{i,t}$ at iteration $t$. We let $P_{i,t}(h_{i,t}, h_{-i,t})$ denote the payment of agent $i$ when both itself and all other agents act truthfully at iteration $t$, and $P_{i,t}(\alpha_{i,t}, h_{-i,t})$ denote the payment of agent $i$ when it acts untruthfully (by taking action $\alpha_{i,t}$) while all other agents act truthfully at iteration $t$, where $h_{-i,t}$ represents the truthful actions of all agents except agent $i$ at iteration $t$.

Next, we introduce several relevant definitions used in our paper.

**Definition 3** (Lipschitz continuity). *A function $f(\theta) : \mathbb{R}^n \mapsto \mathbb{R}$ is L-Lipschitz continuous with some constant $L > 0$ if for any $\theta_1, \theta_2 \in \mathbb{R}^n$, the following inequality always holds:*

$$\|f(\theta_1) - f(\theta_2)\| \le L\|\theta_1 - \theta_2\|.$$

**Definition 4** (Smoothness). *A differentiable function $f(\theta) : \mathbb{R}^n \mapsto \mathbb{R}$ is said to be H-smooth if its gradient is H-Lipschitz continuous with some constant $H > 0$, i.e., for any $\theta_1, \theta_2 \in \mathbb{R}^n$, we have*

$$\|\nabla f(\theta_1) - \nabla f(\theta_2)\| \le H\|\theta_1 - \theta_2\|.$$

**Definition 5** ($\varrho$-solution (Lian et al., 2017)). *For any $i \in [N]$ and some integer $T \ge 0$, if $\mathbb{E}\left[\|\theta_{i,T} - \theta^*\|^2\right] \le \varrho$ holds when $F(\theta)$ is strongly convex, or $\frac{1}{T+1}\sum_{t=0}^T \mathbb{E}\left[F(\theta_{i,t}) - F(\theta^*)\right] \le \varrho$ holds when $F(\theta)$ is general convex, then we say that the sequence $\{\theta_{i,t}\}_{t=0}^T$ can reach a $\varrho$-solution to the problem in equation 1.*

In our game-theoretic framework, the notion of $\varepsilon$-Nash equilibrium presented in Huang et al. (2007) can be formalized as follows:

**Definition 6** ($\varepsilon$-Nash equilibrium). *We let $(\boldsymbol{\alpha}_1, \cdots, \boldsymbol{\alpha}_N) \in \mathcal{P}(\mathcal{A}_1^T) \times \cdots \times \mathcal{P}(\mathcal{A}_N^T)$ be the action trajectory profile of N agents, where $\mathcal{P}(\mathcal{A}_i^T)$ denotes the set of all probability measures over agent $i$'s action space $\mathcal{A}_i^T$. We say $\boldsymbol{\alpha}^* = (\boldsymbol{\alpha}_1^*, \cdots, \boldsymbol{\alpha}_N^*)$ is an $\varepsilon$-Nash equilibrium w.r.t. the net utility $U_{i,0 \to T}^{\mathcal{M}_p}$ in equation 3 if for any $i \in [N]$ and $\boldsymbol{\alpha}_i \in \mathcal{P}(\mathcal{A}_i^T)$, the following inequality holds:*

$$\mathbb{E}\left[U_{i,0 \to T}^{\mathcal{M}_p}(\boldsymbol{\alpha}_1^*, \cdots, \boldsymbol{\alpha}_i^*, \cdots, \boldsymbol{\alpha}_N^*)\right] \ge \mathbb{E}\left[U_{i,0 \to T}^{\mathcal{M}_p}(\boldsymbol{\alpha}_1^*, \cdots, \boldsymbol{\alpha}_i, \cdots, \boldsymbol{\alpha}_N^*)\right] - \varepsilon,$$

*where the expectation is taken over the randomness in agents' data distributions and action trajectories.*

Definition 6 implies that, in an $\varepsilon$-Nash equilibrium, no agent can improve its net utility by more than $\varepsilon$ through unilateral deviation from its equilibrium action trajectory.

## C.2 AUXILIARY LEMMAS

In this subsection, we introduce some known results from the existing literature and some auxiliary lemmas that will be used in our subsequent convergence and truthfulness analysis.

**Lemma 3** (Lemma 7 in Chen & Wang (2025)). *The relation $c\gamma^t \le \frac{1}{t^2}$ always holds for all $t > 0$ and $\gamma \in (0,1)$, where the constant $c$ is given by $c = \frac{(\ln(\gamma)e)^2}{4}$.*

**Lemma 4** (Lemma 5 in Chen et al. (2025)). *Denoting $\eta_t$ as a nonnegative sequence, if there exist sequences $\beta_{1,t} = \frac{\beta_1}{(t+1)^{r_1}}$ and $\beta_{2,t} = \frac{\beta_2}{(t+1)^{r_2}}$ with some $1 > r_1 > \frac{1}{2}$, $r_2 > r_1$, $\beta_1 > r_2 - r_1$, and $\beta_2 > 0$ such that $\eta_{t+1} \le (1 - \beta_{1,t})\eta_t + \beta_{2,t}$ holds, then we always have $\eta_t \le \frac{\beta_1}{\beta_2} \max\{\eta_0, \frac{\beta_2}{\beta_1 - (r_2 - r_1)}\} \frac{\beta_{2,t}}{\beta_{1,t}}$.*

**Lemma 5.** *Denoting $\eta_t$ as a nonnegative sequence, if there exist sequences $\beta_{1,t} = \frac{\beta_1}{(t+1)^{r_1}}$ and $\beta_{2,t} = \frac{\beta_2}{(t+1)^{r_2}}$ with some $1 > r_1 > 0$, $r_2 > r_1$, $\beta_1 > 0$, and $\beta_2 > 0$ such that $\eta_{t+1} \ge (1 - \beta_{1,t})\eta_t + \beta_{2,t}$ holds, then we always have $\eta_t \ge \frac{\beta_1}{\beta_2} \min\{\eta_0, \frac{\beta_2}{\beta_1}\} \frac{\beta_{2,t}}{\beta_{1,t}}$.*

*Proof.* We prove Lemma 5 using mathematical induction.

We define $c_0 = \min\{\eta_0, \frac{\beta_2}{\beta_1}\}$, which implies $\eta_0 \ge c_0$ at initialization. Assuming $\eta_t \ge \frac{c_0}{(t+1)^{r_2 - r_1}}$ at the $t$th iteration, we proceed to prove $\eta_{t+1} \ge \frac{c_0}{(t+2)^{r_2 - r_1}}$ at the $(t+1)$th iteration.

By using the relationship $\eta_{t+1} \ge (1 - \beta_{1,t})\eta_t + \beta_{2,t}$, we have

$$
\begin{aligned}
\eta_{t+1} &\ge \frac{c_0}{(t+1)^{r_2 - r_1}} - \frac{c_0 \beta_1}{(t+1)^{r_2}} + \frac{\beta_2}{(t+1)^{r_2}} \\
&\ge \frac{c_0}{(t+2)^{r_2 - r_1}} + \left(\frac{c_0}{(t+1)^{r_2 - r_1}} - \frac{c_0}{(t+2)^{r_2 - r_1}} - \frac{c_0 \beta_1 - \beta_2}{(t+1)^{r_2}}\right).
\end{aligned}
\tag{31}
$$

Using the mean value theorem, we have $\frac{c_0}{(t+1)^{r_2-r_1}} - \frac{c_0}{(t+2)^{r_2-r_1}} = \frac{c_0(r_2-r_1)}{\varsigma^{r_2-r_1+1}} > \frac{c_0(r_2-r_1)}{(t+2)^{r_2-r_1+1}}$ with some $\varsigma \in (t+1, t+2)$, which, combined with $c_0\beta_1 - \beta_2 \leq 0$, leads to $\frac{c_0(r_2-r_1)}{(t+2)^{r_2-r_1+1}} \geq \frac{c_0\beta_1-\beta_2}{(t+1)^{r_2}}$. Hence, the inequality $\frac{c_0}{(t+1)^{r_2-r_1}} - \frac{c_0}{(t+2)^{r_2-r_1}} \geq \frac{c_0\beta_1-\beta_2}{(t+1)^{r_2}}$ holds for any $t \geq 0$. Further using equation 31, we arrive at $\eta_{t+1} \geq \frac{c_0}{(t+2)^{r_2-r_1}}$, which completes the proof of Lemma 5. $\qquad\square$

**Lemma 6.** *For a nonnegative sequence $\eta_t = \frac{\eta_0}{(t+1)^r}$ with $\eta_0 > 0$ and $r \in (0, 2)$, the inequality $\sum_{k=0}^{t-1} \eta_k^2 \gamma^{2(t-1-k)} \leq c\eta_t^2$ always holds for any $t \geq 1$ and $\gamma \in (0, 1)$, where the constant $c$ is given by $c = \frac{4^{2(r+1)}}{(1-\gamma)(\ln(\sqrt{\gamma})e)^4}$.*

*Proof.* By defining $c_0 = 4^{-1}(\ln(\sqrt{\gamma})e)^2$ and using Lemma 3, we have

$$\eta_k \sqrt{\gamma}^{t-1-k} \leq \eta_k \frac{1}{c_0((t-1)-k)^2} = \frac{\eta_0}{c_0(k+1)^r((t-1)-k)^2}. \tag{32}$$

For some real numbers $a, b, c, d > 0$ satisfying $\frac{c}{d} < \frac{d}{b}$, the inequality $\frac{d}{b} < \frac{c+d}{a+b} < \frac{c}{a}$ always holds. Therefore, for any $t > 0$ and $k \in [0, t-1)$, by setting $a = k$, $b = (t-1) - k$, $c = k$, and $d = 1$, we have

$$\frac{1}{((t-1)-k)^2} < \left(\frac{k+1}{t-1}\right)^2.$$

Combining equation 32 and the relation $\frac{1}{((t-1)-k)^2} < (\frac{k+1}{t-1})^r$ for any $k \in [0, t-1)$ and $r \in (0, 2)$, we obtain

$$\eta_k \sqrt{\gamma}^{t-1-k} < \frac{\eta_0}{c_0(k+1)^r} \left(\frac{k+1}{t-1}\right)^r \leq \frac{4^r \eta_0}{c_0(t+1)^r} = \frac{4^r \eta_t}{c_0}, \tag{33}$$

where we have used the relationship $\frac{1}{t-1} \leq \frac{4}{t+1}$ for any $t > 1$ in the second inequality. Equation 33 further implies $\eta_k^2 \gamma^{t-1-k} \leq 4^{2r} c_0^{-2} \eta_t^2$ for any $t > 1$ and $k \in [0, t-1)$.

Given a constant $c > 1$, the inequality $\eta_0^2 \leq c\eta_0^2$ holds for $t = 1$ and the inequality $\eta_{t-1}^2 \leq c\eta_{t-1}^2$ holds for any $t > 1$ and $k = t - 1$, which, combined with equation 33, leads to

$$\sum_{k=0}^{t-1} \eta_k^2 \gamma^{2(t-1-k)} \leq \sum_{k=0}^{t-1} \gamma^{t-1-k} \frac{4^{2r}\eta_t^2}{c_0^2} \leq \frac{4^{2(r+1)}\eta_t^2}{(1-\gamma)(\ln(\sqrt{\gamma})e)^4} = c\eta_t^2,$$

which completes the proof of Lemma 6. $\qquad\square$

We present the following Lemma 7 to clarify the connection between $\varepsilon$-incentive compatibility and $\varepsilon$-Nash equilibrium.

**Lemma 7.** *If a decentralized learning protocol $\mathcal{M}_p$ is $\varepsilon$-incentive compatible, then the truthful action trajectory profile of all agents $\boldsymbol{h} = (\boldsymbol{h}_1, \cdots, \boldsymbol{h}_N)$ constitutes an $\varepsilon$-Nash equilibrium.*

*Proof.* According to the definition of $\varepsilon$-incentive compatibility in Definition 2, the following inequality holds for any agent $i \in [N]$ and any arbitrary action trajectory $\boldsymbol{\alpha}_i$ of agent $i$:

$$\mathbb{E}[U_{i,0\to T}^{\mathcal{M}_p}(\boldsymbol{h}_i, \boldsymbol{h}_{-i})] \geq \mathbb{E}[U_{i,0\to T}^{\mathcal{M}_p}(\boldsymbol{\alpha}_i, \boldsymbol{h}_{-i})] - \varepsilon, \tag{34}$$

where $\boldsymbol{h}_i$ denotes the truthful action trajectory of agent $i$ and $\boldsymbol{h}_{-i} = \{\boldsymbol{h}_1, \cdots, \boldsymbol{h}_{i-1}, \boldsymbol{h}_{i+1}, \cdots, \boldsymbol{h}_N\}$ denotes the truthful action trajectories of all agents except agent $i$.

By setting $\boldsymbol{h} = (\boldsymbol{h}_1, \cdots, \boldsymbol{h}_i, \cdots, \boldsymbol{h}_N) = \boldsymbol{\alpha}^*$, the preceding equation 34 can be rewritten as

$$\mathbb{E}\left[U_{i,0\to T}^{\mathcal{M}_p}(\boldsymbol{\alpha}_1^*, \cdots, \boldsymbol{\alpha}_i^*, \cdots, \boldsymbol{\alpha}_N^*)\right] \geq \mathbb{E}\left[U_{i,0\to T}^{\mathcal{M}_p}(\boldsymbol{\alpha}_1^*, \cdots, \boldsymbol{\alpha}_i, \cdots, \boldsymbol{\alpha}_N^*)\right] - \varepsilon, \tag{35}$$

which is exactly the condition of an $\varepsilon$-Nash equilibrium in Definition 6. Since $i$ is arbitrary, equation 35 holds for every agent $i \in [N]$, and hence, $\boldsymbol{h} = (\boldsymbol{h}_1, \ldots, \boldsymbol{h}_N)$ is an $\varepsilon$-Nash equilibrium. $\qquad\square$

# D RESULTS ON ALGORITHM 1 WITH OUR PAYMENT MECHANISM

## D.1 PROOF OF LEMMA 1

In this subsection, we prove that under our decentralized payment mechanism (Mechanism 1), the incentive for a strategic agent in Algorithm 1 to deviate from truthful behavior diminishes to zero. This result is summarized in Lemma 9, which corresponds to Lemma 1 in the main text. To prove Lemma 9 (and Lemma 1), we first present the following auxiliary lemma.

**Lemma 8.** *We denote $\theta_{i,T}$ and $\theta'_{i,T}$ as the model parameters generated by the decentralized SGD algorithm at iteration $T-1$ from two different initializations $\theta_{i,t}$ and $\theta'_{i,t}$, respectively. Under the conditions in Lemma 1, the following result holds for the decentralized SGD algorithm:*

$$\mathbb{E}[\|\theta_{i,T} - \theta'_{i,T}\|^2] \leq d_{t \to T} \left( N\mathbb{E}[\|\bar{\theta}_t - \bar{\theta}'_t\|^2] + \sum_{i=1}^{N} \mathbb{E}\left[\|\theta_{i,t} - \theta'_{i,t} - (\bar{\theta}_t - \bar{\theta}'_t)\|^2\right] \right), \quad (36)$$

*where the constant $d_{t \to T}$ is given by $d_{t \to T} = 2e^{\frac{10vH^2(2-\rho)\lambda_0^2}{(1-\rho)(2v-1)}\left(\frac{1}{t^{2v-1}} - \frac{1}{T^{2v-1}}\right)}$.*

*Proof.* According to the update rule of the decentralized SGD algorithm, we have

$$\theta_{i,t+1} = \sum_{j \in \mathcal{N}_i \cup \{i\}} w_{ij}\theta_{j,t} - \lambda_t g_i(\theta_{i,t}) \text{ and } \theta'_{i,t+1} = \sum_{j \in \mathcal{N}_i \cup \{i\}} w_{ij}\theta'_{j,t} - \lambda_t g_i(\theta'_{i,t}). \quad (37)$$

Denoting the sensitivity of the decentralized SGD algorithm as $\Xi_{i,t} = \theta_{i,t} - \theta'_{i,t}$, we can verify that its dynamics is governed by $\Xi_{i,t+1} = \sum_{j \in \mathcal{N}_i \cup \{i\}} w_{ij}\Xi_{j,t} - \lambda_t(g_i(\theta_{i,t}) - g_i(\theta'_{i,t}))$, which implies

$$\bar{\Xi}_{t+1} = \bar{\Xi}_t - \frac{1}{N}\sum_{i=1}^{N}\lambda_t(g_i(\theta_{i,t}) - g_i(\theta'_{i,t})). \quad (38)$$

By taking the squared norm and expectation on both sides of equation 38, we obtain

$$\mathbb{E}[\|\bar{\Xi}_{t+1}\|^2] = \mathbb{E}[\|\bar{\Xi}_t\|^2] + \mathbb{E}\left[\left\|\frac{1}{N}\sum_{i=1}^{N}\lambda_t(g_i(\theta_{i,t}) - g_i(\theta'_{i,t}))\right\|^2\right]$$

$$- \frac{2}{N}\sum_{i=1}^{N}\mathbb{E}\left[\left\langle\bar{\Xi}_t, \lambda_t\left(g_i(\theta_{i,t}) - g_i(\theta'_{i,t})\right)\right\rangle\right]. \quad (39)$$

By using the relationship $\mathbb{E}[g_i(\theta)] = \nabla f_i(\theta)$ from Assumption 1, the term $\mathbb{E}[\langle\bar{\Xi}_t, \lambda_t(g_i(\theta_{i,t}) - g_i(\theta'_{i,t}))\rangle]$ on the right hand side of equation 39 can be rewritten as

$$2\mathbb{E}\left[\left\langle\bar{\Xi}_t, \lambda_t\left(g_i(\theta_{i,t}) - g_i(\theta'_{i,t})\right)\right\rangle\right] = 2\lambda_t\mathbb{E}\left[\left\langle\bar{\Xi}_t, \nabla f_i(\theta_{i,t}) - \nabla f_i(\theta'_{i,t})\right\rangle\right]$$

$$= 2\lambda_t\mathbb{E}\left[\left\langle\Xi_{i,t}, \nabla f_i(\theta_{i,t}) - \nabla f_i(\theta'_{i,t})\right\rangle\right] - 2\lambda_t\mathbb{E}\left[\left\langle\Xi_{i,t} - \bar{\Xi}_t, \nabla f_i(\theta_{i,t}) - \nabla f_i(\theta'_{i,t})\right\rangle\right]. \quad (40)$$

Applying the Young's inequality to the last term on the right hand side of equation 40 yields

$$- 2\lambda_t\mathbb{E}\left[\left\langle\Xi_{i,t} - \bar{\Xi}_t, \nabla f_i(\theta_{i,t}) - \nabla f_i(\theta'_{i,t})\right\rangle\right]$$

$$\geq -\tau_1\mathbb{E}\left[\|\Xi_{i,t} - \bar{\Xi}_t\|^2\right] - \frac{\lambda_t^2}{\tau_1}\mathbb{E}\left[\|\nabla f_i(\theta_{i,t}) - \nabla f_i(\theta'_{i,t})\|^2\right], \quad (41)$$

for any $\tau_1 > 0$.

The convexity of $f_i(\theta)$ implies that the first term on the right hand side of equation 40 is non-negative, which, combined with equation 41, leads to the following results for the last term on the right hand side of equation 39

$$- \frac{2}{N}\sum_{i=1}^{N}\mathbb{E}\left[\left\langle\bar{\Xi}_t, \lambda_t\left(g_i(\theta_{i,t}) - g_i(\theta'_{i,t})\right)\right\rangle\right]$$

$$\leq \frac{\tau_1}{N}\sum_{i=1}^{N}\mathbb{E}[\|\Xi_{i,t} - \bar{\Xi}_t\|^2] + \frac{1}{\tau_1 N}\sum_{i=1}^{N}\lambda_t^2\mathbb{E}[\|\nabla f_i(\theta_{i,t}) - \nabla f_i(\theta'_{i,t})\|^2] \quad (42)$$

$$\leq \frac{\tau_1}{N}\sum_{i=1}^{N}\mathbb{E}[\|\Xi_{i,t} - \bar{\Xi}_t\|^2] + \frac{2H^2\lambda_t^2}{\tau_1}\mathbb{E}[\|\bar{\Xi}_t\|^2] + \frac{2H^2\lambda_t^2}{\tau_1 N}\sum_{i=1}^{N}\mathbb{E}[\|\Xi_{i,t} - \bar{\Xi}_t\|^2],$$

where in the derivation we have used the $H$-Lipschitz continuity of $g_i(\theta)$.

By using Assumption 1, we obtain the following result for the second term on the right hand side of equation 39

$$\mathbb{E}\left[\left\|\frac{1}{N}\sum_{i=1}^{N}\lambda_t(g_i(\theta_{i,t})-g_i(\theta'_{i,t}))\right\|^2\right] \leq \frac{\lambda_t^2}{N}\sum_{i=1}^{N}\mathbb{E}\left[\left\|g_i(\theta_{i,t})-g_i(\theta'_{i,t})\right\|^2\right]$$

$$\leq \frac{H^2\lambda_t^2}{N}\sum_{i=1}^{N}\mathbb{E}[\|\Xi_{i,t}\|^2] \leq 2H^2\lambda_t^2\mathbb{E}[\|\bar{\Xi}_t\|^2] + \frac{2H^2\lambda_t^2}{N}\sum_{i=1}^{N}\mathbb{E}[\|\Xi_{i,t}-\bar{\Xi}_t\|^2]. \tag{43}$$

Substituting equation 42 and equation 43 into equation 39, we arrive at

$$\mathbb{E}[\|\bar{\Xi}_{t+1}\|^2] \leq \left(1 + 2H^2\lambda_t^2 + \frac{2H^2\lambda_t^2}{\tau_1}\right)\mathbb{E}[\|\bar{\Xi}_t\|^2]$$

$$+ \left(\frac{\tau_1}{N} + \frac{2H^2\lambda_t^2}{N} + \frac{2H^2\lambda_t^2}{\tau_1 N}\right)\sum_{i=1}^{N}\mathbb{E}[\|\Xi_{i,t}-\bar{\Xi}_t\|^2]. \tag{44}$$

We proceed to establish an upper bound on $\sum_{i=1}^{N}\mathbb{E}[\|\Xi_{i,t+1}-\bar{\Xi}_{t+1}\|^2]$. According to the dynamics $\Xi_{i,t+1} = \sum_{j\in\mathcal{N}_i\cup\{i\}}w_{ij}\Xi_{j,t} - \lambda_t(g_i(\theta_{i,t})-g_i(\theta'_{i,t}))$ and equation 38, we have

$$\Xi_{i,t+1} - \bar{\Xi}_{t+1} \leq \sum_{j\in\mathcal{N}_i\cup\{i\}}w_{ij}(\Xi_{j,t}-\bar{\Xi}_t) - \lambda_t(g_i(\theta_{i,t})-g_i(\theta'_{i,t}))$$

$$- \frac{1}{N}\sum_{i=1}^{N}\lambda_t(g_i(\theta_{i,t})-g_i(\theta'_{i,t})). \tag{45}$$

By taking the squared norm and expectation on both sides of equation 38 and using Assumption 2, we obtain the following result:

$$\sum_{i=1}^{N}\mathbb{E}[\|\Xi_{i,t+1}-\bar{\Xi}_{t+1}\|^2] \leq (1+(1-\rho))\rho^2\sum_{i=1}^{N}\mathbb{E}[\|\Xi_{i,t}-\bar{\Xi}_t\|^2]$$

$$+ \left(1+\frac{1}{1-\rho}\right)\lambda_t^2\sum_{i=1}^{N}\mathbb{E}\left[\left\|(g_i(\theta_{i,t})-g_i(\theta'_{i,t})) - \frac{1}{N}\sum_{i=1}^{N}(g_i(\theta_{i,t})-g_i(\theta'_{i,t}))\right\|^2\right] \tag{46}$$

$$\leq \left(\rho + \frac{8(2-\rho)H^2\lambda_t^2}{1-\rho}\right)\sum_{i=1}^{N}\mathbb{E}[\|\Xi_{i,t}-\bar{\Xi}_t\|^2] + \frac{8N(2-\rho)H^2\lambda_t^2}{1-\rho}\mathbb{E}[\|\bar{\Xi}_t\|^2].$$

Multiplying both sides of equation 44 by $N$ and adding the resulting expression to both sides of equation 46 yield

$$N\mathbb{E}[\|\bar{\Xi}_{t+1}\|^2] + \sum_{i=1}^{N}\mathbb{E}[\|\Xi_{i,t+1}-\bar{\Xi}_{t+1}\|^2]$$

$$\leq \left(1 + 2H^2\lambda_t^2 + \frac{2H^2\lambda_t^2}{\tau_1} + \frac{8(2-\rho)H^2\lambda_t^2}{1-\rho}\right)N\mathbb{E}[\|\bar{\Xi}_t\|^2]$$

$$+ \left(\rho + \frac{8(2-\rho)H^2\lambda_t^2}{1-\rho} + \tau_1 + 2H^2\lambda_t^2 + \frac{2H^2\lambda_t^2}{\tau_1}\right)\sum_{i=1}^{N}\mathbb{E}[\|\Xi_{i,t}-\bar{\Xi}_t\|^2].$$

By letting $\tau_1 = 1-\rho$, the preceding inequality can be rewritten as

$$N\mathbb{E}[\|\bar{\Xi}_{t+1}\|^2] + \sum_{i=1}^{N}\mathbb{E}[\|\Xi_{i,t+1}-\bar{\Xi}_{t+1}\|^2] \leq (1+d_0\lambda_t^2)\left(N\mathbb{E}[\|\bar{\Xi}_t\|^2] + \sum_{i=1}^{N}\mathbb{E}[\|\Xi_{i,t}-\bar{\Xi}_t\|^2]\right), \tag{47}$$

where the constant $d_0$ is given by $d_0 = 2H^2 + \frac{2H^2+8(2-\rho)H^2}{1-\rho}$.

By iterating equation 47 from $t$ to $T$, we obtain

$$N\mathbb{E}[\|\bar{\Xi}_T\|^2] + \sum_{i=1}^N \mathbb{E}[\|\Xi_{i,T} - \bar{\Xi}_T\|^2]$$

$$\leq \prod_{k=t}^{T-1} (1 + d_0\lambda_k^2) \left( N\mathbb{E}[\|\bar{\Xi}_k\|^2] + \sum_{i=1}^N \mathbb{E}[\|\Xi_{i,k} - \bar{\Xi}_k\|^2] \right). \tag{48}$$

Since $\ln(1 + x) \leq x$ holds for all $x > 0$, we always have $\prod_{k=t}^{T-1}(1 + d_0\lambda_k^2) \leq e^{d_0\lambda_0^2 \sum_{k=t}^{T-1} \frac{1}{(k+1)^{2v}}}$. Further using the relationship $\sum_{k=t}^{T-1} \frac{1}{(k+1)^{2v}} \leq \int_t^T \frac{1}{x^{2v}} dx \leq \frac{1}{2v-1}(\frac{1}{t^{2v-1}} - \frac{1}{T^{2v-1}})$ and the definition $\Xi_{i,k} = \theta_{i,k} - \theta'_{i,k}$, we arrive at

$$N\mathbb{E}[\|\bar{\Xi}_T\|^2] + \sum_{i=1}^N \mathbb{E}[\|\Xi_{i,T} - \bar{\Xi}_T\|^2]$$

$$\leq e^{\frac{d_0\lambda_0^2}{2v-1}\left(\frac{1}{t^{2v-1}} - \frac{1}{T^{2v-1}}\right)} \left( N\mathbb{E}[\|\bar{\theta}_0 - \bar{\theta}'_0\|^2] + \sum_{i=1}^N \mathbb{E}\left[\|\theta_{i,0} - \theta'_{i,0} - (\bar{\theta}_0 - \bar{\theta}'_0)\|^2\right] \right), \tag{49}$$

with $d_0 = 2H^2 + \frac{2H^2 + 8(2-\rho)H^2}{1-\rho}$.

Applying the inequality $\|\Xi_{i,T}\|^2 \leq 2\|\bar{\Xi}_T\|^2 + 2\|\Xi_{i,T} - \bar{\Xi}_T\|^2$ for any $T \geq 0$, we arrive at

$$\mathbb{E}[\|\theta_{i,T} - \theta'_{i,T}\|^2] \leq d_{t \to T} \left( N\mathbb{E}[\|\bar{\theta}_t - \bar{\theta}'_t\|^2] + \sum_{i=1}^N \mathbb{E}\left[\|\theta_{i,t} - \theta'_{i,t} - (\bar{\theta}_t - \bar{\theta}'_t)\|^2\right] \right), \tag{50}$$

with $d_{t \to T} = 2e^{\frac{10vH^2(2-\rho)\lambda_0^2}{(1-\rho)(2v-1)}\left(\frac{1}{t^{2v-1}} - \frac{1}{T^{2v-1}}\right)}$, which completes the proof of Lemma 8. $\qquad\square$

For notational simplicity, we denote the cardinality of $\mathcal{N}_i$ as $\deg(i)$ in the sequel.

**Lemma 9.** *Under Assumption 1 and Assumption 2, for any given $\delta > 0$, $i \in [N]$, and $t \geq 0$, if we set $\lambda_t = \frac{\lambda_0}{(t+1)^v}$, $C_t = \frac{C_0\kappa_t^2}{\delta^2(t+1)^{-2v}}$ with $\kappa_t = \frac{1}{(t+1)^r}$, $v \in (\frac{1}{2}, \frac{2}{3})$, $r \in (1-v, v)$, and $C_0$ given in equation 72, and all neighbors of agent $i$ are truthful, then the optimal action for agent $i$ in Algorithm 1 is $\kappa_t\delta$-truthful. That is, for any $i \in [N]$ and $t \geq 0$, we have*

$$\mathbb{E}[\|\alpha_{i,t}(g_i(\theta_{i,t})) - g_i(\theta_{i,t})\|] \leq \kappa_t\delta. \tag{51}$$

*Furthermore, as the number of iterations tends to infinity, the optimal action for agent $i$ is fully truthful, i.e., $\lim_{t\to\infty} \mathbb{E}[\|\alpha_{i,t}(g_i(\theta_{i,t})) - g_i(\theta_{i,t})\|] = 0$, meaning that an agent will have zero incentive to deviate from truthful behaviors.*

*Proof.* To prove equation 51, we first quantify the difference in payments for agent $i$ between an action $\alpha_{i,t} \in \mathcal{A}_i$ and the truthful action $h_{i,t}$ at iteration $t$. According to our payment mechanism, we have $P_{i,t} = \sum_{j\in\mathcal{N}_i} P_{i,t}^j$, which implies

$$\mathbb{E}\left[P_{i,t}(h_{i,t}, h_{-i,t}) - P_{i,t}(\alpha_{i,t}, h_{-i,t})\right]$$

$$= C_t\mathbb{E}\left[\sum_{j\in\mathcal{N}_i} \left(\|\theta_{i,t+1} - 2\theta_{i,t} + \theta_{i,t-1}\|^2 - \|\theta_{j,t+1} - 2\theta_{j,t} + \theta_{i,t-1}\|^2\right)\right]$$

$$- C_t\mathbb{E}\left[\sum_{j\in\mathcal{N}_i} \left(\|\theta'_{i,t+1} - 2\theta_{i,t} + \theta_{i,t-1}\|^2 - \|\theta_{j,t+1} - 2\theta_{j,t} + \theta_{j,t-1}\|^2\right)\right] \tag{52}$$

$$= -C_t\deg(i)\mathbb{E}\left[\|\theta'_{i,t+1} - 2\theta_{i,t} + \theta_{i,t-1}\|^2 - \|\theta_{i,t+1} - 2\theta_{i,t} + \theta_{i,t-1}\|^2\right],$$

where we have used the fact that agent $i$'s manipulated gradient estimates at iteration $t$ do not affect the current model parameter $\theta_{i,t}$ in the second equality.

The right hand side of equation 52 satisfies

$$
\begin{aligned}
\mathbb{E}&\left[\|\theta'_{i,t+1} - 2\theta_{i,t} + \theta_{i,t-1}\|^2 - \|\theta_{i,t+1} - 2\theta_{i,t} + \theta_{i,t-1}\|^2\right] \\
&= \mathbb{E}[\|\theta'_{i,t+1} - \theta_{i,t+1} - (2\theta_{i,t} - \theta_{i,t-1} + \theta_{i,t+1})\|^2] - \mathbb{E}\left[\|\theta_{i,t+1} - 2\theta_{i,t} + \theta_{i,t-1}\|^2\right] \\
&\geq \frac{1}{2}\mathbb{E}[\|\theta'_{i,t+1} - \theta_{i,t+1}\|^2] - 2\mathbb{E}\left[\|\theta_{i,t+1} - 2\theta_{i,t} + \theta_{i,t-1}\|^2\right],
\end{aligned}
\tag{53}
$$

where in the last inequality we have used the inequality $\|a - b\|^2 \geq \frac{1}{2}\|a\|^2 - \|b\|^2$ valid for any $a, b \in \mathbb{R}^n$.

The last term on the right hand side of equation 53 satisfies

$$
\begin{aligned}
2\mathbb{E}\left[\Delta_{\theta_{i,t}}\right] &= 2\mathbb{E}\left[\|\theta_{i,t+1} - 2\theta_{i,t} + \theta_{i,t-1}\|^2\right] \\
&= 2\mathbb{E}\left[\left\|\sum_{j \in \mathcal{N}_i \cup \{i\}} w_{ij}(\theta_{j,t} - \theta_{j,t-1}) - \lambda_t g_i(\theta_{i,t}) + \lambda_{t-1} g_i(\theta_{i,t-1}) - (\theta_{i,t} - \theta_{i,t-1})\right\|^2\right] \\
&\leq 4\mathbb{E}\left[\|\lambda_t g_i(\theta_{i,t}) - \lambda_{t-1} g_i(\theta_{i,t-1})\|^2\right] \\
&\quad + 4\mathbb{E}\left[\left\|\sum_{j \in \mathcal{N}_i \cup \{i\}} w_{ij}(\theta_{j,t} - \theta_{i,t} - (\theta_{j,t-1} - \theta_{i,t-1}))\right\|^2\right].
\end{aligned}
\tag{54}
$$

The first term on the right hand side of equation 54 satisfies

$$
\begin{aligned}
4\mathbb{E}[\|\lambda_t g_i(\theta_{i,t}) &- \lambda_{t-1} g_i(\theta_{i,t-1})\|^2] \\
&\leq 8(\lambda_t - \lambda_{t-1})^2 \mathbb{E}[\|g_i(\theta_{i,t})\|^2] + 8\lambda_t^2 \mathbb{E}[\|g_i(\theta_{i,t}) - g_i(\theta_{i,t-1})\|^2] \\
&\leq 8(\lambda_t - \lambda_{t-1})^2 \mathbb{E}[\|g_i(\theta_{i,t})\|^2] + 8H^2\lambda_t^2 \mathbb{E}[\|\theta_{i,t} - \theta_{i,t-1}\|^2].
\end{aligned}
\tag{55}
$$

We proceed to estimate an upper bound on $\mathbb{E}[\|g_i(\theta_{i,t})\|^2]$ in equation 55 by using the following decomposition:

$$
\begin{aligned}
\mathbb{E}[\|g_i(\theta_{i,t})\|^2] &= \mathbb{E}[\|g_i(\theta_{i,t}) - \nabla f_i(\theta_{i,t})\|^2] + \mathbb{E}[\|\nabla f_i(\theta_{i,t}) - \nabla f_i(\theta_i^*)\|^2] \\
&\leq \sigma^2 + 2H_i^2 \mathbb{E}[\|\theta_{i,t} - \theta^*\|^2] + 2H_i^2\|\theta_i^* - \theta^*\|^2.
\end{aligned}
\tag{56}
$$

When $f_i(\theta)$ is strongly convex, according to Lemma 8 in Pu et al. (2021), we have that $\theta_{i,t}$ ($\theta_{i,t}$ is generated by Algorithm 1 when all agents are truthful) satisfies $\mathbb{E}[\|\theta_{i,t} - \theta^*\|^2] \leq \mathcal{O}(1)$. Hence, there must exist a constant $d_2 > 0$ such that $d_2 \geq \sigma^2 + 2H_i^2\mathcal{O}(1) + 2H_i^2\|\theta_i^* - \theta^*\|^2$ holds, which, combined with equation 56, leads to $\mathbb{E}[\|g_i(\theta_{i,t})\|^2] \leq d_2$.

On the other hand, when $f_i(\theta)$ is convex, we have $\mathbb{E}[\|g_i(\theta)\|^2] \leq L_f^2 + \sigma^2$ based on Assumption 1. By defining $d_3 = \max\{d_2, L_f^2 + \sigma^2\}$, we obtain $\mathbb{E}[\|g_i(\theta)\|^2] \leq d_3$. Therefore, equation 55 can be rewritten as $4\mathbb{E}[\|\lambda_t g_i(\theta_{i,t}) - \lambda_{t-1} g_i(\theta_{i,t-1})\|^2] \leq 8(\lambda_t - \lambda_{t-1})^2 d_3 + 8H^2\lambda_t^2 \mathbb{E}[\|\theta_{i,t} - \theta_{i,t-1}\|^2]$.

The last term on the right hand side of equation 55 satisfies

$$
\begin{aligned}
8H^2\lambda_t^2 \mathbb{E}[\|\theta_{i,t} - \theta_{i,t-1}\|^2] &= 8H^2\lambda_t^2 \mathbb{E}\left[\left\|\sum_{j \in \mathcal{N}_i \cup \{i\}} w_{ij}\theta_{j,t-1} - \lambda_{t-1} g_i(\theta_{i,t-1}) - \theta_{i,t-1}\right\|^2\right] \\
&\leq 16H^2\lambda_t^2 \mathbb{E}\left[\left\|\sum_{j \in \mathcal{N}_i \cup \{i\}} w_{ij}(\theta_{j,t-1} - \bar{\theta}_{t-1}) - (\theta_{i,t-1} - \bar{\theta}_{t-1})\right\|^2\right] + 16H^2\lambda_t^2\lambda_{t-1}^2 \mathbb{E}[\|g_i(\theta_{i,t-1})\|^2] \\
&\leq 16H^2\lambda_t^2 \mathbb{E}[\|\boldsymbol{\theta}_{t-1} - \mathbf{1}_N \otimes \bar{\theta}_{t-1}\|^2] + 16H^2\lambda_t^2\lambda_{t-1}^2 \mathbb{E}[\|g_i(\theta_{i,t-1})\|^2] \\
&\leq 16H^2\lambda_t^2 d_4\lambda_{t-1}^2 + 16d_3 H^2\lambda_t^2\lambda_{t-1}^2,
\end{aligned}
\tag{57}
$$

where in the last inequality we have used an argument similar to the consensus analysis in Nedic & Ozdaglar (2009).

By substituting equation 57 into equation 55, we arrive at

$$4\mathbb{E}\left[\|\lambda_t g_i(\theta_{i,t}) - \lambda_{t-1} g_i(\theta_{i,t-1})\|^2\right] \le 8d_3(\lambda_t - \lambda_{t-1})^2 + 16H^2\lambda_t^2 d_4\lambda_{t-1}^2 + 16d_3 H^2\lambda_t^2\lambda_{t-1}^2. \quad (58)$$

By defining an auxiliary variable $\Lambda_{i,t} = \theta_{i,t} - \bar{\theta}_t - (\theta_{i,t-1} - \bar{\theta}_{t-1})$, we arrive at the following result for the second term on the right hand side of equation 54:

$$4\mathbb{E}\left[\left\|\sum_{j\in\mathcal{N}_i\cup\{i\}} w_{ij}(\theta_{j,t} - \theta_{i,t} - (\theta_{j,t-1} - \theta_{i,t-1}))\right\|^2\right] = 4\mathbb{E}\left[\left\|\sum_{j\in\mathcal{N}_i\cup\{i\}} w_{ij}(\Lambda_{j,t} - \Lambda_{i,t})\right\|^2\right]. \quad (59)$$

According to the definitions $\boldsymbol{g}(\boldsymbol{\theta}_t) = \mathrm{col}(g_1(\theta_{1,t}),\cdots,g_N(\theta_{N,t}))$ and $\bar{g}(\boldsymbol{\theta}_t) = \frac{1}{N}\sum_{i=1}^N g_i(\theta_{i,t})$, we have

$$\Lambda_{t+1} = (W\otimes I_n)\Lambda_t - \lambda_t(\boldsymbol{g}(\boldsymbol{\theta}_t) - \mathbf{1}_N\otimes\bar{g}(\boldsymbol{\theta}_t)) + \lambda_{t-1}(\boldsymbol{g}(\boldsymbol{\theta}_{t-1}) - \mathbf{1}_N\otimes\bar{g}(\boldsymbol{\theta}_{t-1})),$$

which implies the following inequality:

$$\mathbb{E}[\|\Lambda_{t+1}\|^2] \le (1 + (1-\rho))\rho^2\mathbb{E}[\|\Lambda_t\|^2]$$
$$+ \left(1 + \frac{1}{1-\rho}\right)\mathbb{E}\left[\|\lambda_t(\boldsymbol{g}(\boldsymbol{\theta}_t) - \mathbf{1}_N\otimes\bar{g}(\boldsymbol{\theta}_t)) + \lambda_{t-1}(\boldsymbol{g}(\boldsymbol{\theta}_{t-1}) - \mathbf{1}_N\otimes\bar{g}(\boldsymbol{\theta}_{t-1}))\|^2\right]. \quad (60)$$

By using the relationship $\sum_{i=1}^N\|a_i - \frac{1}{N}\sum_{j=1}^N a_j\|^2 \le \sum_{i=1}^N\|a_i\|^2$ for any $a_i\in\mathbb{R}^n$, we can rewrite the last term on the right hand side of equation 60 as

$$\mathbb{E}[\|\lambda_t(\boldsymbol{g}(\boldsymbol{\theta}_t) - \mathbf{1}_N\otimes\bar{g}(\boldsymbol{\theta}_t)) + \lambda_{t-1}(\boldsymbol{g}(\boldsymbol{\theta}_{t-1}) - \mathbf{1}_N\otimes\bar{g}(\boldsymbol{\theta}_{t-1}))\|^2]$$
$$\le \sum_{i=1}^N\mathbb{E}[\|\lambda_t g_i(\theta_{i,t}) - \lambda_{t-1}g_i(\theta_{i,t-1})\|^2] \quad (61)$$
$$\le 8Nd_3(\lambda_t - \lambda_{t-1})^2 + 16NH^2\lambda_t^2 d_4\lambda_{t-1}^2 + 16Nd_3 H^2\lambda_t^2\lambda_{t-1}^2,$$

where we have used equation 58 in the last inequality.

Substituting equation 61 into equation 60, we obtain

$$\mathbb{E}[\|\Lambda_{t+1}\|^2] \le (1 - (1-\rho))\mathbb{E}[\|\Lambda_t\|^2] + d_5\lambda_t^2\lambda_{t-1}^2, \quad (62)$$

where $d_5$ is given by $d_5 = \frac{(2-\rho)8Nd_3(\lambda_1-\lambda_0)^2}{(1-\rho)\lambda_1^2\lambda_0^2} + \frac{(2-\rho)16NH^2 d_4 + 16Nd_3 H^2}{1-\rho}$.

By combining Lemma 11 in Chen & Wang (2025) and equation 62, we arrive at

$$\mathbb{E}[\|\Lambda_t\|^2] \le d_6\lambda_t^2\lambda_{t-1}^2, \quad (63)$$

where the constant $d_6$ is given by $d_6 = \left(\frac{16v}{e\ln(\frac{2}{1+\rho})}\right)^{4v}\left(\frac{\Lambda_1\rho}{d_5\lambda_1^4} + \frac{2}{1-\rho}\right)$.

Substituting equation 63 into equation 59 and then substituting equation 58 and equation 59 into equation 54, we obtain

$$2\mathbb{E}\left[\|\theta_{i,t+1} - 2\theta_{i,t} + \theta_{i,t-1}\|^2\right] \le d_7\lambda_t^2\lambda_{t-1}^2, \quad (64)$$

where the constant $d_7$ is given by $d_7 = 4d_6 + \frac{d_5(1-\rho)}{N(2-\rho)}$.

Substituting equation 64 into equation 53 and then substituting equation 53 into equation 52, we arrive at

$$\mathbb{E}\left[P_{i,t}(h_{i,t}, h_{-i,t}) - P_{i,t}(\alpha_{i,t}, h_{-i,t})\right] \le -\frac{\lambda_t^2}{2}C_t\deg(i)\mathbb{E}[\|(a_{i,t}-1)g_i(\theta_{i,t})\|^2]$$
$$-\frac{\lambda_t^2}{2}C_t\deg(i)\mathbb{E}[\|b_{i,t}\xi_{i,t}\|^2] + d_7\deg(i)C_t\lambda_t^2\lambda_{t-1}^2. \quad (65)$$

According to the update rule of Algorithm 1, we have $\theta'_{i,t+1} = \theta_{i,t+1} + \lambda_t(1-a_{i,t})g_i(\theta_{i,t}) + b_{i,t}\xi_{i,t}$, which implies

$$
\begin{aligned}
&N\mathbb{E}[\|\bar{\theta}_{t+1} - \bar{\theta}'_{t+1}\|^2] + \sum_{i=1}^{N}\mathbb{E}\left[\|\theta_{i,t+1} - \theta'_{i,t+1} - (\bar{\theta}_{t+1} - \bar{\theta}'_{t+1})\|^2\right] \\
&\leq \lambda_t^2\left(1 + \frac{N-1}{N} + \left(\frac{N-1}{N}\right)^2\right)\left(\mathbb{E}\left[(1-a_{i,t})^2\|g_i(\theta_{i,t})\|^2\right] + \mathbb{E}\left[\|b_{i,t}\xi_{i,t}\|^2\right]\right) \\
&\leq 3\lambda_t^2\left(\mathbb{E}\left[(1-a_{i,t})^2\|g_i(\theta_{i,t})\|^2\right] + \mathbb{E}\left[\|b_{i,t}\xi_{i,t}\|^2\right]\right).
\end{aligned}
\tag{66}
$$

By combining equation 36 from Lemma 8 and equation 66, we obtain

$$
\mathbb{E}[\|\theta_{i,T+1} - \theta'_{i,T+1}\|^2] \leq 3d_{t+1\to T+1}\lambda_t^2\left(\mathbb{E}\left[(1-a_{i,t})^2\|g_i(\theta_{i,t})\|^2\right] + \mathbb{E}\left[\|b_{i,t}\xi_{i,t}\|^2\right]\right).
\tag{67}
$$

Since $\theta_{i,T+1}$ is a constant under a fixed $T > 0$, we denote $d_8 \triangleq \mathbb{E}[\|\theta_{i,T+1} - \theta^*\|^2]$. Following an argument similar to the derivation of Proposition 3 in Lobel & Ozdaglar (2010), we have

$$
\begin{aligned}
&\left|\mathbb{E}\left[\|\theta'_{i,T+1} - \theta^*\|^2 - \|\theta_{i,T+1} - \theta^*\|^2\right]\right| \\
&\leq \mathbb{E}\left[\|\theta'_{i,T+1} - \theta_{i,T+1}\|^2\right] + 2d_8\mathbb{E}\left[\|\theta'_{i,T+1} - \theta_{i,T+1}\|\right],
\end{aligned}
\tag{68}
$$

which, combined with the $L_{R,i}$-Lipschitz continuity of $R_i(f_i(\theta))$ w.r.t. $\theta$, leads to

$$
\begin{aligned}
&\mathbb{E}\left[|R_i(f_i(\theta'_{i,T+1})) - R_i(f_i(\theta_{i,T+1}))|\right] \\
&\leq L_{R,i}\sqrt{3d_{t+1\to T+1}}\lambda_t\left(\sqrt{\mathbb{E}\left[(a_{i,t}-1)^2\|g_i(\theta_{i,t})\|^2\right]} + \sqrt{\mathbb{E}\left[\|b_{i,t}\xi_{i,t}\|^2\right]}\right).
\end{aligned}
$$

By using the preceding inequality and equation 65, we obtain

$$
\begin{aligned}
&\mathbb{E}\left[R_i(f_i(\theta'_{i,T+1})) - K\|\theta'_{i,T+1} - \theta^*\|^2 - P_{i,t}(\alpha_{i,t}, \alpha_{-i,t})\right] \\
&\quad - \mathbb{E}\left[R_i(f_i(\theta_{i,T+1})) - K\|\theta_{i,T+1} - \theta^*\|^2 - P_{i,t}(h_{i,t}, \alpha_{-i,t})\right] \\
&\leq \mathbb{E}\left[|R_i(f_i(\theta'_{i,T+1})) - R_i(f_i(\theta_{i,T+1}))| + P_{i,t}(h_{i,t}, \alpha_{-i,t}) - P_{i,t}(\alpha_{i,t}, \alpha_{-i,t})\right. \\
&\quad \left. + K\|\theta'_{i,T+1} - \theta^*\|^2 - \|\theta_{i,T+1} - \theta^*\|^2|\right] \\
&= (L_{R,i}\sqrt{3d_{t+1\to T+1}} + \sqrt{6d_{t+1\to T+1}d_8})\lambda_t\sqrt{\mathbb{E}\left[(a_{i,t}-1)^2\|g_i(\theta_{i,t})\|^2\right]} \\
&\quad - \left(\frac{C_t\deg(i)\lambda_t^2}{2} + 3Kd_{t+1\to T+1}\right)\mathbb{E}\left[(a_{i,t}-1)^2\|g_i(\theta_{i,t})\|^2\right] \\
&\quad + (L_{R,i}\sqrt{3d_{t+1\to T+1}} + \sqrt{6d_{t+1\to T+1}d_8})\lambda_t\sqrt{\mathbb{E}\left[\|b_{i,t}\xi_{i,t}\|^2\right]} \\
&\quad - \left(\frac{C_t\deg(i)\lambda_t^2}{2} + 3Kd_{t+1\to T+1}\right)\mathbb{E}[\|b_{i,t}\xi_{i,t}\|^2] + \lambda_t^2\lambda_{t-1}^2 C_t d_7\deg(i).
\end{aligned}
\tag{69}
$$

It can be seen that the first and second terms on the right hand side of equation 69 together form a downward-opening quadratic, whose positive root is

$$
\begin{aligned}
\sqrt{\mathbb{E}\left[(a_{i,t}-1)^2\|g_i(\theta_{i,t})\|^2\right]} &= \frac{(L_{R,i}\sqrt{3d_{t+1\to T+1}} + \sqrt{6d_{t+1\to T+1}d_8})\lambda_t}{C_t\deg(i)\lambda_t^2 + 6Kd_{t+1\to T+1}} \\
&+ \frac{\sqrt{(L_{R,i}\sqrt{3d_{t+1\to T+1}} + \sqrt{6d_{t+1\to T+1}d_8})^2\lambda_t^2 + 2(C_t\deg(i)\lambda_t^2 + 6Kd_{t+1\to T+1})\lambda_t^2\lambda_{t-1}^2 C_t d_7\deg(i)}}{C_t\deg(i)\lambda_t^2 + 6Kd_{t+1\to T+1}}.
\end{aligned}
\tag{70}
$$

Similarly, the third and fourth terms on the right hand side of equation 69 together form a downward-opening quadratic, whose positive root is

$$
\begin{aligned}
\sqrt{\mathbb{E}\left[\|b_{i,t}\xi_{i,t}\|^2\right]} &= \frac{(L_{R,i}\sqrt{3d_{t+1\to T+1}} + \sqrt{6d_{t+1\to T+1}d_8})\lambda_t}{C_t\deg(i)\lambda_t^2 + 6Kd_{t+1\to T+1}} \\
&+ \frac{\sqrt{(L_{R,i}\sqrt{3d_{t+1\to T+1}} + \sqrt{6d_{t+1\to T+1}d_8})^2\lambda_t^2 + 2(C_t\deg(i)\lambda_t^2 + 6Kd_{t+1\to T+1})\lambda_t^2\lambda_{t-1}^2 C_t d_7\deg(i)}}{C_t\deg(i)\lambda_t^2 + 6Kd_{t+1\to T+1}}.
\end{aligned}
\tag{71}
$$

From equation 70 and equation 71, we have that for any agent $i$, its optimal action $(a_{i,t}, b_{i,t})$ at iteration $t$ must satisfy $\sqrt{\mathbb{E}[(a_{i,t}-1)^2\|g_i(\theta_{i,t})\|^2]} \leq \frac{\sqrt{2(C_t \deg(i)\lambda_t^2+6Kd_{t+1\to T+1})\lambda_t^2\lambda_{t-1}^2 C_t d_7 \deg(i)}}{C_t \deg(i)\lambda_t^2+6Kd_{t+1\to T+1}} +$ $\frac{2(L_{R,i}\sqrt{3d_{t+1\to T+1}}+\sqrt{6d_{t+1\to T+1}d_8})\lambda_t}{C_t \deg(i)\lambda_t^2+6Kd_{t+1\to T+1}}$ and $\sqrt{\mathbb{E}[\|b_{i,t}\xi_{i,t}\|^2]} \leq \frac{2(L_{R,i}\sqrt{3d_1}+\sqrt{6d_{t+1\to T+1}d_8})\lambda_t}{C_t \deg(i)\lambda_t^2+6Kd_{t+1\to T+1}} +$ $\frac{\sqrt{2(C_t \deg(i)\lambda_t^2+6Kd_{t+1\to T+1})\lambda_t^2\lambda_{t-1}^2 C_t d_7 \deg(i)}}{C_t \deg(i)\lambda_t^2+6Kd_{t+1\to T+1}}$. These two inequalities ensure $\mathbb{E}[R_i(f_i(\theta'_{i,T+1})) - K\|\theta'_{i,T+1}-\theta^*\|^2 - P_{i,t}(\alpha_{i,t}, h_{-i,t})] - \mathbb{E}[R_i(f_i(\theta_{i,T+1})) - K\|\theta_{i,T+1}-\theta^*\|^2 - P_{i,t}(h_{i,t}, h_{-i,t})] \geq 0$.

For any given $\delta > 0$, when we set the payment coefficient $C_t$ as

$$C_t = \frac{C_0 \kappa_t^2}{\delta^2(t+1)^{-2v}} \text{ with } C_0 = \min\left\{ \frac{8(L_R\sqrt{3}+\sqrt{6d_8})^4}{63d_7 \deg_{\max}(Kd_{1\to\infty})^3}, \frac{4(L_R\sqrt{3}+\sqrt{6d_8})^2}{9Kd_{1\to\infty}\deg_{\max}} \right\}, \quad (72)$$

we arrive at equation 51 and prove Lemma 1. Here, $d_{1\to\infty}$ is given by $d_{1\to\infty} = 2e^{\frac{10vH^2(2-\rho)\lambda_0^2}{(1-\rho)(2v-1)}}$ and $\deg_{\max} = \max_{i\in[N]}\{\deg(i)\}$.

Furthermore, since $\kappa_t$ is a decaying sequence, we obtain $\lim_{t\to\infty} \mathbb{E}[\|a_{i,t}g_i(\theta_{i,t}) - g_i(\theta_{i,t})\|] = 0$ and $\lim_{t\to\infty} \mathbb{E}[\|b_{i,t}\xi_{i,t}\|] = 0$, which imply $\lim_{t\to\infty} \mathbb{E}[\|\alpha_{i,t}(g_i(\theta_{i,t})) - g_i(\theta_{i,t})\|] = 0$. $\square$

## D.2 INTUITIVE EXPLANATION AND RIGOROUS ANALYSIS OF CONVERGENCE GUARANTEE

According to the discussion in Section 3.2 in the main text, we know that each agent $i$ can strategically select a (possibly random) action $(a_{i,t}, b_{i,t})$ at every iteration. In this case, gradient manipulation by agents causes Algorithm 1 to essentially minimize a time-varying global objective function, i.e., $F_t(\theta) = \frac{1}{N}\sum_{i=1}^N f_{i,t}(\theta)$ with $f_{i,t}(\theta) = \mathbb{E}[a_{i,t}f_i(\theta) + b_{i,t}\xi_{i,t}^\top\theta]$. Nevertheless, Lemma 1 proves that as the number of iterations tends to infinity, the optimal action for agent $i$ is fully truthful, i.e., an agent will have zero incentive to deviate from truthful behaviors. Building on this, we can ensure that an optimal solution to the modified stochastic optimization problem $\min F_t(\theta)$ converges to an optimal solution to the original problem in equation 1. We present the following Lemma 10 to formally characterize this property, which serves as a theoretical foundation for our subsequent convergence analysis of Algorithm 1 in the presence of persistent strategic manipulation by agents.

**Lemma 10.** *We denote $\theta^*$ as an optimal solution to the original problem in equation 1 and $\theta'^*_t$ as an optimal solution to problem $\min F_t(\theta)$ at iteration $t$. Under the conditions in Lemma 1, for any given $\delta > 0$ and $t \geq 0$, if $F(\theta)$ is $\mu$-strongly convex, then we have*

$$\mathbb{E}[\|\theta'^*_t - \theta^*\|^2] \leq 4\mu^{-2}\kappa_t^2\delta^2. \quad (73)$$

*Furthermore, if $F(\theta)$ is general convex, then we have*

$$\mathbb{E}[F(\theta'^*_t) - F(\theta^*)] \leq d_{\Theta^*}\kappa_t\delta, \quad (74)$$

*where $d_{\Theta^*}$ denotes the diameter of $\Theta^* = \{\theta'^*_0, \theta'^*_1, \cdots, \theta'^*_T, \theta^*\}$ for any $T \in \mathbb{N}$.*

*Proof.* From the inequality $F_t(\theta'^*_t) \leq F_t(\theta^*)$, we obtain

$$\mathbb{E}[F(\theta'^*_t) - F(\theta^*)] \leq \mathbb{E}[F(\theta'^*_t) - F_t(\theta'^*_t)] - \mathbb{E}[F(\theta^*) - F_t(\theta^*)]. \quad (75)$$

By applying the mean value theorem to equation 75, we obtain

$$\mathbb{E}[F(\theta'^*_t) - F_t(\theta'^*_t)] - \mathbb{E}[F(\theta^*) - F_t(\theta^*)] = \mathbb{E}[\langle\nabla F(\chi) - \nabla F_t(\chi), \theta'^*_t - \theta^*\rangle]$$
$$\leq \mathbb{E}[\|\nabla F(\chi) - \nabla F_t(\chi)\|\|\theta'^*_t - \theta^*\|], \quad (76)$$

where the variable $\chi$ is given by $\chi = \tau\theta'^*_t + (1-\tau)\theta^*$ for some $\tau \in (0,1)$.

According to the definitions of $\nabla F(\theta)$ and $\nabla F_t(\theta)$, we have

$$\mathbb{E}[\|\nabla F(\chi) - \nabla F_t(\chi)\|] \leq \left\|\frac{1}{N}\sum_{i=1}^N \mathbb{E}[g_i(\chi)] - \frac{1}{N}\sum_{i=1}^N \mathbb{E}[a_{i,t}g_i(\chi) + b_{i,t}\xi_{i,t}]\right\|$$
$$\leq \frac{1}{N}\sum_{i=1}^N \mathbb{E}[\|g_i(\chi) - a_{i,t}g_i(\chi)\|] \leq \kappa_t\delta, \quad (77)$$

where we have used $\mathbb{E}[\xi_{i,t}] = 0$ in the second inequality and equation 51 in the last inequality.

Substituting equation 77 into equation 76 and then substituting equation 76 into equation 75, we arrive at

$$\mathbb{E}\left[F(\theta_t'^*) - F(\theta^*)\right] \leq \kappa_t \delta \mathbb{E}\left[\|\theta_t'^* - \theta^*\|\right] \leq d_{\Theta^*} \kappa_t \delta, \tag{78}$$

where $d_{\Theta^*}$ denotes the diameter of $\Theta^* = \{\theta_0'^*, \theta_1'^*, \cdots, \theta_T'^*, \theta^*\}$. Equation 78 directly implies equation 74 in Lemma 10.

Furthermore, if $F(\theta)$ is $\mu$-strongly convex, we have $\frac{\mu}{2}\|\theta_t'^* - \theta^*\|^2 \leq F(\theta_t'^*) - F(\theta^*)$. By using equation 78, we obtain

$$\frac{\mu}{2}\mathbb{E}\left[\|\theta_t'^* - \theta^*\|^2\right] \leq \kappa_t \delta \mathbb{E}\left[\|\theta_t'^* - \theta^*\|\right]. \tag{79}$$

Equation 79 implies $\mathbb{E}[\|\theta_t'^* - \theta^*\|] \leq 2\mu^{-1}\kappa_t\delta$, which, when substituted into equation 79, leads to equation 73 in Lemma 10. □

Lemma 10 proves that when the optimal actions of participating agents are $\kappa_t\delta$-truthful at iteration $t$ (as proven in Lemma 1), the time-varying optimal solutions $\theta_t'^*$ will converge to an optimal solution $\theta^*$ to the original problem in equation 1. This is key to ensuring accurate convergence of Algorithm 1 under persistent strategic manipulation by agents over iterations. It is worth noting that this result is unattainable in most existing truthfulness results (in, e.g., Dorner et al. (2023); Chakarov et al. (2024a;b)), where the payment mechanisms cannot ensure that the deviated optimal solution aligns with the optimal solution to the problem in equation 1, which hence leads to an inherent optimization error proportional to $\|\theta_\infty'^* - \theta^*\|$, where $\theta_\infty'^*$ denotes the convergence point of the iterative algorithm in the presence of strategic manipulation.

# E  PROOF OF CONVERGENCE RATES IN THEOREM 1

In this section, we establish the convergence rates of Algorithm 1 in the presence of strategic behavior, for strongly convex and general convex objective functions, respectively. Specifically, the convergence rate for a strongly convex $F(\theta)$ is presented in Section E.1 and for a general convex $F(\theta)$ in Section E.2. Furthermore, the computational complexity analysis of Algorithm 1 under our payment mechanism is summarized in Section E.3, and the diminishing payment analysis of our payment mechanism is provided in Section E.4.

## E.1  CONVERGENCE RATE FOR A STRONGLY CONVEX GLOBAL OBJECTIVE FUNCTION

The following Lemma 11 summarizes the convergence result of Algorithm 1 in the presence of strategic behavior for a strongly convex $F(\theta)$, which corresponds to Theorem 1-(i) in the main text (note that $\theta_{i,T}'$ in Lemma 11 corresponds to $\theta_{i,T}$ in Theorem 1-(i)).

**Lemma 11.** *We denote $\theta^*$ as a solution to the problem in equation 1. Under Mechanism 1 and the conditions in Lemma 1, for any $i \in [N]$ and $T \geq 0$, if the global objective function $F(\theta)$ is $\mu$-strongly convex (not necessarily Lipschitz continuous), then the following inequalities hold for Algorithm 1:*

$$\mathbb{E}[\|\theta_{i,T}' - \bar{\theta}_T'\|^2] \leq \mathcal{O}\left(\frac{H^2(\delta^2 + \sigma^2)}{\mu(1-\rho)(T+1)^v}\right);$$

$$\mathbb{E}[\|\theta_{i,T}' - \theta^*\|^2] \leq \mathcal{O}\left(\frac{H^2(\delta^2 + \sigma^2)}{\mu(1-\rho)^2(T+1)^v}\right). \tag{80}$$

*Proof.* By using the dynamics of $\theta_{i,t+1}$ in Algorithm 1 and the definitions $\boldsymbol{\theta}_t' = \mathrm{col}(\theta_{1,t}', \cdots, \theta_{N,t}')$ and $\boldsymbol{m}_t = \mathrm{col}(m_{1,t}, \cdots, m_{N,t})$ with $m_{i,t} = a_{i,t}g_i(\theta_{i,t}') + b_{i,t}\xi_{i,t}$, we obtain

$$\mathbb{E}\left[\|\boldsymbol{\theta}_{t+1}' - \mathbf{1}_N \otimes \theta^*\|^2\right] \leq \mathbb{E}\left[\|(W \otimes I_n - I_{Nn})\boldsymbol{\theta}_t' - \lambda_t \boldsymbol{m}_t\|^2\right] + \mathbb{E}\left[\|\boldsymbol{\theta}_t' - \mathbf{1}_N \otimes \theta^*\|^2\right]$$
$$+ 2\mathbb{E}\left[\langle(W \otimes I_n - I_{Nn})\boldsymbol{\theta}_t' - \lambda_t \boldsymbol{m}_t, \boldsymbol{\theta}_t' - \mathbf{1}_N \otimes \theta^*\rangle\right]. \tag{81}$$

Using the relation $W\mathbf{1}_N^\top = \mathbf{1}_N$, the first term on the right hand side of equation 81 satisfies

$$
\begin{aligned}
\mathbb{E}\left[\|(W \otimes I_n - I_{Nn})\boldsymbol{\theta}'_t - \lambda_t \boldsymbol{m}_t\|^2\right] &= \mathbb{E}\left[\|(W \otimes I_n - I_{Nn})(\boldsymbol{\theta}'_t - \mathbf{1}_N \otimes \bar{\theta}'_t) - \lambda_t \boldsymbol{m}_t\|^2\right] \\
&\leq 2\lambda_t^2 \mathbb{E}\left[\|\boldsymbol{m}_t\|^2\right] \\
&\quad + 2\mathbb{E}\left[\|(W \otimes I_n - I_{Nn})(\boldsymbol{\theta}'_t - \mathbf{1}_N \otimes \bar{\theta}'_t)\|^2\right].
\end{aligned}
\tag{82}
$$

Assumption 1 and equation 51 imply that the first term on the right hand side of equation 82 satisfies

$$
\begin{aligned}
\mathbb{E}\left[\|\boldsymbol{m}_t\|^2\right] &= \sum_{i=1}^N \mathbb{E}\left[\|a_{i,t}g_i(\theta'_{i,t}) - g_i(\theta'_{i,t})\|^2\right] + \sum_{i=1}^N \mathbb{E}\left[\|g_i(\theta'_{i,t}) - \nabla f_i(\theta'_{i,t})\|^2\right] \\
&\quad + \sum_{i=1}^N \mathbb{E}\left[\|\nabla f_i(\theta'_{i,t})\|^2\right] + \sum_{i=1}^N \mathbb{E}\left[\|b_{i,t}\xi_{i,t}\|^2\right] \\
&\quad + \sum_{i=1}^N 2\mathbb{E}\left[\langle a_{i,t}g_i(\theta'_{i,t}) - g_i(\theta'_{i,t}), \nabla f_i(\theta'_{i,t})\rangle\right] \\
&\leq N(3\kappa_t^2\delta^2 + \sigma^2) + 2\sum_{i=1}^N \mathbb{E}\left[\|\nabla f_i(\theta'_{i,t}) - \nabla f_i(\theta_i^*)\|^2\right] \\
&\leq N(3\kappa_t^2\delta^2 + \sigma^2) + 4H^2\mathbb{E}\left[\|\boldsymbol{\theta}'_t - \mathbf{1}_N \otimes \theta^*\|^2\right] + 4H^2\mathbb{E}\left[\|\mathbf{1}_N \otimes \theta^* - \boldsymbol{\theta}^*\|^2\right],
\end{aligned}
\tag{83}
$$

where $\theta^*$ represents a solution to problem 1 and $\boldsymbol{\theta}^*$ is given by $\boldsymbol{\theta}^* = \mathrm{col}(\theta_1^*, \cdots, \theta_N^*)$.

Using the relation $W\mathbf{1}_N^\top = \mathbf{1}_N$, the last term on the right hand side of equation 81 satisfies

$$
\begin{aligned}
&2\mathbb{E}\left[\langle(W \otimes I_n - I_{Nn})\boldsymbol{\theta}'_t - \lambda_t \boldsymbol{m}_t, \boldsymbol{\theta}'_t - \mathbf{1}_N \otimes \theta^*\rangle\right] \\
&= 2\mathbb{E}\left[(\boldsymbol{\theta}'_t - \mathbf{1}_N \otimes \theta^*)^\top (W \otimes I_n - I_{Nn})(\boldsymbol{\theta}'_t - \mathbf{1}_N \otimes \theta^*)\right] - 2\mathbb{E}\left[\langle \lambda_t \boldsymbol{m}_t, \boldsymbol{\theta}'_t - \mathbf{1}_N \otimes \theta^*\rangle\right] \\
&\leq 2\sum_{i=1}^N \mathbb{E}\left[\langle -\lambda_t(a_{i,t}g_i(\theta'_{i,t}) + b_{i,t}\xi_{i,t}) + \lambda_t \nabla f_i(\bar{\theta}'_t), \theta'_{i,t} - \theta^*\rangle\right] \\
&\quad - 2\sum_{i=1}^N \mathbb{E}\left[\langle \lambda_t \nabla f_i(\bar{\theta}'_t), \theta'_{i,t} - \theta^*\rangle\right],
\end{aligned}
\tag{84}
$$

where we have omitted the negative term $2\mathbb{E}\left[(\boldsymbol{\theta}'_t - \mathbf{1}_N \otimes \theta^*)^\top (W \otimes I_n - I_{Nn})(\boldsymbol{\theta}'_t - \mathbf{1}_N \otimes \theta^*)\right]$ in the last inequality due to $\rho = \max\{\pi_2|, |\pi_N|\} < 1$.

By using the Young's inequality and the relations $\mathbb{E}[\xi_{i,t}] = 0$ and $\mathbb{E}[g_i(\theta'_{i,t})] = \nabla f_i(\theta'_{i,t})$, the first term on the right hand side of equation 84 satisfies

$$
\begin{aligned}
&2\sum_{i=1}^N \mathbb{E}\left[\langle -\lambda_t(a_{i,t}g_i(\theta'_{i,t}) + b_{i,t}\xi_{i,t}) + \lambda_t \nabla f_i(\bar{\theta}'_t), \theta'_{i,t} - \theta^*\rangle\right] \\
&\leq \frac{4N}{\mu}\lambda_t\kappa_t^2\delta^2 + \frac{\mu}{4}\lambda_t\mathbb{E}\left[\|\boldsymbol{\theta}'_t - \mathbf{1}_N \otimes \theta^*\|^2\right] \\
&\quad + \frac{4}{\mu}\lambda_t H_i^2 \mathbb{E}\left[\|\boldsymbol{\theta}'_t - \mathbf{1}_N \otimes \bar{\theta}'_t\|^2\right] + \frac{\mu}{4}\lambda_t\mathbb{E}\left[\|\boldsymbol{\theta}'_t - \mathbf{1}_N \otimes \theta^*\|^2\right],
\end{aligned}
\tag{85}
$$

where we have used equation 51 and the $H_i$-smoothness of $f_i(\theta)$ in the derivation.

The last term on the right hand side of equation 84 satisfies

$$
\begin{aligned}
-2\sum_{i=1}^N \mathbb{E}\left[\langle \lambda_t \nabla f_i(\bar{\theta}'_t), \theta'_{i,t} - \theta^*\rangle\right] &\leq -2\sum_{i=1}^N \mathbb{E}\left[\langle \lambda_t \nabla f_i(\bar{\theta}'_t), \theta'_{i,t} - \bar{\theta}'_t\rangle\right] \\
&\quad - 2N\mathbb{E}\left[\langle \lambda_t(\nabla F(\bar{\theta}'_t) - \nabla F(\theta^*)), \bar{\theta}'_t - \theta^*\rangle\right].
\end{aligned}
\tag{86}
$$

Applying the Young's inequality to the first term on the right hand side of equation 86 yields

$$
-2\sum_{i=1}^{N} \mathbb{E}\left[\left\langle \lambda_t \nabla f_i(\bar{\theta}_t'), \theta_{i,t}' - \bar{\theta}_t' \right\rangle\right] = -2\sum_{i=1}^{N} \mathbb{E}\left[\left\langle \lambda_t \left(\nabla f_i(\bar{\theta}_t') - \nabla f_i(\theta_i^*)\right), \theta_{i,t}' - \bar{\theta}_t' \right\rangle\right]
$$

$$
\leq \lambda_t^2 H^2 \sum_{i=1}^{N} \mathbb{E}\left[\|\bar{\theta}_t' - \theta_i^*\|^2\right] + \mathbb{E}\left[\|\boldsymbol{\theta}_t' - \mathbf{1}_N \otimes \bar{\theta}_t'\|^2\right] \tag{87}
$$

$$
\leq 2\lambda_t^2 H^2 N \mathbb{E}\left[\|\bar{\theta}_t' - \theta^*\|^2\right] + 2\lambda_t^2 H^2 \mathbb{E}\left[\|\boldsymbol{\theta}^* - \mathbf{1}_N \otimes \theta^*\|^2\right] + \mathbb{E}\left[\|\boldsymbol{\theta}_t' - \mathbf{1}_N \otimes \bar{\theta}_t'\|^2\right].
$$

By using the $\mu$-strong convexity of $F(\theta)$, we have that the last term on the right hand side of equation 86 satisfies $-2N\mathbb{E}[\langle \lambda_t(\nabla F(\bar{\theta}_t') - \nabla F(\theta^*)), \bar{\theta}_t' - \theta^* \rangle] \leq -2N\lambda_t \mu \mathbb{E}[\|\bar{\theta}_t' - \theta^*\|^2]$, which, combined with equation 87, leads to

$$
-2\sum_{i=1}^{N} \mathbb{E}\left[\left\langle \lambda_t \nabla f_i(\bar{\theta}_t'), \theta_{i,t}' - \theta^* \right\rangle\right] \leq \mathbb{E}\left[\|\boldsymbol{\theta}_t' - \mathbf{1}_N \otimes \bar{\theta}_t'\|^2\right] + 2H^2 N \lambda_t^2 \mathbb{E}\left[\|\theta_i^* - \theta^*\|^2\right]
$$

$$
- 2\lambda_t \left(\mu - 2H^2\lambda_t\right) \left(\frac{1}{2}\mathbb{E}\left[\|\boldsymbol{\theta}_t' - \mathbf{1}_N \otimes \theta^*\|^2\right] - \mathbb{E}\left[\|\boldsymbol{\theta}_t' - \mathbf{1}_N \otimes \bar{\theta}_t'\|^2\right]\right) \tag{88}
$$

$$
\leq -\lambda_t \left(\mu - 2H^2\lambda_t\right) \mathbb{E}\left[\|\boldsymbol{\theta}_t' - \mathbf{1}_N \otimes \theta^*\|^2\right]
$$

$$
+ \left(2\lambda_t \left(\mu - 2H^2\lambda_t\right) + 1\right) \mathbb{E}\left[\|\boldsymbol{\theta}_t' - \mathbf{1}_N \otimes \bar{\theta}_t'\|^2\right] + 2H^2\lambda_t^2 \mathbb{E}\left[\|\mathbf{1}_N \otimes \theta^* - \boldsymbol{\theta}^*\|^2\right],
$$

where we have used $N\mathbb{E}[\|\bar{\theta}_t' - \theta^*\|^2] = \sum_{i=1}^{N} \mathbb{E}[\|\bar{\theta}_t' - \theta_{i,t}' + \theta_{i,t}' - \theta^*\|^2] \geq \frac{1}{2}\sum_{i=1}^{N} \mathbb{E}[\|\theta_{i,t}' - \theta^*\|^2] - \sum_{i=1}^{N} \mathbb{E}[\|\theta_{i,t}' - \bar{\theta}_t'\|^2]$ in the second inequality.

Substituting equation 85 and equation 88 into equation 84, we arrive at

$$
2\mathbb{E}\left[\langle (W \otimes I_n - I_{Nn})\boldsymbol{\theta}_t' - \lambda_t \boldsymbol{m}_t, \boldsymbol{\theta}_t' - \mathbf{1}_N \otimes \theta^* \rangle\right]
$$

$$
\leq \left(2H^2\lambda_t - \frac{\mu}{2}\right)\lambda_t \mathbb{E}\left[\|\boldsymbol{\theta}_t' - \mathbf{1}_N \otimes \theta^*\|^2\right] + 2H^2\lambda_t^2 \mathbb{E}\left[\|\boldsymbol{\theta}^* - \mathbf{1}_N \otimes \theta^*\|^2\right] \tag{89}
$$

$$
+ \left(\frac{4H^2}{\mu}\lambda_t + 2\lambda_t \left(\mu - 2H^2\lambda_t\right) + 1\right) \mathbb{E}\left[\|\boldsymbol{\theta}_t' - \mathbf{1}_N \otimes \bar{\theta}_t'\|^2\right] + \frac{4N}{\mu}\lambda_t \kappa_t^2 \delta^2.
$$

Further substituting equation 83 into equation 82, and then substituting equation 82 and equation 89 into equation 81, we obtain

$$
\mathbb{E}\left[\|\boldsymbol{\theta}_{t+1}' - \mathbf{1}_N \otimes \theta^*\|^2\right] \leq \left(1 - \frac{\mu}{2}\lambda_t + 10H^2\lambda_t^2\right) \mathbb{E}\left[\|\boldsymbol{\theta}_t' - \mathbf{1}_N \otimes \theta^*\|^2\right]
$$

$$
+ (5 + 2\lambda_t \mu) \mathbb{E}\left[\|\boldsymbol{\theta}_t' - \mathbf{1}_N \otimes \bar{\theta}_t'\|^2\right] + 10H^2\lambda_t^2 \mathbb{E}\left[\|\boldsymbol{\theta}^* - \mathbf{1}_N \otimes \theta^*\|^2\right] \tag{90}
$$

$$
+ N(6\lambda_0 + 4\mu^{-1})\lambda_t \kappa_t^2 \delta^2 + 2N\lambda_t^2 \sigma^2,
$$

where we have omitted the negative term $-4H^2\lambda_t^2 \mathbb{E}\left[\|\boldsymbol{\theta}_t' - \mathbf{1}_N \otimes \bar{\theta}_t'\|^2\right]$ in the derivation.

We proceed to characterize the consensus term $\mathbb{E}\left[\|\boldsymbol{\theta}_t' - \mathbf{1}_N \otimes \bar{\theta}_t'\|^2\right]$ in equation 90.

According to the definitions $\boldsymbol{\theta}_t' = \mathrm{col}(\theta_{1,t}', \cdots, \theta_{N,t}')$, $\boldsymbol{m}_t = \mathrm{col}(m_{1,t}, \cdots, m_{N,t})$, and $\bar{m}_t = \frac{1}{N}\sum_{i=1}^{N} m_{i,t}$ with $m_{i,t} = a_{i,t} g_i(\theta_{i,t}') + b_{i,t}\xi_{i,t}$, we have

$$
\mathbb{E}\left[\|\boldsymbol{\theta}_{t+1}' - \mathbf{1}_N \otimes \bar{\theta}_{t+1}'\|^2\right] = \mathbb{E}\left[\|(W \otimes I_n)\boldsymbol{\theta}_t' - \lambda_t(\boldsymbol{m}_t - \mathbf{1}_N \otimes \bar{m}_t) - \mathbf{1}_N \otimes \bar{\theta}_t'\|^2\right]
$$

$$
\leq (1 + (1 - \rho)) \mathbb{E}\left[\|(W \otimes I_n)(\boldsymbol{\theta}_t' - \mathbf{1}_N \otimes \bar{\theta}_t')\|^2\right] + \left(1 + \frac{1}{1 - \rho}\right)\lambda_t^2 \mathbb{E}\left[\|\boldsymbol{m}_t\|^2\right],
$$

where we have used the relationship $\mathbb{E}[\|\boldsymbol{m}_t - \mathbf{1}_N \otimes \bar{m}_t\|^2] \leq \mathbb{E}[\|\boldsymbol{m}_t\|^2]$.

By using the inequality $(1 + 1 - \rho)\rho = 1 - (1 - \rho)^2 < 1$ (note $\rho < 1$), we obtain

$$
\mathbb{E}\left[\|\boldsymbol{\theta}_{t+1}' - \mathbf{1}_N \otimes \bar{\theta}_{t+1}'\|^2\right] \leq (1 + (1 - \rho))\rho^2 \mathbb{E}\left[\|\boldsymbol{\theta}_t' - \mathbf{1}_N \otimes \bar{\theta}_t'\|^2\right] + \frac{2 - \rho}{1 - \rho}\lambda_t^2 \mathbb{E}\left[\|\boldsymbol{m}_t\|^2\right]
$$

$$
\leq \rho \mathbb{E}\left[\|\boldsymbol{\theta}_t' - \mathbf{1}_N \otimes \bar{\theta}_t'\|^2\right] + \frac{2 - \rho}{1 - \rho}\lambda_t^2 \mathbb{E}\left[\|\boldsymbol{m}_t\|^2\right]. \tag{91}
$$

Substituting equation 83 into equation 91, we have

$$\mathbb{E}\left[\|\boldsymbol{\theta}'_{t+1} - \mathbf{1}_N \otimes \bar{\theta}'_{t+1}\|^2\right] \leq \frac{4H^2(2-\rho)}{1-\rho}\lambda_t^2 \mathbb{E}\left[\|\boldsymbol{\theta}'_t - \mathbf{1}_N \otimes \theta^*\|^2\right] + \rho\mathbb{E}\left[\|\boldsymbol{\theta}'_t - \mathbf{1}_N \otimes \bar{\theta}'_t\|^2\right]$$

$$+ \frac{4H^2(2-\rho)}{1-\rho}\lambda_t^2 \mathbb{E}\left[\|\theta^* - \mathbf{1}_N \otimes \theta^*\|^2\right] + \frac{3N(2-\rho)\lambda_0}{1-\rho}\lambda_t \kappa_t^2 \delta^2 + \frac{N(2-\rho)}{1-\rho}\lambda_t^2 \sigma^2. \quad (92)$$

Multiplying both sides of equation 92 by $\frac{12}{1-\rho}$ and adding the resulting expression to both sides of equation 90 yield

$$\mathbb{E}\left[\|\boldsymbol{\theta}'_{t+1} - \mathbf{1}_N \otimes \theta^*\|^2\right] + \frac{12}{1-\rho}\mathbb{E}\left[\|\boldsymbol{\theta}'_{t+1} - \mathbf{1}_N \otimes \bar{\theta}'_{t+1}\|^2\right]$$

$$\leq \left(1 - \frac{\mu}{2}\lambda_t + 10H^2\lambda_t^2 + \frac{48H^2(2-\rho)}{(1-\rho)^2}\lambda_t^2\right)\mathbb{E}\left[\|\boldsymbol{\theta}'_t - \mathbf{1}_N \otimes \theta^*\|^2\right] \quad (93)$$

$$+ \left(5 + 2\lambda_t\mu + \frac{12\rho}{1-\rho}\right)\mathbb{E}\left[\|\boldsymbol{\theta}'_t - \mathbf{1}_N \otimes \bar{\theta}'_t\|^2\right] + c_1\lambda_t\kappa_t^2\delta^2 + c_2\lambda_t^2,$$

where $c_1 = N(6\lambda_0 + 4\mu^{-1}) + \frac{36N(2-\rho)\lambda_0}{(1-\rho)^2}$ and $c_2 = 2N\sigma^2 + \frac{12N\sigma^2(2-\rho)}{(1-\rho)^2}$.

Since $\lambda_t$ is a decaying sequence, we can set $\lambda_0$ such that $10H^2\lambda_0 + \frac{48H^2(2-\rho)}{(1-\rho)^2}\lambda_0 \leq \frac{\mu}{4}$, $2\lambda_0\mu < 1$, and $1 - \frac{\mu}{4}\lambda_0 > \frac{1+\rho}{2}$ hold. In this case, equation 93 can be rewritten as follows:

$$\mathbb{E}\left[\|\boldsymbol{\theta}'_{t+1} - \mathbf{1}_N \otimes \theta^*\|^2\right] + \frac{12}{1-\rho}\mathbb{E}\left[\|\boldsymbol{\theta}'_{t+1} - \mathbf{1}_N \otimes \bar{\theta}'_{t+1}\|^2\right]$$

$$\leq \left(1 - \frac{\mu}{4}\lambda_t\right)\left(\mathbb{E}\left[\|\boldsymbol{\theta}'_t - \mathbf{1}_N \otimes \theta^*\|^2\right] + \frac{12}{1-\rho}\mathbb{E}\left[\|\boldsymbol{\theta}'_t - \mathbf{1}_N \otimes \bar{\theta}'_t\|^2\right]\right) + \left(\frac{c_1\delta^2}{\lambda_0} + c_2\right)\lambda_t^2.$$

Combining Lemma 4 and the preceding inequality, we arrive at

$$\sum_{i=1}^N \mathbb{E}[\|\theta'_{i,t} - \theta^*\|^2] \leq C_1(t+1)^{-v} \text{ and } \sum_{i=1}^N \mathbb{E}[\|\theta'_{i,t} - \bar{\theta}'_t\|^2] \leq 12^{-1}C_1(1-\rho)(t+1)^{-v}, \quad (94)$$

where $C_1 = \max\{\mathbb{E}\left[\|\boldsymbol{\theta}'_0 - \mathbf{1}_N \otimes \theta^*\|^2\right] + \frac{12}{1-\rho}\mathbb{E}\left[\|\boldsymbol{\theta}'_0 - \mathbf{1}_N \otimes \bar{\theta}'_0\|^2\right], \frac{4(c_1\delta^2 + c_2\lambda_0)\lambda_0}{\mu\lambda_0 - 4v}\}$ with $c_1$ and $c_2$ given in equation 93.

In addition, according to the definition of $C_1$, we have $C_1 \leq \mathcal{O}\left(\frac{H^2(\sigma^2 + \sigma^2)}{\mu(1-\rho)^2}\right)$, which, combined with equation 94, implies equation 80. $\qquad\square$

## E.2 CONVERGENCE RATE FOR A GENERAL CONVEX GLOBAL OBJECTIVE FUNCTION

In this subsection, we present Lemma 13 to summarize the convergence result of Algorithm 1 in the presence of strategic behavior for a general convex $F(\theta)$, which corresponds to Theorem 1-(ii) in the main text. To prove Lemma 13 (or Theorem 1-(ii)), we introduce an auxiliary lemma (see Lemma 12), which characterizes the consensus error of Algorithm 1 with our payment mechanism under strategic manipulation by agents in a general convex setting.

**Lemma 12.** *Under our decentralized payment mechanism and the conditions in Lemma 1, the consensus error of Algorithm 1 satisfies*

$$\mathbb{E}\left[\|\boldsymbol{\theta}'_t - \mathbf{1}_N \otimes \bar{\theta}'_t\|^2\right]$$

$$\leq \left(\frac{16\mathbb{E}[\|\boldsymbol{\theta}_0 - \mathbf{1}_N \otimes \bar{\theta}_0\|^2]}{e^2(1-\rho)^2\lambda_0^2} + \frac{4N(2-\rho)(2\delta^2 + \sigma^2 + L_f^2 + 2\delta L_f)}{(1-\rho)^2}\right)\lambda_t^2 \triangleq c_3\lambda_t^2. \quad (95)$$

*Furthermore, since the action $(a_{i,t}, b_{i,t}) = (1, 0)$ is a special case of the action $(a_{i,t}, b_{i,t})$ with $a_{i,t} \geq 1$ and $b_{i,t} \in \mathbb{R}$, equation 95 still holds for Algorithm 1 without strategic manipulation.*

*Proof.* According to the definitions $\boldsymbol{\theta}'_t = \text{col}(\theta'_{1,t}, \cdots, \theta'_{N,t})$, $\boldsymbol{m}_t = \text{col}(m_{1,t}, \cdots, m_{N,t})$, and $\bar{m}_t = \frac{1}{N} \sum_{i=1}^N m_{i,t}$ with $m_{i,t} = a_{i,t} g_i(\theta'_{i,t}) + b_{i,t}\xi_{i,t}$, we have

$$\mathbb{E}\left[\|\boldsymbol{\theta}'_{t+1} - \mathbf{1}_N \otimes \bar{\theta}'_{t+1}\|^2\right] = \mathbb{E}\left[\|(W \otimes I_n)\boldsymbol{\theta}'_t - \lambda_t(\boldsymbol{m}_t - \mathbf{1}_N \otimes \bar{m}_t) - \mathbf{1}_N \otimes \bar{\theta}'_t\|^2\right]$$

$$\leq (1 + \tau_1)\mathbb{E}\left[\|(W \otimes I_n)(\boldsymbol{\theta}'_t - \mathbf{1}_N \otimes \bar{\theta}'_t)\|^2\right] + \left(1 + \frac{1}{\tau_1}\right)\lambda_t^2\mathbb{E}\left[\|\boldsymbol{m}_t\|^2\right],$$

for any $\tau_1 > 0$, where we have used the relationship $\mathbb{E}[\|\boldsymbol{m}_t - \mathbf{1}_N \otimes \bar{m}_t\|^2] \leq \mathbb{E}[\|\boldsymbol{m}_t\|^2]$.

By using the inequality $(1 + 1 - \rho)\rho = 1 - (1 - \rho)^2 < 1$ (due to $\rho < 1$) and setting $\tau_1 = 1 - \rho$, we obtain

$$\mathbb{E}\left[\|\boldsymbol{\theta}'_{t+1} - \mathbf{1}_N \otimes \bar{\theta}'_{t+1}\|^2\right] \leq (1 + (1 - \rho))\rho^2 \mathbb{E}\left[\|\boldsymbol{\theta}'_t - \mathbf{1}_N \otimes \bar{\theta}'_t\|^2\right] + \frac{2 - \rho}{1 - \rho}\lambda_t^2 \mathbb{E}\left[\|\boldsymbol{m}_t\|^2\right]$$

$$\leq \rho\mathbb{E}\left[\|\boldsymbol{\theta}'_t - \mathbf{1}_N \otimes \bar{\theta}'_t\|^2\right] + \frac{2 - \rho}{1 - \rho}\lambda_t^2 \mathbb{E}\left[\|\boldsymbol{m}_t\|^2\right]. \tag{96}$$

Assumption 1 and equation 51 imply that the last term on the right hand side of equation 96 satisfies

$$\mathbb{E}\left[\|\boldsymbol{m}_t\|^2\right] = \sum_{i=1}^N \mathbb{E}\left[\|a_{i,t} g_i(\theta'_{i,t}) - g_i(\theta'_{i,t})\|^2\right] + \sum_{i=1}^N \mathbb{E}\left[\|g_i(\theta'_{i,t}) - \nabla f_i(\theta'_{i,t})\|^2\right]$$

$$+ \sum_{i=1}^N \mathbb{E}\left[\|\nabla f_i(\theta'_{i,t})\|^2\right] + \sum_{i=1}^N \mathbb{E}\left[\|b_{i,t}\xi_{i,t}\|^2\right] + \sum_{i=1}^N 2\mathbb{E}\left[\langle a_{i,t} g_i(\theta'_{i,t}) - g_i(\theta'_{i,t}), \nabla f_i(\theta'_{i,t})\rangle\right]$$

$$\leq N(\kappa_t^2\delta^2 + \sigma^2 + L_f^2 + \kappa_t^2\delta^2 + 2\kappa_t\delta L_f) \leq N(2\delta^2 + \sigma^2 + L_f^2 + 2\delta L_f), \tag{97}$$

where we have used the $L_f$-Lipschitz continuity of $f_i(\theta)$ in the first inequality and the fact $\kappa_t \leq \kappa_0 = 1$ in the last inequality.

Substituting equation 97 into equation 96, we obtain

$$\mathbb{E}\left[\|\boldsymbol{\theta}'_{t+1} - \mathbf{1}_N \otimes \bar{\theta}'_{t+1}\|^2\right] \leq \rho\mathbb{E}\left[\|\boldsymbol{\theta}'_t - \mathbf{1}_N \otimes \bar{\theta}'_t\|^2\right]$$

$$+ \frac{4N(2 - \rho)(2\delta^2 + \sigma^2 + L_f^2 + 2\delta L_f)}{1 - \rho}\lambda_t^2. \tag{98}$$

By using the relationship $\mathbb{E}[\|\boldsymbol{\theta}'_0 - \mathbf{1}_N \otimes \bar{\theta}'_0\|^2] = \mathbb{E}[\|\boldsymbol{\theta}_0 - \mathbf{1}_N \otimes \bar{\theta}_0\|^2]$ and iterating equation 98 from 0 to $t$, we arrive at

$$\mathbb{E}\left[\|\boldsymbol{\theta}'_t - \mathbf{1}_N \otimes \bar{\theta}'_t\|^2\right] \leq \rho^t \mathbb{E}[\|\boldsymbol{\theta}_0 - \mathbf{1}_N \otimes \bar{\theta}_0\|^2]$$

$$+ \frac{4N(2 - \rho)(2\delta^2 + \sigma^2 + L_f^2 + 2\delta L_f)}{(1 - \rho)^2}\lambda_t^2. \tag{99}$$

Lemma 3 proves $\rho^t \leq \frac{16}{e^2(\ln(\rho))^2(t+1)^2} \leq \frac{16\lambda_t^2}{e^2(\ln(\rho))^2\lambda_0^2}$, which implies that for any $\rho \in (0, 1)$, we have $-\ln(\rho) \geq 1 - \rho$. Hence, equation 99 implies equation 95 in Lemma 12. $\qquad\square$

**Lemma 13.** *We denote $\theta^*$ as a solution to the problem in equation 1. Under our fully decentralized payment mechanism (Mechanism 1) and the conditions in Lemma 1, for any $i \in [N]$ and $T \geq 0$, if the global objective function $F(\theta)$ is convex, then the following inequalities hold for Algorithm 1 in the presence of strategic behaviors:*

$$\mathbb{E}\left[\|\theta'_{i,T} - \bar{\theta}'_T\|^2\right] \leq \mathcal{O}\left(\frac{(\sigma^2 + L_f^2 + \delta^2)}{(1 - \rho)^2(T + 1)^{2v}}\right);$$

$$\frac{1}{T + 1}\mathbb{E}[F(\bar{\theta}'_t)] - F(\theta^*) \leq \mathcal{O}\left(\frac{H^2(\sigma^2 + L_f^2 + \delta^2)}{(1 - \rho)^2(T + 1)^{1-v}}\right); \tag{100}$$

$$\frac{1}{T + 1}\sum_{t=0}^T \mathbb{E}[F(\theta'_{i,t})] - F(\theta^*) \leq \mathcal{O}\left(\frac{H^2(\sigma^2 + L_f^2 + \delta^2)}{(1 - \rho)^2(T + 1)^{1-v}}\right).$$

*Proof.* 1) The first inequality in equation 100 follows naturally from equation 95 in Lemma 12.

2) We proceed to prove the second inequality in equation 100. By using the dynamics of $\theta_{i,t}$ in Algorithm 1 and the definitions $\boldsymbol{\theta}_t' = \text{col}(\theta_{1,t}', \cdots, \theta_{N,t}')$ and $\boldsymbol{m}_t = \text{col}(m_{1,t}, \cdots, m_{N,t})$ with $m_{i,t} = a_{i,t} g_i(\theta_{i,t}') + b_{i,t} \xi_{i,t}$, we obtain

$$
\begin{aligned}
\mathbb{E}\left[\|\boldsymbol{\theta}_{t+1}' - \mathbf{1}_N \otimes \theta^*\|^2\right] &\leq \mathbb{E}\left[\|(W \otimes I_n - I_{Nn})\boldsymbol{\theta}_t' - \lambda_t \boldsymbol{m}_t\|^2\right] + \mathbb{E}\left[\|\boldsymbol{\theta}_t' - \mathbf{1}_N \otimes \theta^*\|^2\right] \\
&+ 2\mathbb{E}\left[\langle (W \otimes I_n - I_{Nn})\boldsymbol{\theta}_t' - \lambda_t \boldsymbol{m}_t, \boldsymbol{\theta}_t' - \mathbf{1}_N \otimes \theta^*\rangle\right].
\end{aligned}
\tag{101}
$$

Using the relationship $W\mathbf{1}_N^\top = \mathbf{1}_N$ from Assumption 2, the first term on the right hand side of equation 101 satisfies

$$
\begin{aligned}
&\mathbb{E}\left[\|(W \otimes I_n - I_{Nn})\boldsymbol{\theta}_t' - \lambda_t \boldsymbol{m}_t\|^2\right] \\
&= \mathbb{E}\left[\|(W \otimes I_n - I_{Nn})(\boldsymbol{\theta}_t' - \mathbf{1}_N \otimes \bar{\theta}_t') - \lambda_t \boldsymbol{m}_t\|^2\right] \\
&\leq 2\mathbb{E}\left[\|(W \otimes I_n - I_{Nn})(\boldsymbol{\theta}_t' - \mathbf{1}_N \otimes \bar{\theta}_t')\|^2\right] + 2\lambda_t^2 \mathbb{E}\left[\|\boldsymbol{m}_t\|^2\right].
\end{aligned}
\tag{102}
$$

Similarly, by using the relationship $W\mathbf{1}_N^\top = \mathbf{1}_N$ from Assumption 2, the last term on the right hand side of equation 101 satisfies

$$
\begin{aligned}
&2\mathbb{E}\left[\langle (W \otimes I_n - I_{Nn})\boldsymbol{\theta}_t' - \lambda_t \boldsymbol{m}_t, \boldsymbol{\theta}_t' - \mathbf{1}_N \otimes \theta^*\rangle\right] \\
&= 2\mathbb{E}\left[(\boldsymbol{\theta}_t' - \mathbf{1}_N \otimes \theta^*)^\top (W \otimes I_n - I_{Nn})(\boldsymbol{\theta}_t' - \mathbf{1}_N \otimes \theta^*)\right] \\
&- 2\mathbb{E}\left[\langle \lambda_t \boldsymbol{m}_t, \boldsymbol{\theta}_t' - \mathbf{1}_N \otimes \theta^*\rangle\right].
\end{aligned}
\tag{103}
$$

Substituting equation 102 and equation 103 into equation 101, and omitting the negative term $2\mathbb{E}[(\boldsymbol{\theta}_t' - \mathbf{1}_N \otimes \theta^*)^\top (W \otimes I_n - I_{Nn})(\boldsymbol{\theta}_t' - \mathbf{1}_N \otimes \theta^*)]$ from equation 103, we obtain

$$
\begin{aligned}
\mathbb{E}\left[\|\boldsymbol{\theta}_{t+1}' - \mathbf{1}_N \otimes \theta^*\|^2\right] &\leq \mathbb{E}\left[\|\boldsymbol{\theta}_t' - \mathbf{1}_N \otimes \theta^*\|^2\right] \\
&+ 2\mathbb{E}\left[\|(W \otimes I_n - I_{Nn})(\boldsymbol{\theta}_t' - \mathbf{1}_N \otimes \bar{\theta}_t')\|^2\right] + 2\lambda_t^2 \mathbb{E}\left[\|\boldsymbol{m}_t\|^2\right] \\
&- 2\mathbb{E}\left[\langle \lambda_t \boldsymbol{m}_t, \boldsymbol{\theta}_t' - \mathbf{1}_N \otimes \theta^*\rangle\right].
\end{aligned}
\tag{104}
$$

The last term on the right hand side of equation 104 satisfies

$$
\begin{aligned}
&-2\mathbb{E}\left[\langle \lambda_t \boldsymbol{m}_t, \boldsymbol{\theta}_t' - \mathbf{1}_N \otimes \theta^*\rangle\right] \\
&= -2\sum_{i=1}^N \mathbb{E}\left[\langle \lambda_t g_i(\theta_{i,t}'), \theta_{i,t}' - \theta^*\rangle\right] + 2\sum_{i=1}^N \mathbb{E}\left[\langle \lambda_t g_i(\theta_{i,t}') - \lambda_t m_{i,t}, \theta_{i,t}' - \theta^*\rangle\right].
\end{aligned}
\tag{105}
$$

Using the relation $\mathbb{E}[g_i(\theta)] = \nabla f_i(\theta)$, the first term on the right hand side of equation 105 satisfies

$$
\begin{aligned}
&-2\sum_{i=1}^N \mathbb{E}\left[\langle \lambda_t g_i(\theta_{i,t}'), \theta_{i,t}' - \theta^*\rangle\right] = -2\lambda_t \sum_{i=1}^N \mathbb{E}\left[\langle g_i(\theta_{i,t}') - \nabla f_i(\theta_{i,t}'), \theta_{i,t}' - \theta^*\rangle\right] \\
&-2\lambda_t \sum_{i=1}^N \mathbb{E}\left[\langle \nabla f_i(\theta_{i,t}') - \nabla f_i(\bar{\theta}_t'), \theta_{i,t}' - \theta^*\rangle\right] - 2\lambda_t \sum_{i=1}^N \mathbb{E}\left[\langle \nabla f_i(\bar{\theta}_t'), \theta_{i,t}' - \theta^*\rangle\right] \\
&= -2\lambda_t \sum_{i=1}^N \mathbb{E}\left[\langle \nabla f_i(\theta_{i,t}') - \nabla f_i(\bar{\theta}_t'), \theta_{i,t}' - \theta^*\rangle\right] - 2\lambda_t \sum_{i=1}^N \mathbb{E}\left[\langle \nabla f_i(\bar{\theta}_t'), \theta_{i,t}' - \theta^*\rangle\right].
\end{aligned}
\tag{106}
$$

By introducing an auxiliary sequence $\beta_t$ that satisfies $\beta_t = \frac{1}{(t+1)^u}$ with $1 < u < 2v$, the first term on the right hand side of equation 106 satisfies

$$
\begin{aligned}
&-2\lambda_t \sum_{i=1}^N \mathbb{E}\left[\langle \nabla f_i(\theta_{i,t}') - \nabla f_i(\bar{\theta}_t'), \theta_{i,t}' - \theta^*\rangle\right] \\
&\leq \frac{\lambda_t^2}{\beta_t} H^2 \mathbb{E}\left[\|\boldsymbol{\theta}_t' - \mathbf{1}_N \otimes \bar{\theta}_t'\|^2\right] + \beta_t \mathbb{E}\left[\|\boldsymbol{\theta}_t' - \mathbf{1}_N \otimes \theta^*\|^2\right].
\end{aligned}
\tag{107}
$$

By using the convexity of $F(\theta)$ and the relations $\sum_{i=1}^N \nabla f_i(\bar{\theta}_t') = N\nabla F(\bar{\theta}_t')$ and $\sum_{i=1}^N \theta_{i,t}' = N\bar{\theta}_t'$, the second term on the right hand side of equation 106 satisfies

$$
\begin{aligned}
&-2\lambda_t \sum_{i=1}^N \mathbb{E}\left[\langle \nabla f_i(\bar{\theta}_t'), \theta_{i,t}' - \theta^*\rangle\right] \\
&= -2\lambda_t N \mathbb{E}\left[\langle \nabla F(\bar{\theta}_t'), \bar{\theta}_t' - \theta^*\rangle\right] \leq -2N\lambda_t \mathbb{E}\left[F(\bar{\theta}_t') - F(\theta^*)\right].
\end{aligned}
\tag{108}
$$

Substituting equation 107 and equation 108 into equation 106, the first term on the right hand side of equation 105 satisfies

$$
\begin{aligned}
-2\sum_{i=1}^{N} \mathbb{E}\left[\langle \lambda_t g_i(\theta'_{i,t}), \theta'_{i,t} - \theta^* \rangle\right] &\leq -2N\lambda_t \mathbb{E}\left[F(\bar{\theta}'_t) - F(\theta^*)\right] \\
&+ \frac{\lambda_t^2}{\beta_t}H^2 \mathbb{E}\left[\|\boldsymbol{\theta}'_t - \mathbf{1}_N \otimes \bar{\theta}'_t\|^2\right] + \beta_t \mathbb{E}\left[\|\boldsymbol{\theta}'_t - \mathbf{1}_N \otimes \theta^*\|^2\right].
\end{aligned}
\tag{109}
$$

We proceed to characterize the second term on the right hand side of equation 105. By using the Young's inequality, we have

$$
\begin{aligned}
2\sum_{i=1}^{N} &\mathbb{E}\left[\langle \lambda_t g_i(\theta'_{i,t}) - \lambda_t m_{i,t}, \theta'_{i,t} - \theta^* \rangle\right] \\
&= 2\sum_{i=1}^{N} \mathbb{E}\left[\langle \lambda_t g_i(\theta'_{i,t}) - \lambda_t(a_{i,t}g_i(\theta'_{i,t}) + b_{i,t}\xi_{i,t}), \theta'_{i,t} - \theta^* \rangle\right] \\
&\leq \frac{\lambda_t^2}{\beta_t}\sum_{i=1}^{N} \mathbb{E}\left[\|a_{i,t}g_i(\theta'_{i,t}) - g_i(\theta'_{i,t})\|^2\right] + \beta_t \mathbb{E}\left[\|\boldsymbol{\theta}'_t - \mathbf{1}_N \otimes \theta^*\|^2\right].
\end{aligned}
\tag{110}
$$

Substituting equation 109 and equation 110 into equation 105 and then substituting equation 105 into equation 104, we arrive at

$$
\begin{aligned}
\mathbb{E}\left[\|\boldsymbol{\theta}'_{t+1} - \mathbf{1}_N \otimes \theta^*\|^2\right] &\leq -2N\lambda_t \mathbb{E}\left[F(\bar{\theta}'_t) - F(\theta^*)\right] \\
&+ (1 + 2\beta_t)\mathbb{E}\left[\|\boldsymbol{\theta}'_t - \mathbf{1}_N \otimes \theta^*\|^2\right] + \Phi_t,
\end{aligned}
\tag{111}
$$

where the term $\Phi_t$ is given by

$$
\begin{aligned}
\Phi_t = \frac{\lambda_t^2}{\beta_t}\sum_{i=1}^{N} &\mathbb{E}\left[\|a_{i,t}g_i(\theta'_{i,t}) - g_i(\theta'_{i,t})\|^2\right] + 2\lambda_t^2 \mathbb{E}\left[\|\boldsymbol{m}_t\|^2\right] \\
&+ \left(2 + \frac{\lambda_t^2}{\beta_t}H^2\right)\mathbb{E}\left[\|\boldsymbol{\theta}'_t - \mathbf{1}_N \otimes \bar{\theta}'_t\|^2\right].
\end{aligned}
\tag{112}
$$

Since the relationship $F(\bar{\theta}'_t) \geq F(\theta^*)$ always holds, we omit the negative term $-2N\lambda_t \mathbb{E}[F(\bar{\theta}'_t) - F(\theta^*)]$ in equation 111 to obtain

$$
\begin{aligned}
\mathbb{E}\left[\|\boldsymbol{\theta}'_{t+1} - \mathbf{1}_N \otimes \theta^*\|^2\right] &\leq (1 + 2\beta_t)\mathbb{E}\left[\|\boldsymbol{\theta}'_t - \mathbf{1}_N \otimes \theta^*\|^2\right] + \Phi_t, \\
&\leq \left(\prod_{k=0}^{t}(1 + 2\beta_k)\right)\left(\mathbb{E}[\|\boldsymbol{\theta}'_0 - \mathbf{1}_N \otimes \theta^*\|^2] + \sum_{k=0}^{t}\Phi_k\right).
\end{aligned}
\tag{113}
$$

By using the relationship $\ln(1 + x) \leq x$ for any $x > 0$ and the definition $\beta_k = \frac{1}{(k+1)^u}$ with $1 < u < 2v$, we have

$$
\ln\left(\prod_{k=0}^{t}(1 + 2\beta_k)\right) = \sum_{k=0}^{t}\ln(1 + 2\beta_k) \leq 2\beta_0 + \int_{1}^{\infty}\frac{1}{x^u}dx \leq \frac{2\beta_0(u-1) + 1}{u - 1},
\tag{114}
$$

which implies $\prod_{k=0}^{t}(1 + 2\beta_k) \leq e^{\frac{2\beta_0(u-1)+1}{u-1}}$. Then, equation 113 can be rewritten as follows:

$$
\mathbb{E}\left[\|\boldsymbol{\theta}'_{t+1} - \mathbf{1}_N \otimes \theta^*\|^2\right] \leq e^{\frac{2\beta_0(u-1)+1}{u-1}}\left(\mathbb{E}\left[\|\boldsymbol{\theta}'_0 - \mathbf{1}_N \otimes \theta^*\|^2\right] + \sum_{k=0}^{t}\Phi_k\right).
\tag{115}
$$

Next, we estimate an upper bound on $\sum_{k=0}^{t}\Phi_k$. By substituting equation 95 and equation 97 into equation 112 and using the relationship $\mathbb{E}[\|a_{i,k}g_i(\theta'_{i,k}) - g_i(\theta'_{i,k})\|^2] \leq \kappa_k^2\delta^2$, we obtain

$$
\begin{aligned}
\sum_{k=0}^{t}\Phi_k &\leq \sum_{k=0}^{t}\left(\frac{N\lambda_k^2\kappa_k^2\delta^2}{\beta_k} + 2N(2\delta^2 + \sigma^2 + L_f^2 + 2\delta L_f)\lambda_k^2 + \left(2 + \frac{\lambda_k^2}{\beta_k}H^2\right)c_3\lambda_k^2\right) \\
&\leq \sum_{k=0}^{t}\left(\frac{N\lambda_0^2\delta^2}{(k+1)^{2v+2r-u}} + \frac{c_4}{(k+1)^{2v}}\right),
\end{aligned}
\tag{116}
$$

where the constant $c_4$ is given by $c_4 = 4N\delta^2 + 2N\sigma^2 + 2NL_f^2 + 4N\delta L_f + c_3(2 + \lambda_0^2 H^2)$ with $c_3$ given in equation 95. Here, we have used the relations $1 < u < 2v$ and $\frac{\lambda_k^2}{\beta_k} = \frac{\lambda_0^2}{(k+1)^{2v-u}} \leq \lambda_0^2$.

By using the following inequality:

$$\sum_{k=0}^{t} \frac{1}{(k+1)^u} = 1 + \sum_{k=1}^{t} \frac{1}{(k+1)^u} \leq 1 + \int_{k=1}^{\infty} \frac{1}{x^u} dx \leq \frac{u}{u-1}, \tag{117}$$

which is true for any $u > 1$, we can rewrite equation 116 as follows:

$$\sum_{k=0}^{t} \Phi_k \leq \frac{(2v + 2r - u)N\lambda_0^2\delta^2}{2v + 2r - u - 1} + \frac{2vc_4}{2v-1} \triangleq c_5, \tag{118}$$

where the constant $c_4$ is given in equation 116.

Substituting equation 118 into equation 115, we can arrive at

$$\mathbb{E}\left[\|\boldsymbol{\theta}_{t+1}' - \mathbf{1}_N \otimes \theta^*\|^2\right] \leq e^{\frac{2\beta_0(u-1)+1}{u-1}} \left(\mathbb{E}\left[\|\boldsymbol{\theta}_0' - \mathbf{1}_N \otimes \theta^*\|^2\right] + c_5\right). \tag{119}$$

We proceed to sum both sides of equation 111 from 0 to $T$:

$$\sum_{t=0}^{T} 2N\lambda_t \mathbb{E}\left[F(\bar{\theta}_t') - F(\theta^*)\right] \leq - \sum_{t=0}^{T} \mathbb{E}\left[\|\boldsymbol{\theta}_{t+1}' - \mathbf{1}_N \otimes \theta^*\|^2\right]$$
$$+ \sum_{t=0}^{T}(1 + 2\beta_t)\mathbb{E}\left[\|\boldsymbol{\theta}_t' - \mathbf{1}_N \otimes \theta^*\|^2\right] + \sum_{t=0}^{T} \Phi_t. \tag{120}$$

The first and second terms on the right hand side of equation 120 can be simplified as follows:

$$\sum_{t=0}^{T}(1 + 2\beta_t)\mathbb{E}\left[\|\boldsymbol{\theta}_t' - \mathbf{1}_N \otimes \theta^*\|^2\right] - \sum_{t=0}^{T} \mathbb{E}\left[\|\boldsymbol{\theta}_{t+1}' - \mathbf{1}_N \otimes \theta^*\|^2\right]$$

$$\leq 2\beta_0 \mathbb{E}\left[\|\boldsymbol{\theta}_0' - \mathbf{1}_N \otimes \theta^*\|^2\right] + \sum_{t=1}^{T} 2\beta_t \mathbb{E}\left[\|\boldsymbol{\theta}_t' - \mathbf{1}_N \otimes \theta^*\|^2\right] + \mathbb{E}\left[\|\boldsymbol{\theta}_0' - \mathbf{1}_N \otimes \theta^*\|^2\right]$$

$$- \mathbb{E}\left[\|\boldsymbol{\theta}_{t+1}' - \mathbf{1}_N \otimes \theta^*\|^2\right]$$

$$\leq \sum_{t=1}^{T} \frac{2}{(t+1)^u} \left(e^{\frac{2\beta_0(u-1)+1}{u-1}} \left(\mathbb{E}[\|\boldsymbol{\theta}_0' - \mathbf{1}_N \otimes \theta^*\|^2] + c_4\right)\right) + (1 + 2\beta_0)\mathbb{E}\left[\|\boldsymbol{\theta}_0' - \mathbf{1}_N \otimes \theta^*\|^2\right]$$

$$\leq \left(\frac{2ue^{\frac{2\beta_0(u-1)+1}{u-1}}}{u-1} + (1 + 2\beta_0)\right) \mathbb{E}\left[\|\boldsymbol{\theta}_0' - \mathbf{1}_N \otimes \theta^*\|^2\right] + \frac{2c_5u}{u-1} \triangleq c_6, \tag{121}$$

where we have used equation 119 in the second inequality and equation 117 in the last inequality.

Substituting equation 118 and equation 121 into equation 120, and using $\lambda_T \leq \lambda_t$ for any $t \in [0, T]$, we have $\sum_{t=0}^{T} 2N\lambda_t \mathbb{E}[F(\bar{\theta}_t') - F(\theta^*)] \leq c_5 + c_6$, which further implies

$$\frac{1}{T+1} \sum_{t=0}^{T} \mathbb{E}\left[F(\bar{\theta}_t') - F(\theta^*)\right] \leq \frac{c_5 + c_6}{2\lambda_0 N(T+1)^{1-v}} \leq \frac{C_2}{N(T+1)^{1-v}}, \tag{122}$$

where $C_2 = \frac{c_5+c_6}{2\lambda_0}$ with $c_5$ given in equation 118 and $c_6$ given in equation 121.

According to the definition of $C_2$ given in equation 122, we have $\frac{C_2}{N} \leq \mathcal{O}\left(\frac{H^2(\sigma^2 + L_f^2 + \delta^2)}{(1-\rho)^2}\right)$, which, combined with equation 122, implies the second inequality in equation 100.

3) We now prove the third inequality in equation 100.

Assumption 1 implies $\mathbb{E}[F(\theta_{i,t}') - F(\bar{\theta}_t')] \leq L_f \mathbb{E}[\|\theta_{i,t}' - \bar{\theta}_t'\|]$. By using equation 95, we have

$$\mathbb{E}\left[F(\theta_{i,t}') - F(\bar{\theta}_t')\right] \leq \frac{L_f\sqrt{c_3}\lambda_0}{(t+1)^v}, \tag{123}$$

where the constant $c_3$ is given in equation 95.

Since $\sum_{t=0}^{T} \frac{1}{(t+1)^p} \leq \int_{x=0}^{T+1} \frac{1}{x^p} dx \leq \frac{(T+1)^{1-p}}{1-p}$ always holds for any $p \in (0,1)$, we arrive at

$$\frac{1}{T+1} \sum_{t=0}^{T} \mathbb{E}\left[F(\theta'_{i,t}) - F(\bar{\theta}'_t)\right] \leq \frac{L_f \sqrt{c_3} \lambda_0}{(1-v)(T+1)^v}. \tag{124}$$

According to the condition $\frac{1}{2} \leq v < 1$ given in the lemma statement, we have $1 - v \leq v$. Hence, by combining equation 122 and equation 124, we arrive at

$$\frac{1}{T+1} \sum_{i=1}^{N} \sum_{t=0}^{T} \mathbb{E}\left[F(\theta'_{i,t}) - F(\theta^*)\right] \leq \frac{N L_f \sqrt{c_3} \lambda_0}{(1-v)(T+1)^v} + \frac{C_2}{(T+1)^{1-v}} = C_3(T+1)^{-v}, \tag{125}$$

where the constant $C_3$ is given by $C_3 = \frac{N L_f \sqrt{c_3} \lambda_0}{1-v} + C_2$ with $c_3$ given in equation 95 and $C_2$ given in equation 122.

According to the definition of $C_3$ given in equation 125, we have $C_3 \leq \mathcal{O}\left(\frac{H^2(\sigma^2 + L_f^2 + \delta^2)}{(1-\rho)^2}\right)$, which, combined with equation 125, implies the third inequality in equation 100 and equation 5 in Theorem 1-(ii). $\square$

### E.3 COMPUTATIONAL COMPLEXITY OF ALGORITHM 1 UNDER OUR PAYMENT MECHANISM

Building on the convergence results established in Theorem 1, we quantify the computational complexities of Algorithm 1 with our payment mechanism. The definition of a $\varrho$-solution is given in Definition 5 and the computational-complexity result is summarized in the following Corollary 2.

**Corollary 2** (Computational complexity). *(i) For a strongly convex $F(\theta)$, if we choose $T = \mathcal{O}(\varrho^{-\frac{1}{v}})$, then the computational complexity of Algorithm 1 under our payment mechanism is $\mathcal{O}(n\varrho^{-\frac{1}{v}})$ in finding a $\varrho$-solution. For example, setting $v = 0.5 + \varsigma$ with an arbitrarily small $\varsigma > 0$ yields a convergence rate arbitrarily close to $\mathcal{O}(1/\sqrt{T})$ and a computational complexity arbitrarily close to $\mathcal{O}(n\varrho^{-2})$.*

*(ii) For a general convex $F(\theta)$, if we choose $T = \mathcal{O}(\varrho^{-\frac{1}{1-v}})$, then the computational complexity of Algorithm 1 under our payment mechanism is $\mathcal{O}(n\varrho^{-\frac{1}{1-v}})$ in finding a $\varrho$-solution. For example, setting $v = 0.5 + \varsigma$ with an arbitrarily small $\varsigma > 0$ yields a convergence rate arbitrarily close to $\mathcal{O}(1/\sqrt{T})$ and a computational complexity arbitrarily close to $\mathcal{O}(n\varrho^{-2})$.*

*Proof.* (i) For a strongly convex $F(\theta)$, the convergence rate of Algorithm 1 under our payment mechanism is $\mathcal{O}\left(\frac{H^2(\delta^2 + \sigma^2)}{\mu(1-\rho)^2(T+1)^v}\right)$ based on equation 4. Therefore, setting $\frac{H^2(\delta^2 + \sigma^2)}{\mu(1-\rho)^2} T^{-v} = \varrho$ leads to an iteration complexity of $\mathcal{O}\left(\left(\frac{H^2(\delta^2 + \sigma^2)}{\mu(1-\rho)^2 \varrho}\right)^{\frac{1}{v}}\right)$ in finding a $\varrho$-solution. Furthermore, since the per-iteration complexity of Algorithm 1 and our payment mechanism is of order $n$, the computational complexity of Algorithm 1 under our payment mechanism is $\mathcal{O}\left(n\left(\frac{H^2(\delta^2 + \sigma^2)}{\mu(1-\rho)^2 \varrho}\right)^{\frac{1}{v}}\right)$ in finding a $\varrho$-solution.

According to the condition $\frac{1}{2} < v < \frac{2}{3}$ given in Lemma 1, we can choose $v = 0.5 + \varsigma$ with an arbitrarily small $\varsigma > 0$ to yield a convergence rate arbitrarily close to $\mathcal{O}(1/\sqrt{T})$ and a computational complexity arbitrarily close to $\mathcal{O}\left(n\left(\frac{H^2(\delta^2 + \sigma^2)}{\mu(1-\rho)^2 \varrho}\right)^2\right)$.

(ii) Similarly, for a general convex $F(\theta)$, the convergence rate of Algorithm 1 under our payment mechanism is $\mathcal{O}\left(\frac{H^2(\sigma^2 + L_f^2 + \delta^2)}{(1-\rho)^2(T+1)^{1-v}}\right)$ based on equation 5. Therefore, the computational complexity is $\mathcal{O}\left(n\left(\frac{H^2(\delta^2 + L_f^2 + \sigma^2)}{(1-\rho)^2 \varrho}\right)^{\frac{1}{1-v}}\right)$ in finding a $\varrho$-solution. Recalling the conditions $\frac{1}{2} < v < \frac{2}{3}$ given in Lemma 1, we can select $v = 0.5 + \varsigma$ with an arbitrarily small $\varsigma > 0$ to yield $1 - v = 0.5 - \varsigma$, which implies a convergence rate arbitrarily close to $\mathcal{O}(1/\sqrt{T})$ and a computational complexity arbitrarily close to $\mathcal{O}\left(n\left(\frac{H^2(\delta^2 + L_f^2 + \sigma^2)}{(1-\rho)^2 \varrho}\right)^2\right)$. $\square$

It is worth noting that our Mechanism 1 does not require additional communication compared with the classic decentralized SGD algorithm and only involve one-dimensional scalar variables (i.e., the norm of model-parameter differences between agents). Therefore, its communication and computational complexities are negligible.

Corollary 2 proves that in general convex settings, Algorithm 1 with our decentralized payment mechanism achieves a convergence rate of $\mathcal{O}(1/\sqrt{T})$ and a computational complexity of $\mathcal{O}(n\varrho^{-2})$ similar to those attained by the standard decentralized SGD algorithm in Koloskova et al. (2020c), even under the constraint of truthfulness. Although the results of convergence rate and computational complexity in Corollary 2 are inferior to those achieved by Koloskova et al. (2020c) in strongly convex settings, we would like to emphasize that this degradation is a comparatively mild cost for ensuring both accurate convergence and $\varepsilon$-incentive compatibility (truthfulness) in fully decentralized optimization and learning. This is different from existing truthfulness results in, e.g., Han et al. (2015); Hale & Egerstedt (2015); Dorner et al. (2023); Chakarov et al. (2024a;b); Chen et al. (2025), all of which sacrifice convergence accuracy to guarantee truthfulness.

### E.4 DIMINISHING PAYMENT GUARANTEE

In this section, we prove that our payment mechanism can ensure $\lim_{t\to\infty}[P_{i,t}] = 0$. This guarantees that no payment is required from agent $i$ when it behaves truthfully in an infinite time horizon.

**Corollary 3.** *Under the conditions in Lemma 1, we have $\lim_{t\to\infty} \mathbb{E}[P_{i,t}] = 0$.*

*Proof.* According to our decentralized payment mechanism in Mechanism 1, we have

$$P_{i,t}(\alpha_i, \alpha_{-i}) = C_t \sum_{j\in\mathcal{N}_i} (\|\theta'_{i,t+1} - 2\theta'_{i,t} + \theta'_{i,t-1}\|^2 - \|\theta'_{j,t+1} - 2\theta'_{j,t} + \theta'_{j,t-1}\|^2). \tag{126}$$

The first term on the right hand side of equation 126 satisfies

$$C_t\mathbb{E}[\|\theta'_{i,t+1} - 2\theta'_{i,t} + \theta'_{i,t-1}\|^2] \le 2C_t\mathbb{E}[\|\theta'_{i,t+1} - \theta'_{i,t}\|^2] + 2C_t\mathbb{E}[\|\theta'_{i,t} - \theta'_{i,t-1}\|^2]. \tag{127}$$

According to the dynamics of $\theta_{i,t}$ in Algorithm 1, the first term on the right hand side of equation 127 satisfies

$$C_t\mathbb{E}[\|\theta'_{i,t+1} - \theta'_{i,t}\|^2] = C_t\mathbb{E}\left[\left\|\sum_{j\in\mathcal{N}_i\cup\{i\}} w_{ij}(\theta'_{j,t} - \bar{\theta}'_t - (\theta'_{i,t} - \bar{\theta}'_t)) - \lambda_t g_i(\theta'_{i,t})\right\|^2\right]$$
$$\le 2C_t\mathbb{E}[\|\boldsymbol{\theta}'_t - \mathbf{1}_N \otimes \bar{\theta}'_t\|^2] + 2C_t\lambda_t^2\mathbb{E}[\|g_i(\theta'_{i,t}) - \nabla f_i(\theta'_{i,t}) + \nabla f_i(\theta'_{i,t})\|^2]. \tag{128}$$

When $f_i(x)$ is general convex, the $L_f$-Lipschitz continuity of $f_i(\theta)$ implies

$$C_t\mathbb{E}[\|\theta'_{i,t+1} - \theta'_{i,t}\|^2] \le 2C_t\mathbb{E}[\|\boldsymbol{\theta}'_t - \mathbf{1}_N \otimes \bar{\theta}'_t\|^2] + 2C_t\lambda_t^2(L_f^2 + \sigma^2)$$
$$\le \frac{C_0(2c_3 + 2(L_f^2 + \sigma^2))\lambda_0^2}{\delta^2(t+1)^{2r}}, \tag{129}$$

where we have used equation 95 and the definition $C_t = \frac{C_0\kappa_t^2}{\delta^2(t+1)^{-2v}}$ in the last inequality.

When $f_i(x)$ is $\mu$-strongly convex, we have

$$C_t\mathbb{E}[\|\theta_{i,t+1} - \theta_{i,t}\|^2] \le 2C_t\left(\mathbb{E}[\|\boldsymbol{\theta}'_t - \mathbf{1}_N \otimes \bar{\theta}'_t\|^2] + \lambda_t^2\mathbb{E}[\|\nabla f_i(\theta'_{i,t}) - \nabla f_i(\theta_i^*)\|^2] + \sigma^2\lambda_t^2\right)$$
$$\le 2C_t\left(\mathbb{E}[\|\boldsymbol{\theta}'_t - \mathbf{1}_N \otimes \bar{\theta}'_t\|^2] + \lambda_t^2 H^2\mathbb{E}[\|\theta'_{i,t} - \theta^* + \theta^* - \theta_i^*\|^2] + \sigma^2\lambda_t^2\right)$$
$$\le 2C_t\mathbb{E}[\|\boldsymbol{\theta}'_t - \mathbf{1}_N \otimes \bar{\theta}'_t\|^2] + \frac{4C_0C_1H^2\lambda_0^2}{\delta^2(t+1)^{v+2r}} + \frac{4\lambda_0^2C_0H^2\mathbb{E}[\|\theta^* - \theta_i^*\|^2]}{\delta^2(t+1)^{2r}} + \frac{2C_0\sigma^2\lambda_0^2}{\delta^2(t+1)^{2r}}, \tag{130}$$

where we have used equation 94 and the definition $C_t = \frac{C_0\kappa_t^2}{\delta^2(t+1)^{-2v}}$ in the last inequality.

By substituting equation 94 into equation 92, we obtain

$$
\mathbb{E}\left[\|\boldsymbol{\theta}'_{t+1} - \mathbf{1}_N \otimes \bar{\theta}'_{t+1}\|^2\right] \leq (1 - (1-\rho))\mathbb{E}\left[\|\boldsymbol{\theta}'_t - \mathbf{1}_N \otimes \bar{\theta}'_t\|^2\right] + \frac{4H^2(2-\rho)\lambda_0^2}{(1-\rho)(t+1)^{3v}}
$$

$$
+ \frac{4H^2(2-\rho)\mathbb{E}\left[\|\boldsymbol{\theta}^* - \mathbf{1}_N \otimes \theta^*\|^2\right]\lambda_0}{(1-\rho)(t+1)^{2v}} + \frac{3N(2-\rho)\lambda_0^2\delta^2}{(1-\rho)(t+1)^{v+2r}} + \frac{N(2-\rho)\lambda_0^2\sigma^2}{(1-\rho)(t+1)^{2v}}
$$

$$
\leq (1 - (1-\rho))\mathbb{E}\left[\|\boldsymbol{\theta}'_t - \mathbf{1}_N \otimes \bar{\theta}'_t\|^2\right] + \frac{c_7}{(t+1)^{2v}}, \tag{131}
$$

with $c_7 = \frac{4H^2(2-\rho)\lambda_0^2 + 4H^2(2-\rho)\mathbb{E}\left[\|\boldsymbol{\theta}^* - \mathbf{1}_N \otimes \theta^*\|^2\right]\lambda_0 + 3N(2-\rho)\lambda_0^2\delta^2 + N(2-\rho)\lambda_0^2\sigma^2}{1-\rho}$.

By combining Lemma 11 in Chen & Wang (2025) and equation 131, we arrive at

$$
\mathbb{E}\left[\|\boldsymbol{\theta}'_t - \mathbf{1}_N \otimes \bar{\theta}'_t\|^2\right] \leq \frac{c_8}{(t+1)^{2v}}, \tag{132}
$$

where the constant $C_4$ is given by $c_8 = \left(\frac{8v}{e \ln(\frac{2}{1+\rho})}\right)^{2v}\left(\frac{\mathbb{E}\left[\|\boldsymbol{\theta}'_0 - \mathbf{1}_N \otimes \bar{\theta}'_0\|^2\right]\rho}{d_7} + \frac{2}{1-\rho}\right)$.

Substituting equation 132 into equation 130, we arrive at

$$
C_t\mathbb{E}[\|\theta'_{i,t+1} - \theta'_{i,t}\|^2] \leq \frac{2C_0 c_8 + \lambda_0^2 C_0(4H^2\mathbb{E}[\|\theta^* - \theta_i^*\|^2] + 2\sigma^2)}{\delta^2\lambda_0(t+1)^{2r}} + \frac{4C_0 C_1 H^2\lambda_0^2}{\delta^2(t+1)^{v+2r}}, \tag{133}
$$

Using the fact $v > r$ from the statement of Lemma 1, we have $\lim_{t\to\infty} C_t\mathbb{E}[\|\theta'_{i,t+1} - \theta'_{i,t}\|^2] = 0$ based on equation 129 and equation 133. Furthermore, by using an argument similar to the derivation of equation 129 and equation 133, we have $\lim_{t\to\infty} C_t\mathbb{E}[\|\theta'_{i,t} - \theta'_{i,t-1}\|^2] = 0$, which, combined with equation 127, leads to $\lim_{t\to\infty} C_t\mathbb{E}[\|\theta'_{i,t+1} - 2\theta'_{i,t} + \theta'_{i,t-1}\|^2] = 0$ for any $i \in [N]$. Therefore, based on equation 126, we have $\lim_{t\to\infty} \mathbb{E}[P_{i,t}] = 0$. $\qquad\square$

Corollary 3 guarantees that no payment is required from agent $i$ when it behaves fully truthfully as the number of iterations tends to infinity.

# F PROOF OF $\varepsilon$-INCENTIVE COMPATIBILITY IN THEOREM 2

In this section, we prove that in addition to achieving accurate convergence, our fully decentralized payment mechanism (Mechanism 1) also simultaneously ensures that Algorithm 1 is $\varepsilon$-incentive compatible in the presence of strategic behaviors.

## F.1 $\varepsilon$-INCENTIVE COMPATIBILITY FOR A STRONGLY CONVEX $F(\theta)$

We first present Lemma 14 to quantify the difference between the model parameters generated by Algorithm 1 in the absence of strategic behaviors and those generated by Algorithm 1 under our payment mechanism in the presence of strategic behaviors.

**Lemma 14.** *We denote $\theta'_{i,t}$ as the model parameter of agent $i$ generated by Algorithm 1 under our payment mechanism at iteration $t$ and $\theta_{i,t}$ as the model parameter of agent $i$ generated by the standard decentralized SGD at iteration $t$. Under the conditions in Lemma 1, for any $t \geq 0$, if $F(\theta)$ is $\mu$-strongly convex, the following inequality holds:*

$$
\sum_{i=1}^{N} \mathbb{E}\left[\|\theta_{i,t} - \theta'_{i,t}\|^2\right] \leq \frac{D_1}{(t+1)^v}, \tag{134}
$$

*where the constant $D_1$ is given by $D_1 = \frac{4\lambda_0^2(\lambda_0 + 2/\mu)(d_1 d_2\mu + 2N\delta^2)}{\mu\lambda_0 - 2v}$ with $d_1 = \frac{(3-\rho)^2}{1-\rho}(4H^2C_1 + \frac{4N}{\lambda_0}\delta^2) + \frac{N(3-\rho)}{1-\rho}(2H^2C_1 + \frac{4\delta^2}{\lambda_0})$, $d_2 = \frac{4^{3v+1}}{(1-\bar{\rho})(\ln(\sqrt{\bar{\rho}})e)^4}$, $C_1$ given in equation 94, and $\bar{\rho} = \frac{1+\rho}{2}$.*

*Proof.* Based on the dynamics of $\theta_{i,t}$ in Algorithm 1, we have

$$
\mathbb{E}\left[\|\boldsymbol{\theta}_{t+1} - \boldsymbol{\theta}'_{t+1}\|^2\right] \leq \mathbb{E}\left[\|(W \otimes I_n)(\boldsymbol{\theta}_t - \boldsymbol{\theta}'_t) - \lambda_t(\boldsymbol{g}(\boldsymbol{\theta}_t) - \boldsymbol{m}_t)\|^2\right], \tag{135}
$$

where $g(\boldsymbol{\theta}_t)$ and $\boldsymbol{m}_t$ satisfy

$$
\boldsymbol{g}(\boldsymbol{\theta}_t) - \boldsymbol{m}_t = \underbrace{\begin{pmatrix} g_1(\theta_{1,t}) - g_1(\theta'_{1,t}) \\ \vdots \\ g_i(\theta_{i,t}) - g_i(\theta'_{i,t}) \\ \vdots \\ g_N(\theta_{N,t}) - g_N(\theta'_{N,t}) \end{pmatrix}}_{G_{1,t}} + \underbrace{\begin{pmatrix} (1 - a_{1,t})g_1(\theta'_{1,t}) + b_{1,t}\xi_{1,t} \\ \vdots \\ (1 - a_{i,t})g_i(\theta'_{i,t}) + b_{i,t}\xi_{i,t} \\ \vdots \\ (1 - a_{N,t})g_N(\theta'_{N,t}) + b_{N,t}\xi_{N,t} \end{pmatrix}}_{G_{2,t}}. \tag{136}
$$

By substituting equation 136 into equation 135 and using the Young's inequality, we obtain

$$
\begin{aligned}
\mathbb{E}\left[\|\boldsymbol{\theta}_{t+1} - \boldsymbol{\theta}'_{t+1}\|^2\right] &\le (1 + \tau_1)\mathbb{E}\left[\|\boldsymbol{\theta}_t - \boldsymbol{\theta}'_t - \lambda_t G_{1,t}\|^2\right] \\
&+ \left(1 + \frac{1}{\tau_1}\right)\mathbb{E}\left[\|(W \otimes I_n - I_{Nn})(\boldsymbol{\theta}_t - \boldsymbol{\theta}'_t) - \lambda_t G_{2,t}\|^2\right],
\end{aligned} \tag{137}
$$

for any $\tau_1 > 0$.

According to the definition of $G_{1,t}$ in equation 136, the first term on the right hand side of equation 137 satisfies

$$
\begin{aligned}
\mathbb{E}\left[\|\boldsymbol{\theta}_t - \boldsymbol{\theta}'_t - \lambda_t G_{1,t}\|^2\right] &\le \mathbb{E}\left[\|\boldsymbol{\theta}_t - \boldsymbol{\theta}'_t\|^2\right] + \lambda_t^2 \mathbb{E}\left[\|G_{1,t}\|^2\right] + 2\mathbb{E}\left[\langle \boldsymbol{\theta}_t - \boldsymbol{\theta}'_t, -\lambda_t G_{1,t}\rangle\right] \\
&\le (1 - 2\mu\lambda_t)\mathbb{E}\left[\|\boldsymbol{\theta}_t - \boldsymbol{\theta}'_t\|^2\right] + \lambda_t^2 \mathbb{E}\left[\|G_{1,t}\|^2\right] \\
&\le \left(1 - 2\mu\lambda_t + H^2\lambda_t^2\right)\mathbb{E}\left[\|\boldsymbol{\theta}_t - \boldsymbol{\theta}'_t\|^2\right],
\end{aligned} \tag{138}
$$

where we have used the $\mu$-strong convexity of $F(\theta)$ in the second inequality and the $H$-Lipschitz continuity of $g_i(\theta)$ in the last inequality.

By using the relationship $W\mathbf{1}_N = \mathbf{1}_N$ from Assumption 2, the second term on the right hand side of equation 137 satisfies

$$
\begin{aligned}
&\mathbb{E}\left[\|(W \otimes I_n - I_{Nn})(\boldsymbol{\theta}_t - \boldsymbol{\theta}'_t) - \lambda_t G_{2,t}\|^2\right] \\
&\le 2\mathbb{E}\left[\|(W \otimes I_n - I_{Nn})\left((\boldsymbol{\theta}_t - \boldsymbol{\theta}'_t) - \mathbf{1}_N \otimes (\bar{\theta}_t - \bar{\theta}'_t)\right)\|^2\right] + 2\lambda_t^2 \mathbb{E}\left[\|G_{2,t}\|^2\right].
\end{aligned} \tag{139}
$$

We proceed to characterize the first term on the right hand side of equation 139. Based on the dynamics of $\theta_{i,t}$ in Algorithm 1, we have

$$
\begin{aligned}
&\mathbb{E}\left[\|(\boldsymbol{\theta}_{t+1} - \boldsymbol{\theta}'_{t+1}) - \mathbf{1}_N \otimes (\bar{\theta}_{t+1} - \bar{\theta}'_{t+1})\|^2\right] \\
&\le (1 + \tau_2)\mathbb{E}\left[\|(W \otimes I_n)\boldsymbol{\theta}_t - \lambda_t \boldsymbol{g}(\boldsymbol{\theta}_t) - \mathbf{1}_N \otimes \bar{\theta}_t - \left((W \otimes I_n)\boldsymbol{\theta}'_t - \lambda_t \boldsymbol{m}_t - \mathbf{1}_N \otimes \bar{\theta}'_t\right)\|^2\right] \\
&+ \left(1 + \frac{1}{\tau_2}\right)\mathbb{E}\left[\|\mathbf{1}_N \otimes (\bar{\theta}_t - \bar{\theta}'_t - \bar{\theta}_{t+1} + \bar{\theta}'_{t+1})\|^2\right] \\
&\le (1 + \tau_2)(1 + \tau_3)\mathbb{E}\left[\|(W \otimes I_n)\left((\boldsymbol{\theta}_t - \boldsymbol{\theta}'_t) - \mathbf{1}_N \otimes (\bar{\theta}_t - \bar{\theta}'_t)\right)\|^2\right] \\
&+ (1 + \tau_2)\left(1 + \frac{1}{\tau_3}\right)\lambda_t^2 \mathbb{E}\left[\|\boldsymbol{g}(\boldsymbol{\theta}_t) - \boldsymbol{m}_t\|^2\right] \\
&+ N\left(1 + \frac{1}{\tau_2}\right)\lambda_t^2 \mathbb{E}\left[\left\|\frac{1}{N}\sum_{i=1}^N g_i(\theta_{i,t}) - \frac{1}{N}\sum_{i=1}^N m_{i,t}\right\|^2\right],
\end{aligned}
$$

for any $\tau_2 > 0$ and $\tau_3 > 0$.

By setting $\tau_2 = \tau_3 = \frac{1-\rho}{2}$ and using the relationship $\left(1 + \frac{1-\rho}{2}\right)\rho = \left(1 + \frac{1-\rho}{2}\right)(1 - (1-\rho)) < 1 - \frac{1-\rho}{2} < 1$, we obtain

$$
\begin{aligned}
&\mathbb{E}\left[\|(\boldsymbol{\theta}_{t+1} - \boldsymbol{\theta}'_{t+1}) - \mathbf{1}_N \otimes (\bar{\theta}_{t+1} - \bar{\theta}'_{t+1})\|^2\right] \\
&\le \left(\frac{1+\rho}{2}\right)^2 \mathbb{E}\left[\|(\boldsymbol{\theta}_t - \boldsymbol{\theta}'_t) - \mathbf{1}_N \otimes (\bar{\theta}_t - \bar{\theta}'_t)\|^2\right] + \frac{(3-\rho)^2}{1-\rho}\lambda_t^2 \mathbb{E}\left[\|\boldsymbol{g}(\boldsymbol{\theta}_t) - \boldsymbol{m}_t\|^2\right] \\
&+ \frac{N(3-\rho)}{1-\rho}\lambda_t^2 \mathbb{E}\left[\left\|\frac{1}{N}\sum_{i=1}^N g_i(\theta_{i,t}) - \frac{1}{N}\sum_{i=1}^N m_{i,t}\right\|^2\right].
\end{aligned} \tag{140}
$$

From equation 51, we have $\mathbb{E}[\|G_{2,t}\|^2] \leq \sum_{i=1}^{N} \mathbb{E}[(a_{i,t}-1)^2\|g_i(\theta'_{i,t})\|^2] + \sum_{i=1}^{N} \mathbb{E}[\|b_{i,t}\xi_{i,t}\|^2] \leq 2N\kappa_t^2\delta^2$, which implies that the second term on the right hand side of equation 140 satisfies

$$
\begin{aligned}
\mathbb{E}\left[\|\boldsymbol{g}(\boldsymbol{\theta}_t) - \boldsymbol{m}_t\|^2\right] &\leq 2\mathbb{E}\left[\|G_{1,t}\|^2\right] + 2\mathbb{E}\left[\|G_{2,t}\|^2\right] \\
&\leq 2H^2\mathbb{E}\left[\|\boldsymbol{\theta}_t - \mathbf{1}_N \otimes \theta^* - (\boldsymbol{\theta}'_t - \mathbf{1}_N \otimes \theta^*)\|^2\right] + 4N\kappa_t^2\delta^2 \\
&\leq 4H^2 C_1 \lambda_t + 4N\kappa_t^2\delta^2,
\end{aligned}
\tag{141}
$$

where we have used equation 94 in the last inequality and the constant $C_1$ is given in equation 94.

Similarly, the last term on the right hand side of equation 140 satisfies

$$
\begin{aligned}
&\mathbb{E}\left[\left\|\frac{1}{N}\sum_{i=1}^{N} g_i(\theta_{i,t}) - \frac{1}{N}\sum_{i=1}^{N} m_{i,t}\right\|^2\right] \\
&\leq \mathbb{E}\left[\left\|\frac{1}{N}\sum_{i=1}^{N} g_i(\theta_{i,t}) - \frac{1}{N}\sum_{i=1}^{N} g_i(\theta'_{i,t}) + \frac{1}{N}\sum_{i=1}^{N}\left((1-a_{i,t})g_i(\theta'_{i,t}) + b_{i,t}\xi_{i,t}\right)\right\|^2\right] \\
&\leq 2H^2\frac{1}{N}\sum_{i=1}^{N}\mathbb{E}\left[\|\theta_{i,t} - \theta'_{i,t}\|^2\right] + 4\kappa_t^2\delta^2 \leq 2H^2 C_1\lambda_t + 4\kappa_t^2\delta^2.
\end{aligned}
\tag{142}
$$

Substituting equation 141 and equation 142 into equation 140, we obtain

$$
\begin{aligned}
&\mathbb{E}\left[\|(\boldsymbol{\theta}_{t+1} - \boldsymbol{\theta}'_{t+1}) - \mathbf{1}_N \otimes (\bar{\theta}_{t+1} - \bar{\theta}'_{t+1})\|^2\right] \\
&\leq \left(\frac{1+\rho}{2}\right)^2 \mathbb{E}\left[\|(\boldsymbol{\theta}_t - \boldsymbol{\theta}'_t) - \mathbf{1}_N \otimes (\bar{\theta}_t - \bar{\theta}'_t)\|^2\right] + \frac{(3-\rho)^2}{1-\rho}\lambda_t^2\left(4H^2 C_1\lambda_t + 4N\kappa_t^2\delta^2\right) \\
&\quad + \frac{N(3-\rho)}{1-\rho}\lambda_t^2\left(2H^2 C_1\lambda_t + 4\kappa_t^2\delta^2\right).
\end{aligned}
\tag{143}
$$

Given that the decaying rates of $\lambda_t$ and $\kappa_t$ satisfy $v \leq 2r$, equation 143 can be simplified as follows:

$$
\begin{aligned}
&\mathbb{E}\left[\|(\boldsymbol{\theta}_{t+1} - \boldsymbol{\theta}'_{t+1}) - \mathbf{1}_N \otimes (\bar{\theta}_{t+1} - \bar{\theta}'_{t+1})\|^2\right] \\
&\leq \left(\frac{1+\rho}{2}\right)^2 \mathbb{E}\left[\|(\boldsymbol{\theta}_t - \boldsymbol{\theta}'_t) - \mathbf{1}_N \otimes (\bar{\theta}_t - \bar{\theta}'_t)\|^2\right] + d_1\lambda_t^3,
\end{aligned}
\tag{144}
$$

where the constant $\rho$ is from Assumption 2 and the constant $d_1$ is given by $d_1 = \frac{(3-\rho)^2}{1-\rho}(4H^2 C_1 + \frac{4N}{\lambda_0}\delta^2) + \frac{N(3-\rho)}{1-\rho}(2H^2 C_1 + \frac{4\delta^2}{\lambda_0})$ with $C_1$ given in equation 94.

By telescoping equation 144 from 0 to $t-1$ and using $\mathbb{E}[\|\boldsymbol{\theta}_0 - \boldsymbol{\theta}'_0\|^2] = 0$, we obtain

$$
\mathbb{E}\left[\|(\boldsymbol{\theta}_t - \boldsymbol{\theta}'_t) - \mathbf{1}_N \otimes (\bar{\theta}_t - \bar{\theta}'_t)\|^2\right] \leq d_1\sum_{k=0}^{t-1}\lambda_k^3\left(\frac{1+\rho}{2}\right)^{2(t-1-k)}.
\tag{145}
$$

Substituting equation 145 and the relationship $\mathbb{E}[\|G_{2,t}\|^2] \leq 2N\kappa_t^2\delta^2$ into equation 139, we obtain

$$
\begin{aligned}
&\mathbb{E}\left[\|(W \otimes I_n - I_{Nn})(\boldsymbol{\theta}_t - \boldsymbol{\theta}'_t) - \lambda_t G_{2,t}\|^2\right] \\
&\leq 2d_1\sum_{k=0}^{t-1}(\lambda_k\sqrt{\lambda_k})^2\left(\frac{1+\rho}{2}\right)^{2(t-1-k)} + 4N\lambda_t^2\kappa_t^2\delta^2,
\end{aligned}
\tag{146}
$$

which, combined with Lemma 6, leads to

$$
\mathbb{E}\left[\|(W \otimes I_n - I_{Nn})(\boldsymbol{\theta}_t - \boldsymbol{\theta}'_t) - \lambda_t G_{2,t}\|^2\right] \leq \left(2d_1 d_2 + \frac{4N\delta^2}{\lambda_0}\right)\lambda_t^3,
\tag{147}
$$

where the constant $d_2$ is given by $d_2 = \frac{4^{3v+1}}{(1-\bar{\rho})(\ln(\sqrt{\bar{\rho}})e)^4}$ with $\bar{\rho} = \frac{1+\rho}{2}$.

Substituting equation 138 and equation 147 into equation 137 and letting $\tau_1 = \frac{\mu\lambda_t}{2}$, we arrive at

$$
\begin{aligned}
\mathbb{E}\left[\|\boldsymbol{\theta}_{t+1} - \boldsymbol{\theta}'_{t+1}\|^2\right] &\leq \left(1 + \frac{\mu\lambda_t}{2}\right)\left(1 - 2\mu\lambda_t + H^2\lambda_t^2\right)\mathbb{E}\left[\|\boldsymbol{\theta}_t - \boldsymbol{\theta}'_t\|^2\right] \\
&\quad + \left(1 + \frac{2}{\mu\lambda_t}\right)\left(2d_1 d_2 + \frac{4N\delta^2}{\lambda_0}\right)\lambda_t^3 \\
&\leq \left(1 - \frac{\mu\lambda_t}{2}\right)\mathbb{E}\left[\|\boldsymbol{\theta}_t - \boldsymbol{\theta}'_t\|^2\right] + d_3\lambda_t^2,
\end{aligned}
\tag{148}
$$

where the constant $d_3$ is given by $d_3 = (2d_1 d_2 + \frac{4N\delta^2}{\lambda_0})(\lambda_0 + \frac{2}{\mu})$ with $d_1$ and $d_2$ given in equation 144 and equation 147, respectively.

By combining Lemma 4 and equation 148, we arrive at equation 134. $\qquad\square$

We are now in a position to prove the $\varepsilon$-incentive compatibility of Algorithm 1 under our payment mechanism for a strongly convex $F(\theta)$. The result is summarized in Lemma 15, which corresponds to Theorem 2 in the main text.

**Lemma 15.** *Under our decentralized payment mechanism and the conditions in Theorem 2, if the global objective function $F(\theta)$ is strongly convex, then tAlgorithm 1 is $\varepsilon$-incentive compatible. Namely, for any $i \in [N]$, $\delta > 0$, and $T \geq 0$ (which includes the case of $T = \infty$), the following inequality holds:*

$$
\mathbb{E}[U_{i,0\to T}^{\mathcal{M}_p}(\boldsymbol{\alpha}_i, \boldsymbol{h}_{-i}) - U_{i,0\to T}^{\mathcal{M}_p}(\boldsymbol{h}_i, \boldsymbol{h}_{-i})] \leq \varepsilon,
\tag{149}
$$

*with $U_{i,0\to T}^{\mathcal{M}_p}(\boldsymbol{\alpha}_i, \boldsymbol{h}_{-i})$ and $U_{i,0\to T}^{\mathcal{M}_p}(\boldsymbol{h}_i, \boldsymbol{h}_{-i})$ defined in equation 3 and $\varepsilon$ given by $\varepsilon = \mathcal{O}\left(\frac{L_{R,i}\delta}{v+r-1}\right)$.*

*Proof.* By using the $L_{R,i}$-Lipschitz continuity of $R_i(f_i(\theta))$ w.r.t. $\theta$, we obtain

$$
(\mathbb{E}[|R_i(f_i(\theta'_{i,T+1})) - R_i(f_i(\theta_{i,T+1}))|])^2 \leq L_{R,i}^2\mathbb{E}[\|\theta'_{i,T+1} - \theta_{i,T+1}\|^2] \leq L_{R,i}^2 D_1(T+1)^{-v},
$$

where we have used equation 134 in the last inequality.

By using Lyapunov's inequality for moments, we have

$$
\mathbb{E}[|R_i(f_i(\theta'_{i,t+1})) - R_i(f_i(\theta_{i,t+1}))|] \leq L_{R,i}\sqrt{D_1}(T+1)^{-\frac{v}{2}}.
\tag{150}
$$

According to equation 65, we have

$$
\begin{aligned}
\mathbb{E}\left[P_{i,t}(h_{i,t}, h_{-i,t}) - P_{i,t}(\alpha_{i,t}, h_{-i,t})\right] &\leq -\frac{\lambda_t^2}{2}C_t\deg(i)\mathbb{E}[\|(a_{i,t}-1)g_i(\theta_{i,t})\|^2] \\
&\quad - \frac{\lambda_t^2}{2}C_t\deg(i)\mathbb{E}[\|b_{i,t}\xi_{i,t}\|^2] + d_7\deg(i)C_t\lambda_t^2\lambda_{t-1}^2.
\end{aligned}
\tag{151}
$$

Summing both sides of equation 151 from $t = 1$ to $t = T$ and using equation 51 yield

$$
\begin{aligned}
\sum_{t=1}^{T}&\mathbb{E}\left[P_{i,t}(h_{i,t}, h_{-i,t}) - P_{i,t}(\alpha_{i,t}, h_{-i,t})\right] \\
&\leq -\sum_{t=1}^{T}\deg(i)C_t\lambda_t^2\kappa_t^2\delta^2 - \sum_{t=1}^{T}\deg(i)d_7 C_t\lambda_t^2\lambda_{t-1}^2 \leq -2\deg(i)\sum_{t=1}^{T}C_t\lambda_t^2\kappa_t^2\delta^2,
\end{aligned}
\tag{152}
$$

where in the derivation we have used the fact that the decaying rate of $\lambda_t^2\lambda_{t-1}^2$ is faster than $\lambda_t^2\kappa_t^2$.

Furthermore, by using equation 94, we have

$$
-K\mathbb{E}[\|\theta'_{i,T+1} - \theta^*\|^2] + K\mathbb{E}[\|\theta_{i,T+1} - \theta^*\|^2] \leq \frac{2KC_1}{(T+1)^v},
\tag{153}
$$

where $C_1$ is given in equation 94 and $K$ is any positive constant.

Combining equation 150, equation 152, and equation 153, and using the definition $C_t = \frac{C_0 \kappa_t^2}{\delta^2 (t+1)^{-2v}}$, we have

$$
\mathbb{E}\left[ R_i(f_i(\theta'_{i,T+1})) - K\|\theta'_{i,T+1} - \theta^*\|^2 - \sum_{t=1}^{T} P_{i,t}(\alpha_{i,t}, h_{-i,t}) \right]
$$

$$
- \mathbb{E}\left[ R_i(f_i(\theta_{i,T+1})) - K\|\theta_{i,T+1} - \theta^*\|^2 - \sum_{t=1}^{T} P_{i,t}(h_{i,t}, h_{-i,t}) \right] \tag{154}
$$

$$
\leq \frac{L_{R,i}\sqrt{D_1}}{(T+1)^{\frac{v}{2}}} + \frac{2KC_1}{(T+1)^v} + \frac{2\deg(i)C_0\lambda_0^2}{(2r-1)(T+1)^{2r-1}} + \frac{2\deg(i)C_0\lambda_0^2}{2r-1},
$$

where in the derivation we have used the inequality $\sum_{t=1}^{T} \lambda_t \kappa_t \geq \int_1^{T+1} \frac{\lambda_0}{(t+1)^{2r}} dt = \frac{\lambda_0(1-(T+1)^{1-2r})}{2r}$.

Based on the relation $\frac{1}{2} < r$ from the statement of Lemma 1 (or Theorem 2) and the relationship $C_0 \leq \mathcal{O}(L_R \delta)$ (see the definition of $C_0$ in equation 72), we arrive at equation 149 and equation 6 in Theorem 2. $\qquad \square$

### F.2 $\varepsilon$-INCENTIVE COMPATIBILITY FOR A GENERAL CONVEX $F(\theta)$

We present the following Lemma 16 to prove the $\varepsilon$-incentive compatibility of Algorithm 1 under our payment mechanism for a general convex $F(\theta)$, which corresponds to Theorem 2 in the main text.

**Lemma 16.** *Under our decentralized payment mechanism and the conditions in Theorem 2, if the global objective function $F(\theta)$ is general convex, then Algorithm 1 is $\varepsilon$-incentive compatible. Namely, for any $i \in [N]$, $\delta > 0$, and $T \geq 0$ (which includes the case of $T = \infty$), the following inequality holds:*

$$
\mathbb{E}[U_{i,0\to T}^{\mathcal{M}_p}(\boldsymbol{\alpha}_i, \boldsymbol{h}_{-i}) - U_{i,0\to T}^{\mathcal{M}_p}(\boldsymbol{h}_i, \boldsymbol{h}_{-i})] \leq \varepsilon, \tag{155}
$$

*with $U_{i,0\to T}^{\mathcal{M}_p}(\boldsymbol{\alpha}_i, \boldsymbol{h}_{-i})$ and $U_{i,0\to T}^{\mathcal{M}_p}(\boldsymbol{h}_i, \boldsymbol{h}_{-i})$ defined in equation 3 and $\varepsilon$ given by $\varepsilon = \mathcal{O}\left(\frac{L_R\delta}{v+r-1}\right)$.*

*Proof.* According to equation 100, we have $\frac{1}{T+1}\sum_{t=0}^{T} \mathbb{E}\left[\|\nabla F(\bar\theta'_t)\|^2\right] \leq \mathcal{O}\left(\frac{H^3(\sigma^2 + L_f^2 + \delta^2)}{(1-\rho)^2(T+1)^{1-v}}\right)$. According to the Stolz–Cesàro theorem (for two real-number sequences $\{a_t\}_{t\geq 1}$ and $\{b_t\}_{t\geq 1}$, if $\{b_t\}$ is strictly monotonic and tends to infinity, and both $\lim_{t\to\infty} \frac{a_t}{b_t} = \ell$ and $\lim_{t\to\infty} \frac{a_{t+1}-a_t}{b_{t+1}-b_t}$ hold, then we have $\lim_{t\to\infty} \frac{a_{t+1}-a_t}{b_{t+1}-b_t} = \ell$), we obtain

$$
\lim_{t\to\infty} \frac{1}{t} \sum_{k=0}^{t-1} \mathbb{E}\left[\|\nabla F(\bar\theta'_k)\|^2\right] = 0 \implies \lim_{t\to\infty} \mathbb{E}[\|\nabla F(\bar\theta'_t)\|^2] = 0, \tag{156}
$$

where we have set $a_t = \sum_{k=0}^{t-1} \mathbb{E}\left[\|\nabla F(\bar\theta'_k)\|^2\right]$ and $b_t = t$ when we use the Stolz–Cesàro theorem.

We let $\Theta^* = \{\theta \in \mathbb{R}^n \mid \nabla F(\theta) = 0\}$ denote the optimal-solution set. By the continuity of $\nabla F(\theta)$ together with equation 156, we have $\lim_{t\to\infty} d(\bar\theta'_t, \Theta^*) = 0$, where $d(x, \Theta^*) = \inf_{y\in\Theta^*} \|x-y\|$ denotes the distance from $x$ to $\Theta^*$. Furthermore, by the continuity of $f_i(\theta)$, we have $\lim_{t\to\infty} f_i(\bar\theta'_t) = f_i(\theta'^*)$ for some $\theta'^* \in \Theta^*$.

Given that $m_{i,t} = g_i(\theta_{i,t})$ is a special case of $m_{i,t} = a_{i,t} g_i(\theta'_{i,t}) + b_{i,t} \xi_{i,t}$ with $a_{i,t} \geq 1$ and $b_{i,t} \in \mathbb{R}$, we have $\lim_{t\to\infty} f_i(\bar\theta_t) = f_i(\theta^*)$ for some $\theta^* \in \Theta^*$. Note that when $F(\theta)$ is convex, the optimal solutions $\theta^*$ and $\theta'^*$ may be different elements in $\Theta^*$.

We proceed to prove that for any two points $\theta_1^*, \theta_2^* \in \Theta^*$, the relationship $f_i(\theta_1^*) = f_i(\theta_2^*)$ always holds. Since $F(\theta)$ is convex, its optimal-solution set $\Theta^*$ is convex, closed, and connected. We choose some point $\theta_0^* \in \Theta^*$ and consider a direction $b \in \mathbb{R}^n$ such that the segment $\theta_0^* + \varsigma b$ lies within $\Theta^*$ for an arbitrarily small $\varsigma > 0$. Since $F(\theta)$ is constant over $\Theta^*$, its first and second directional derivatives at $\theta_0^*$ are zero, i.e., $\langle b, \nabla F(\theta_0^*)\rangle = 0$ and $b^\top \nabla^2 F(\theta_0^*) b = 0$. Recalling $F(\theta) = \frac{1}{N}\sum_{i=1}^{N} f_i(\theta)$, we have $b^\top \frac{1}{N}\sum_{i=1}^{N} \nabla^2 f_i(\theta_0^*) b = 0$. Furthermore, since each $f_i$ is convex and twice differentiable, we have $\nabla^2 f_i(\theta_0^*) \succeq 0$ and $b^\top \nabla^2 f_i(\theta_0^*) b \succeq 0$, which, combined with

$b^\top \nabla^2 F(\theta_0^*) b = 0$, leads to $b^\top \nabla^2 f_i(\theta_0^*) b = 0$. Hence, each $f_i(\theta)$ has zero second directional derivative along any direction in $\Theta^*$, meaning that $f_i(\theta)$ is constant on $\Theta^*$. Hence, for any $\theta_1^*, \theta_2^* \in \Theta^*$, we have $f_i(\theta_1^*) = f_i(\theta_2^*)$, which naturally leads to $f_i(\theta'^*) = f_i(\theta^*)$.

By using the relationship $f_i(\theta'^*) = f_i(\theta^*)$ and equation 100, we have

$$\lim_{T\to\infty} \mathbb{E}[R_i(f_i(\theta'_{i,T+1})) - R_i(f_i(\theta_{i,T+1}))] = \mathbb{E}[R_i(f_i(\theta'^*)) - R_i(f_i(\theta^*))] = 0. \tag{157}$$

Furthermore, since the relations $\lim_{T\to\infty} \mathbb{E}[f_i(\theta'_{i,T})] = f_i(\theta'^*)$ and $\lim_{T\to\infty} \mathbb{E}[f_i(\theta_{i,T})] = f_i(\theta^*)$ hold, we have $\lim_{T\to\infty} \mathbb{E}[\|\theta'_{i,T+1} - \theta'^*\|] = 0$ and $\lim_{T\to\infty} \mathbb{E}[\|\theta_{i,T+1} - \theta^*\|] = 0$, which implies

$$\lim_{T\to\infty} \mathbb{E}[-K\|\theta'_{i,T+1} - \theta^*\|^2 + K\|\theta_{i,T+1} - \theta^*\|^2] \le 2K\mathrm{d}(\Theta^*), \tag{158}$$

where $\mathrm{d}(\Theta^*) = \sup_{x,y\in\Theta^*} \|x-y\|$ denotes the diameter of the optimal solution set $\Theta^*$.

According to equation 152 and the definition $C_t = \frac{C_0\kappa^2}{\delta^2(t+1)^{-2v}}$, we have

$$\sum_{t=1}^T \mathbb{E}\left[P_{i,t}(h_{i,t}, h_{-i,t}) - P_{i,t}(\alpha_{i,t}, h_{-i,t})\right] \le -2\deg(i)C_0 \sum_{t=1}^T \frac{\lambda_t^2\kappa_t^2}{(t+1)^{-2v}} \tag{159}$$
$$\le \frac{2\deg(i)C_0\lambda_0^2(1-(T+1)^{1-2r})}{2r-1},$$

where in the derivation we have used $\sum_{t=1}^T \kappa_t^2 \ge \int_1^{T+1} \frac{1}{(t+1)^{2r}} dt = \frac{\lambda_0(1-(T+1)^{1-2r})}{2r-1}$.

Combining equation 157, equation 158, and equation 159, we arrive at

$$\lim_{T\to\infty} \mathbb{E}\left[R_i(f_i(\theta'_{i,T+1})) - K\|\theta'_{i,T+1} - \theta^*\|^2 - \sum_{t=1}^T P_{i,t}(\alpha_{i,t}, h_{-i,t})\right]$$
$$- \lim_{T\to\infty} \mathbb{E}\left[R_i(f_i(\theta_{i,T+1})) - K\|\theta_{i,T+1} - \theta^*\|^2 - \sum_{t=1}^T P_{i,t}(h_{i,t}, h_{-i,t})\right] \tag{160}$$
$$\le 2K\mathrm{d}(\Theta^*) + \frac{2\deg(i)C_0\lambda_0^2}{2r-1}.$$

Furthermore, according to equation 160, for any finite $T$, there must exist a $C > 0$ such that the following inequality holds:

$$\mathbb{E}\left[R_i(f_i(\theta'_{i,T+1})) - K\|\theta'_{i,T+1} - \theta^*\|^2 - \sum_{t=1}^T P_{i,t}(\alpha_{i,t}, h_{-i,t})\right]$$
$$- \mathbb{E}\left[R_i(f_i(\theta_{i,T+1})) - K\|\theta_{i,T+1} - \theta^*\|^2 - \sum_{t=1}^T P_{i,t}(h_{i,t}, h_{-i,t})\right] \le C\frac{2\deg(i)C_0\lambda_0^2}{2r-1}, \tag{161}$$

which, combined with equation 160 and the relationship $C_0 \le \mathcal{O}(L_R\delta)$ (see the definition of $C_0$ in equation 72), leads to equation 155 and equation 6 in Theorem 2. $\square$

## G  DISCUSSION ON PARAMETER MANIPULATION

In this section, we demonstrate that a strategic agent gains no clear advantage in decentralized learning by misreporting its model parameter as a constant in an attempt to increase its received payments.

**Lemma 17.** *Under the conditions of Lemma 1, for a strategic agent whose neighboring agents are honest, if the agent reports a fake constant $c > 0$ in Algorithm 1 in an attempt to increase its received payments, the total payments received by the agent remain finite.*

*Proof.* Without loss of generality, we assume that agent $i$ shares a fake constant model parameter $c$ in Algorithm 1. Then, for an agent $j \in \mathcal{N}_i$, its dynamics of $\theta_{j,t}$ satisfies $\theta_{j,t+1} =$

$\sum_{k \in \mathcal{N}_j \cup \{j\}} w_{jk} \theta_{k,t} - \lambda_t g_j(\theta_{j,t})$. By defining $\tilde{\theta}_{i,t} = \theta_{i,t} - c$ and using the relationship $W \mathbf{1}_N = \mathbf{1}_N$, we obtain

$$\tilde{\theta}_{j,t+1} = \sum_{k \in \mathcal{N}_j \cup \{j\}} w_{jk} \tilde{\theta}_{k,t} - \lambda_t g_j(\theta_{j,t}). \tag{162}$$

Defining two auxiliary variables $\tilde{\boldsymbol{\theta}}_{-i,t} = \mathrm{col}(\tilde{\theta}_{1,t}, \cdots, \tilde{\theta}_{i-1,t}, \tilde{\theta}_{i+1,t}, \cdots, \tilde{\theta}_{N,t})$ and $\boldsymbol{g}_{-i,t} = \mathrm{col}(g_1(\theta_{1,t}), \cdots, g_i(\theta_{i-1,t}), g_i(\theta_{i+1,t}), \cdots, g_N(\theta_{N,t}))$, the dynamics of the stacked variable $\tilde{\boldsymbol{\theta}}_{-i,t}$ can be written as

$$\tilde{\boldsymbol{\theta}}_{-i,t+1} = (\tilde{W} \otimes I_n) \tilde{\boldsymbol{\theta}}_{-i,t} - \lambda_t \boldsymbol{g}_{-i,t}, \tag{163}$$

where $\tilde{W}$ denotes the submatrix of $W$ obtained by removing its $i$th row and column.

We define $L = I_N - W$. According to Assumption 2, we have that the eigenvalues of $L$ satisfy $0, 1 - \pi_2, \ldots, 1 - \pi_N$. Note that $\pi_1 = 1$.

We denote $\tilde{L} \in \mathbb{R}^{(N-1) \times (N-1)}$ as the submatrix of $L$ obtained by removing the $i$th row and column. By applying a similarity transformation, the matrix $\pi I_N - L$ with $\pi \in \mathbb{C}$ can be written in the block form as follows:

$$\begin{bmatrix} \pi & 0 & \cdots & 0 \\ * & \pi I_{N-1} - \tilde{L} \end{bmatrix},$$

which implies $\det(\pi I_N - L) = \pi \det(\pi I_{N-1} - \tilde{L})$.

Furthermore, since the eigenvalues of $L$ are $0, 1 - \pi_2, \ldots, 1 - \pi_N$, we have $\det(\pi I_N - L) = \pi \prod_{i=2}^{N} (\pi - (1 - \pi_i))$, which implies $\det(\pi I_{N-1} - \tilde{L}) = \prod_{i=2}^{N} (\pi - (1 - \pi_i))$.

Therefore, the eigenvalues of $\tilde{L}$ are $1 - \pi_2, \ldots, 1 - \pi_N$. Furthermore, according to the definition of $\tilde{W} = I_{N-1} - \tilde{L}$, the eigenvalues of $\tilde{W}$ are $\pi_2, \ldots, \pi_N$, which implies $\|\tilde{W}\| = \rho$.

By using the Young's inequality, we have

$$\mathbb{E}[\|\tilde{\boldsymbol{\theta}}_{-i,t+1}\|^2] = (1 + (1-\rho))\rho^2 \mathbb{E}[\|\tilde{\boldsymbol{\theta}}_{-i,t}\|^2] + \left(1 + \frac{1}{1-\rho}\right) \lambda_t^2 \mathbb{E}[\|\boldsymbol{g}_{-i,t}\|^2]. \tag{164}$$

The last term on the right hand side of equation 164 satisfies

$$\mathbb{E}[\|\boldsymbol{g}_{-i,t}\|^2] = \sum_{j \neq i}^{N} \left( \mathbb{E}[\|g_{j,t}(\theta_{j,t}) - \nabla f_j(\theta_{j,t})\|^2] + \mathbb{E}[\|\nabla f_j(\theta_{j,t})\|^2] \right)$$
$$\leq (N-1)\sigma^2 + 2H^2 \sum_{j \neq i}^{N} \mathbb{E}[\|\theta_{j,t} - c\|^2] + 2H^2 \sum_{j \neq i}^{N} \|\theta_j^* - c\|^2. \tag{165}$$

Substituting equation 165 into equation 164, we arrive at

$$\mathbb{E}[\|\tilde{\boldsymbol{\theta}}_{-i,t+1}\|^2] = (1 + (1-\rho))\rho^2 \mathbb{E}[\|\tilde{\boldsymbol{\theta}}_{-i,t}\|^2] + \frac{2-\rho}{1-\rho} \lambda_t^2 (N-1)\sigma^2$$

$$+ \frac{2H^2(2-\rho)}{1-\rho} \lambda_t^2 \mathbb{E}[\|\tilde{\boldsymbol{\theta}}_{-i,t}\|] + \frac{2H^2(2-\rho)}{1-\rho} \lambda_t^2 \sum_{j \neq i}^{N} \|\theta_j^* - c\|^2 \tag{166}$$

$$\leq \left( \rho + \frac{2H^2(2-\rho)}{1-\rho} \lambda_t^2 \right) \mathbb{E}[\|\tilde{\boldsymbol{\theta}}_{-i,t}\|^2] + d_9 \lambda_t^2 \leq \left( 1 - \frac{1-\rho}{2} \right) \mathbb{E}[\|\tilde{\boldsymbol{\theta}}_{-i,t}\|^2] + d_9 \lambda_t^2,$$

where in the last inequality we have used the fact that we can choose $\lambda_0 < \frac{(1-\rho)^2}{4H^2(2-\rho)}$ such that $\frac{2H^2(2-\rho)}{1-\rho} < \frac{1-\rho}{2}$ holds under the decaying sequence $\lambda_t$.

By applying Lemma 11 in Chen & Wang (2025) to equation 166, we arrive at

$$\mathbb{E}[\|\tilde{\theta}_{j,t}\|] \leq \mathbb{E}[\|\tilde{\boldsymbol{\theta}}_{-i,t}\|^2] \leq d_{10} \lambda_t^2, \tag{167}$$

where $d_{10}$ is given by $d_{10} = \left(\frac{8v}{e \ln(\frac{4}{3+\rho})}\right)^{2v} \left(\frac{\mathbb{E}[\|\tilde{\boldsymbol{\theta}}_{-i,0}\|^2](1+\rho)}{2d_9 \lambda_0^2} + \frac{4}{1-\rho}\right)$.

Next, we estimate an upper bound on $\mathbb{E}[\|\theta_{j,t+1} - 2\theta_{j,t} + \theta_{j,t-1}\|^2]$.

$$\mathbb{E}[\|\theta_{j,t+1} - 2\theta_{j,t} + \theta_{j,t-1}\|^2]$$

$$= \mathbb{E}\left[\left\|\sum_{k \in \mathcal{N}_j \cup \{j\}} w_{jk}\theta_{k,t} - \lambda_t g_j(\theta_{j,t}) - \theta_{j,t} - \left(\sum_{k \in \mathcal{N}_j \cup \{j\}} w_{jk}\theta_{k,t-1} - \lambda_t g_j(\theta_{j,t-1}) - \theta_{j,t-1}\right)\right\|^2\right]$$

$$\leq 3\mathbb{E}\left[\left\|\sum_{k \in \mathcal{N}_j \cup \{j\}} w_{jk}(\theta_{k,t} - \theta_{k,t-1})\right\|^2\right]$$

$$+ 3\mathbb{E}[\|\theta_{j,t} - \theta_{j,t-1}\|^2] + 3\mathbb{E}[\|\lambda_t g_j(\theta_{j,t}) - \lambda_{t-1}g_j(\theta_{j,t-1})\|^2]. \tag{168}$$

The first term on the right hand side of equation 168 satisfies

$$3\mathbb{E}\left[\left\|\sum_{k \in \mathcal{N}_j \cup \{j\}} w_{jk}(\theta_{k,t} - \theta_{k,t-1})\right\|^2\right] \leq 6\mathbb{E}[\|\tilde{\boldsymbol{\theta}}_{-i,t}\|^2] + 6\mathbb{E}[\|\tilde{\boldsymbol{\theta}}_{-i,t-1}\|^2] \leq 12d_{10}\lambda_{t-1}^2, \tag{169}$$

where in the derivation we have used equation 167.

By using equation 167, the second term on the right hand side of equation 168 can be verified to satisfy $3\mathbb{E}[\|\theta_{j,t} - \theta_{j,t-1}\|^2] \leq 12d_{10}\lambda_{t-1}^2$.

The last term on the right hand side of equation 168 satisfies

$$3\mathbb{E}[\|\lambda_t g_j(\theta_{j,t}) - \lambda_{t-1}g_j(\theta_{j,t-1})\|^2]$$

$$= 3\mathbb{E}[\|\lambda_t g_j(\theta_{j,t}) - \lambda_t \nabla f_j(\theta_{j,t}) - (\lambda_{t-1}g_j(\theta_{j,t-1}) - \lambda_{t-1}\nabla f_j(\theta_{j,t-1}))\|^2]$$

$$+ 3\mathbb{E}[\|\lambda_t \nabla f_j(\theta_{j,t}) - \lambda_{t-1}\nabla f_j(\theta_{j,t-1})\|^2]$$

$$\leq 3\sigma^2(\lambda_t^2 + \lambda_{t-1}^2) + 6(\lambda_t - \lambda_{t-1})^2\mathbb{E}[\|\nabla f_j(\theta_{j,t-1})\|^2] + 6\lambda_t^2 H^2\mathbb{E}[\|\theta_{j,t} - \theta_{j,t-1}\|^2] \tag{170}$$

$$\leq 6\sigma^2\lambda_{t-1}^2 + 6(\lambda_t - \lambda_{t-1})^2 H^2\mathbb{E}[\|\theta_{j,t-1} - c + c - \theta_j^*\|^2] + 24\lambda_t^2 H^2 d_{10}\lambda_{t-1}^2$$

$$\leq 6\sigma^2\lambda_{t-1}^2 + 12(\lambda_t - \lambda_{t-1})^2 H^2\|c - \theta_j^*\|^2 + 12(\lambda_t - \lambda_{t-1})^2 H^2 d_{10}\lambda_{t-1}^2$$

$$+ 24H^2 d_{10}\lambda_t^2\lambda_{t-1}^2.$$

Substituting equation 169, equation 170, and the relationship $3\mathbb{E}[\|\theta_{j,t} - \theta_{j,t-1}\|^2] \leq 12d_{10}\lambda_{t-1}^2$ into equation 168, we arrive at

$$\mathbb{E}[\|\theta_{j,t+1} - 2\theta_{j,t} + \theta_{j,t-1}\|^2] \leq (24d_{10} + 6\sigma^2)\lambda_{t-1}^2 + 12(\lambda_t - \lambda_{t-1})^2 H^2\|c - \theta_j^*\|^2$$

$$+ 12(\lambda_t - \lambda_{t-1})^2 H^2 d_{10}\lambda_{t-1}^2 + 24H^2 d_{10}\lambda_t^2\lambda_{t-1}^2 \leq d_{11}\lambda_{t-1}^2, \tag{171}$$

where the constant $d_{11}$ is given by $d_{11} = 24d_{10} + 6\sigma^2 + 48H^2\|c - \theta_j^*\|^2 + 12(\lambda_1 - \lambda_0)^2 H^2 d_{10} + 24H^2 d_{10}\lambda_0^2$.

Recalling the payment mechanism in Mechanism 1, when agent $i$ shares a constant $c$ over iterations, its payment $P_{i,t}$ satisfies

$$\mathbb{E}[P_{i,t}] = -\sum_{j \in \mathcal{N}_i} C_t \mathbb{E}[\|\theta_{j,t+1} - 2\theta_{j,t} + \theta_{j,t-1}\|^2] \leq \frac{2^{2v}\deg(i)C_0 d_{11}\lambda_0^2}{\delta^2(t+1)^{2r-2v+2v}}, \tag{172}$$

where in the derivation we have used the definitions $C_t = \frac{C_0\kappa_t^2}{\delta^2(t+1)^{-2v}}$ and $\kappa_t = \frac{1}{(t+1)^{2r}}$ with $1-v < r < v$ and $\frac{1}{2} < v < 1$. By setting $\frac{1}{2} < r < v$, we arrive at

$$\sum_{t=0}^{T}\mathbb{E}[P_{i,t}] = \sum_{t=0}^{T}\frac{2^{2v}\deg(i)C_0 d_{11}\lambda_0^2}{\delta^2(t+1)^{2r}}, \tag{173}$$

with the exponent $2r$ for $t+1$ satisfying $2r > 1$, implying that the the cumulative payment of agent $i$ remains finite even as $T \to \infty$. $\qquad\square$

Lemma 17 shows that even if an agent falsely reports a constant value for its model parameter—which would place it in the most advantageous position to obtain payments from its neighbors under our payment Mechanism 1—its cumulative payment still remains finite. Moreover, this cumulative payment depends on external factors such as the network structure and the properties of the functions of its neighbors, none of which the agent can accurately predict or control. Consequently, since an agent's net utility is determined by both the quality of the final learned model and the payments it receives, a strategic agent cannot guarantee that its net utility will be maximized by ignoring the learned model's quality and focusing solely on manipulating its reported model parameters to increase payments.

