# OpenReview forum: "Incentivizing Truthfulness in Fully Decentralized Learning with Guaranteed Accurate Convergence"
_ICLR.cc/2026/Conference — Submitted to ICLR 2026_

### Official Review · Reviewer_m7Yo · 2025-10-20

**Soundness:** 1
**Presentation:** 2
**Contribution:** 1
**Rating:** 2
**Confidence:** 4

**Summary:**

This paper proposes a new payment scheme to incentivize honest gradient reporting in decentralized SGD. For agents that only care about the quality of the learnt model on their own distribution,  the proposed mechanism adresses two kinds of "attacks": Noise injection and gradient inflation. It works by having agents compare their squared gradient norms, and the agent with the larger one making a payment proportional to the difference. Unlike prior works, the proposed mechanism does not require a central server, is budget-balanced, and is claimed to yield similar convergence guarantees to the fully honest case, even without convexity assumptions.

**Strengths:**

- The paper tackles an interesting and relevant problem
- The empirical results seem to show that the propsed mechanism succeeds
- The writing is mostly easy to follow

**Weaknesses:**

- **Main issue**: The proof of Lemma 1 seems to be highly incomplete (and the statement is likely wrong at the level of generality the papers considers). The proof appears to only show that an agent that cares about model quality at time t+1 is not incentivized to manipulate gradients at time t, but this does not seem to establish that the same is true for agents that care about the model quality at time T+1 (which is what determines utility).
   - It seems like some of the improvements over prior work, such as the extension to non-convexity are only possible due to this incomplete treatment of delayed effects.
- Some of the claimed contributions are unclear: The paper claims that various prior works do not achieve eps-incentive compatibility and accurate convergence, but does not provide much support for the claim. In a similar vein, it is unclear whether the extension to the non-convex case is due to an improvement in the mechanism, or refined analysis.
- Relatedly, the discussion mostly focuses on connections to VCG mechanisms, but the  proposed mechanism appears to be heavily inspired by Chakarov et al. Specifically, in a fully connected graph, the proposed mechanism seems to be equivalent to Chakarov's up to rescaling.
- As in Chakarov et al, the attack model is somewhat restricted. In particular without the seemingly arbitrary assumption of $\alpha\geq 1$, it seems like under the proposed scheme some agents could be incentivized to report zeros for the gradients in order to collect payments, while "free-riding" on other players' gradient information.
- Nitpicks
  - The notation for $\lambda$ is overloaded (step size and Eigenvalues)
  - The notation for the mechanism could be simplified by allowing for negative payments and simply setting agent i's payment to agent j as the difference in squared gradient norms.

**Questions:**

- Can you elaborate in more detail, how most of the cited mechanisms are based on VGC? I am no expert on this, but after looking at the definition, the connection seems unclear for some works like Chakarov et al.

---

> ### Author Response · Authors · 2025-11-20
> **Response to Reviewer m7Yo---Part I**
>
> $\rm\color{blue}{@m7Yo\ @Weakness\ 1:}$  We thank the reviewer for the time spent reviewing our manuscript.
>
> First, we clarify that in the initial version, we primarily considered myopic agents, in the sense that at time $t$, an agent chooses whether to manipulate its gradient based on the expected gain at time $t+1$. This step-wise incentive structure is standard and has been adopted in existing results for server–assisted federated learning in Chakarov et al. (2024a;b). Our Lemma 1 established that our decentralized payment mechanism ensures that an agent's manipulation at any iteration $t$ does not improve its utility at iteration $t+1$. Consequently, the agent has no incentive to behave untruthfully at any iteration $t\in[0,T]$.
>
> Nevertheless, following the reviewer's suggestion, we have revised the proof of Lemma 1 in the manuscript to show that, even when agents consider long-term rewards, the incentive for an agent (assuming truthful neighbors) to deviate from truthful behavior still converges to zero (see Appendix Section D.1 in the revised manuscript). For the reviewer's convenience, we present the revised Lemma 1 below (we also note that our payment mechanism has been refined: rather than relying on differences in gradient magnitudes, it now uses differences in model-parameter increments, which allows us to address a broader class of attacks, including free-riding and payment manipulation):
>
> **''Lemma 1. Under Assumption 1 and Assumption 2, for any $i\in[N]$, $\delta>0$, and $t\geq 0$, if we set $C_{t}=\frac{C_{0}}{\delta^2\lambda_{t}\kappa_{t}}$ with $\lambda_{t}=\frac{\lambda_{0}}{(t+1)^{v}}$, $\kappa_{t}=\frac{1}{(t+1)^{r}}$, $v\in(\frac{1}{2},\frac{2}{3})$, $r\in(1-v,v)$, and $C_{0}$ given in equation 71, and all neighbors of agent $i$ are truthful, then the optimal action for agent $i$ in decentralized SGD is $\kappa_{t}\delta$-truthful. Moreover, as the number of iterations tends to infinity, the optimal action for agent $i$ is fully truthful, i.e., an agent will have zero incentive to deviate from truthful behavior."**
>
> Since Lemma 1 continues to hold for farsighted agents caring about long-term rewards, our accurate convergence and $\varepsilon$-incentive compatibility results remain valid.
>
> In addition, we would like to note that ensuring truthful actions of farsighted agents requires to analyze the algorithmic sensitivity of the learning dynamics (as indicated in Theorem E.1 in Dorner et al. (2023)). However, in general nonconvex settings—without additional conditions such as the Polyak–Łojasiewicz condition—even centralized SGD can exhibit divergent algorithmic sensitivity, that is, two trajectories initialized at $\theta_{t}$ and $\theta_{t}^{\prime}$ may evolve in completely opposite directions, leading to $\mathbb{E}[||\theta_{T}-\theta_{T}^{\prime}||^2]\rightarrow\infty$. This divergence makes it impossible to guarantee $\kappa_{t}\delta$-truthful action of farsighted agents in general nonconvex settings. Hence, in the revised manuscript, we have removed the claim of theoretical guarantees for the nonconvex case. Nevertheless, our results remain stronger than existing truthfulness guarantees for server-assisted federated learning—such as those in Dorner et al. (2023) which considers farsighted agents and Chakarov et al. (2024a;b) which consider myopic agents—in three key respects: (1) existing results rely on a server for payment collection and computation whereas our result can be implemented in a fully decentralized setting without a server; (2) existing results are subject to an optimization error, whereas our result ensures exact convergence; and (3) existing results rely on strong convexity assumptions, while our results hold under general convexity.

---

> ### Author Response · Authors · 2025-11-20
> **Response to Reviewer m7Yo---Part II**
>
> $\rm\color{blue}{@m7Yo\ @Weakness\ 2:}$  Thank you for the comment. We clarify that existing incentive-based truthfulness results for server-assisted federated learning in, e.g., Dorner et al. (2023); Chakarov et al. (2024a;b), cannot **simultaneously** guarantee
> $\varepsilon$-incentive compatibility and accurate convergence. Specifically, the mechanism in Dorner et al. (2023) ensures $\varepsilon$-incentive compatibility but it fails to ensure accurate convergence. According to the statement of Theorem 6.1 in Dorner et al. (2023), its convergence analysis requires the conditions:
>
> (i) $P(\exists  t\leq T: \Pi_{W}(\theta_t^s-\gamma_t\bar{m}_t)\neq \theta_t^s-\gamma_t\bar{m}_t)\in\mathcal{O}(\frac{1}{NT})$  and (ii) the boundedness of $W$, where $\theta_t^s$ is the model parameter computed by the server, $W$ is a projection set, $\gamma_t$ is the stepsize, and $\bar{m}_t$ is the average (manipulated) gradients reported by all agents.
>
> Condition (i) holds only when the projection set $W$ grows at a rate of $\mathcal{O}(T)$ (as stated in Appendix Section ''Discussion on the projection assumptions" in Dorner et al. (2023)). Meanwhile, Condition (ii) requires $W$ to be bounded, which is possible only when the number of iterations $T$ is finite (under general strongly convex objective functions). Therefore, the convergence result in Dorner et al. (2023) is subject to an error of $\mathcal{O}(\frac{1+M+\varepsilon^2}{NT})+\mathcal{O}(\frac{1}{T^2})$ which is strictly larger than $0$ because $T$ cannot be made arbitrarily large—doing so would violate Condition (ii) under general strongly convex functions. A similar issue also exists in Chakarov et al. (2024a) (see Theorem 9 and footnote 2 therein). Different from Dorner et al. (2023) and Chakarov et al. (2024a), the convergence analysis in Chakarov et al. (2024b) removes these two conditions. However, its definition  $G=\sum_{t=1}^{T}\gamma_{t}\sqrt{\mathcal{C}_{t}}$ in Theorem 5.1 implies that $\varepsilon=\mathcal{O}(G)$ is finite only when $T$ is finite, indicating that both its $\varepsilon$-incentive compatibility and convergence statement fail to hold in an infinite time horizon.
>
> To make these points clear, we have added a paragraph under Theorem 2 on page 9:
>
> **''Existing incentive-based truthfulness results for server-assisted federated learning in, e.g., Dorner
> et al. (2023); Chakarov et al. (2024a;b), cannot simultaneously guarantee $\varepsilon$-incentive compatibility and accurate convergence. Specifically, the convergence analysis in Dorner et al. (2023) requires the conditions:**
>
> **(i) $P(\exists t\leq T: \Pi_{W}(\theta_t^s-\gamma_t\bar{m}_t)\neq \theta_t^s-\gamma_t\bar{m}_t)\in\mathcal{O}(\frac{1}{NT})$ and (ii) the boundedness of $W$ (see Theorem 6.1 in Dorner et al. (2023)), where $\theta_t^s$ is the model parameter computed by the centralized server, $W$ is a projection set, $\gamma_t$ is the stepsize, and $\bar{m}_t$ is the average (manipulated) gradients reported by all agents.**
>
> **Condition (i) holds only when $W$ grows at a rate of $\mathcal{O}(T)$ (as stated in Appendix section ''Discussion on the projection assumptions" in Dorner et al. (2023)). Meanwhile, Condition (ii) requires $W$ to be bounded, which is possible only when $T$ is finite (under general strongly convex objective functions). Therefore, the convergence result in Dorner et al. (2023) is subject to an error of $\mathcal{O}(\frac{1+M+\varepsilon^2}{NT})+\mathcal{O}(\frac{1}{T^2})$ which is strictly larger than $0$ because $T$ cannot be made arbitrarily large—doing so would violate Condition (ii) under general strongly convex functions. A similar issue also exists in Chakarov et al. (2024a) (see Theorem 9 and footnote 2 therein).**
>
> **Although the convergence analysis in Chakarov et al. (2024b) removes these two conditions, its definition  $G=\sum_{t=1}^T\gamma_{t}\sqrt{\mathcal{C}_t}$ in Theorem 5.1 implies that $\varepsilon=\mathcal{O}(G)$ is finite only when $T$ is finite, indicating that both its $\varepsilon$-incentive compatibility and convergence statement fail to hold in an infinite time horizon.''**
>
> In addition,  our theoretical results for the non-convex case in the initial version were based on the myopic strategic agents setting as in Chakarov et al. (2024a;b), which consider federated learning under the assistance of a server. Under this setting, we developed new  analytical techniques to ensure both accurate convergence and $\varepsilon$-incentive compatibility in fully decentralized nonconvex optimization.
>
> However, when considering farsighted strategic agents who care about long-term rewards, guaranteeing accurate convergence and $\varepsilon$-incentive compatibility in nonconvex settings becomes challenging (the reason is detailed in the last paragraph of our responses to Weakness 1). Therefore, in the revised manuscript, we have removed all results for non-convex objective functions.

---

> > ### Author Response · Authors · 2025-11-20
> > **Response to Reviewer m7Yo---Part III**
> >
> > $\rm\color{blue}{@m7Yo\ @Weakness\ 3:}$  Thank you for the comment. Although our payment mechanism is inspired by Chakarov et al. (2024a;b) which consider  server-assisted federated learning, it has several fundamental differences: (1) our payment mechanism is implemented and computed in a fully decentralized manner, and hence, is applicable to arbitrary connected communication graphs. In contrast, Chakarov et al. compute payments at a centralized server that aggregates gradients from all agents, which restricts their approaches to a centralized communication structure; and (2) furthermore, to prevent free-riding and manipulative behaviors of agents in our payment mechanism, in the revised manuscript, we have refined our payment mechanism by shifting from gradient-magnitude differences to differences in model-parameter increments (see our response to Weakness 4 for details). This modification enables pairwise verifiability using only locally available information and represents a fundamental design departure from Chakarov et al. (2024a;b). In addition, it is worth noting that our payment mechanism attains $\varepsilon$-incentive compatibility with a finite $\varepsilon$ even in an infinite time horizon, whereas the $\varepsilon$ in Chakarov et al. (2024a;b) grows to infinity as the number of iterations tends to infinity (see Claim 23 in Chakarov et al. (2024a) and Theorem 5.1 in Chakarov et al. (2024b)). This also implies a fundamental difference between our payment design and the mechanism in Chakarov et al. (2024a;b).
> >
> > In the revised manuscript, we have added a paragraph on page 7 to discuss  these differences between our approach and the mechanism in Chakarov et al. (2024a;b).
> >
> > **''The existing approaches most closely related to ours are the payment mechanisms proposed by Chakarov et al. (2024a;b). However, there are several fundamental differences: 1) our payment mechanism is implementable in a fully decentralized manner, and hence, is applicable to arbitrary connected communication graphs, whereas the mechanisms in Chakarov et al. (2024a;b) rely on a centralized server to aggregate gradients from all agents, and thus operate only under a centralized
> > communication structure; and 2) our payment mechanism achieves $\varepsilon$-incentive compatibility with a finite $\varepsilon$ even in an infinite time horizon (see Theorem 2), whereas the incentive $\varepsilon$ in Chakarov et al. (2024a;b) becomes unbounded as the number of iterations tends to infinity (see Claim 23 in Chakarov et al. (2024a) or Theorem 5.1 in Chakarov et al. (2024b)), leading to a vanishing incentive compatibility guarantee over iterations."**

---

> ### Author Response · Authors · 2025-11-20
> **Response to Reviewer m7Yo---Part IV**
>
> $\rm\color{blue}{@m7Yo\ @Weakness\ 4:}$ Thank you for the comment. In the revised manuscript, we have refined our payment mechanism by shifting from gradient-magnitude differences to differences in model-parameter increments, enabling us to address more general attacks such as free-riding and payment manipulation.
>
> Specifically, in the revised version, we have updated the payment from agent $i$ to agent $j$ at iteration $t$ as $P_{i,t}^j=C_t(\Delta_{\theta_i,t}-\Delta_{\theta_j,t})$ with $\Delta_{\theta_i,t}\triangleq||\theta_{i,t+1}-2\theta_{i,t}+\theta_{i,t-1}||^2$ and $\Delta_{\theta_j,t}\triangleq||\theta_{j,t+1}-2\theta_{j,t}+\theta_{j,t-1}||^2$. Because $\theta_{\iota,t-1}$, $\theta_{\iota,t}$, $\theta_{\iota,t+1}$ for $\iota\in${$i, j$} are available to both agents $i$ and $j$ under Algorithm 1 (note that $\theta_{\iota,t+1}$ for $\iota\in${$i, j$} have been shared at the end of iteration $t$), the two agents can cross-verify the computed payment value, making the mechanism robust to unilateral manipulation. Moreover, since agent $i$'s model-parameter increment $\Delta_{\theta_i,t}$ is determined by both consensus error (see dynamics of $\Delta_{\theta_i,t}$ in Eq. (54)) and the (sign-indefinite) local gradients $m_{i,t}$ and $m_{i,t-1}$, even if agent $i$ shares a zero gradient, the resulting value of $\Delta_{\theta_i,t}$ is not necessarily smaller than that of its neighbors. Hence, agent $i$ cannot reliably increase its payment gains by sharing zero gradients. Meanwhile, free-riding behavior (or using $a<1$) invariably degrades agent $i$'s own reward $R_i(f_i(\theta_{i,T}))$, as it weakens the influence of agent $i$'s data in collaborative learning and leads to a worse final model for the agent itself. Hence, free-riding does not constitute a utility-improving choice for a rational agent.
>
> We have added a paragraph on page 7 in the revised manuscript to make these points clear:
>
> **''Our payment mechanism can effectively discourage agents from free-riding. Specifically, the model-parameter increment $||\theta_{i,t+1}-2\theta_{i,t}+\theta_{i,t-1}||$ of agent $i$ depends on both the consensus errors (see the dynamics of $\Delta_{\theta_i,t}$ in equation (54)) and the (sign-indefinite) local gradients. Consequently, even if agent $i$ uses a zero (or low) gradient, there is no guarantee that  $||\theta_{i,t+1}-2\theta_{i,t}+\theta_{i,t-1}||$ will be smaller than that of its neighbor $j$, meaning that leveraging zero gradients does not reliably increase agent $i$'s payment gains. In contrast, free-riding behavior (or using low gradients) invariably degrades agent $i$’s own reward $R_i(f_i(\theta_{i,T}))$, as it weakens the influence of agent $i$'s data in collaborative learning and ultimately leads to a worse final model for the agent itself. Hence, free-riding is not a utility-improving choice for a rational agent.''**
>
> $\rm\color{blue}{@m7Yo\ @Weakness\ 5:}$ We thank the reviewer for the suggestions.
>
> (1) In the revised version, we have introduced a new notation $\pi_i,i\in[N]$ to denote the eigenvalues of the matrix $W,$ while $\lambda_t$ continues to denote the stepsize.
>
> (2) Following the reviewer's suggestion, in the revised manuscript, we use the differences in model-parameter increments to represent the monetary transfer between any pair of interacting agents. Furthermore, we use $P_{i,t}^j$ to denote the payment of agent $i$ to neighboring agent $j$ and $P_{j,t}^i$ to denote the payment of agent $j$ to neighboring agent $i$, both of which are allowed to take negative values.
>
> $\rm\color{blue}{@m7Yo\ @Question\ 1:}$ Thank you for the comment. We illustrate that the mechanisms in, e.g., Chakarov et al. (2024a;b), are conceptually inspired by the VCG framework in three aspects: (1) both approaches ensure that a strategic agent bears the cost of the negative effect caused by its untruthful behavior. In VCG, an agent must pay for the impact its misreporting imposes on the global social welfare. Similarly, in Chakarov et al. (2024a;b), a strategic agent incurs a penalty that reflects the influence of its manipulated gradient update on the collaborative optimization process; (2) both approaches adopt a design that separates the global optimization objective from the mechanism component that enforces truthful behavior. In VCG, the socially optimal allocation is determined independently of the payment rule that counteracts strategic behavior. Similarly, in Chakarov et al. (2024a;b), the federated learning rule governs optimization, while a distinct payment mechanism specifically disincentivizes strategic gradient manipulation; and (3) both approaches modify each agent's utility function through an additive payment term to reshape incentives. In VCG, a payment term is added to ensure that truthful reporting maximizes the agent's utility. Similarly, in Chakarov et al. (2024a;b), an additive payment term is introduced so that any manipulation of gradient information reduces the agent's effective utility.

---

> > ### Comment · Reviewer_m7Yo · 2025-11-23
> >
> > Thank you for the reply, here are some thoughts and follow-up questions.
> >
> > - Is the updated mechanism's purpose solely to reduce free-riding, or were there other reasons to choose it?
> > - Is there a formal statement regarding incentives around free-riding?
> >
> > - How does the updated analysis in Lemma 9 account for the potential effects of $\alpha_{i,t-1}$ on the $P_{i,t}$ ?
> >
> > - Overall, I remain a bit confused about whether this paper's main contribution is supposed to be a new mechanism, or a refined analysis that could also be applicable to previously proposed mechanisms (either is fine, but this should be made clear). In particular, after looking at the quoted passage from Dorner et al. (2023), it seems like that work states that $W$ growing as $\Omega(W)$ (not $O(W)$) is a sufficient condition for their guarantee, not a necessary one. Correspondingly, while accurate convergence is not proven in previous works, it seems unclear whether that is due to the proposed mechanisms or due to less refined analysis.
> > - On an unrelated note, having looked into Dorner et al. (2023) in a bit more detail, that paper seems to claim budget balance, while your table claims the opposite.

---

> > > ### Comment · Reviewer_m7Yo · 2025-11-23
> > >
> > > Maybe it is also worth noting, that there do not exist any strongly convex and globally Lipschitz functions on $\mathbb{R}^n$, such that the first part of Theorem 1 is vacuous in its current form.

---

> ### Author Response · Authors · 2025-11-24
> **Second-round Response to Reviewer m7Yo---Part I**
>
> $\rm\color{blue}{\text{@m7Yo\ @Further\ Question\ 1:\ Is\ the\ updated\ mechanism's\ purpose\ solely\ to\ reduce\ free-riding?}}$
>
> $\rm Response$: Thank you very much for the quick response. The answer is negative. In fact, beyond mitigating potential free-riding behaviors, another key motivation for updating the mechanism is to **address payment manipulation**. Unlike existing payment mechanisms (in, e.g., Angeli \& Manfredi (2023); Dorner et al. (2023); Chakarov et al. (2024a;b)), which all assume that the  computation of payments (by the server therein) is trustworthy—a premise that may not always hold in practice (see, e.g., [r1]–[r3] for discussions on the potential untrustworthiness of servers in federated learning)—our refined payment mechanism allows two interacting agents to compute and cross-verify payments in a fully decentralized manner, and is therefore robust against payment manipulation.
>
> Specifically, as stated in our response to Weakness 4, we have refined our payment mechanism by shifting from gradient-magnitude differences to differences in model-parameter increments. This change enables us to address payment manipulation and ensure agents' honest participation. Specifically, the payment from agent $i$ to agent $j$ at iteration $t$ is calculated as $P_{i,t}^j=C_t(\Delta_{\theta_i,t}-\Delta_{\theta_j,t})$ with $\Delta_{\theta_i,t}\triangleq||\theta_{i,t+1}-2\theta_{i,t}+\theta_{i,t-1}||^2$ and $\Delta_{\theta_j,t}\triangleq||\theta_{j,t+1}-2\theta_{j,t}+\theta_{j,t-1}||^2$ (see Mechanism 1 on page 6 in the revised manuscript; note that $\theta_{\iota,t+1}$ for $\iota\in${$i,j$} has been shared at the end of iteration $t$ under Algorithm 1). This modification eliminates the possibility of manipulating the payment mechanism, because all information required to compute the payment is concurrently available to both agents $i$ and $j$ from the updates in Algorithm 1. Specifically, with both agents $i$ and $j$ having access to $\theta_{\iota,t-1}$, $\theta_{\iota,t}$, and $\theta_{\iota,t+1}$ for $\iota\in${$i,j$} under Algorithm 1, both agents can cross-verify the computed payment value $P_{i,t}^j$. As a result, manipulation of the payment mechanism becomes infeasible and our updated mechanism no longer relies on the assumption of honesty in the payment computation.
>
> Compared with existing payment mechanisms for server-assisted collaborative learning in, e.g., Angeli \& Manfredi (2023); Dorner et al. (2023); Chakarov et al. (2024a;b), our decentralized payment mechanism has two key advantages: (1) existing mechanisms require a **trusted** server for payment collection and computation, whereas our mechanism **does not rely on any trusted third party or mutual trust among participants in payment computation**;
> and (2) existing mechanisms in Angeli \& Manfredi (2023) and Dorner et al. (2023) do not provide general guarantees for budget balance (see explanations in our response to Question 5), whereas our mechanism has guaranteed **budget balance**.
>
> [r1] Xu G, Li H, Liu S, et al. VerifyNet: Secure and verifiable federated learning. *IEEE Trans. Inf. Forensics Secur.*, 2019.
>
> [r2] Buyukates B, So J, Mahdavifar H, et al. LightVeriFL: A lightweight and verifiable secure aggregation for federated learning. *IEEE J. Sel. Areas Inf. Theory*, 2024.
>
> [r3] Fraboni Y, Vidal R, Lorenzi M. Free-rider attacks on model aggregation in federated learning. *Int. Conf. Artif. Intell. Stat.*, 2021.

---

> ### Author Response · Authors · 2025-11-24
> **Second-round Response to Reviewer m7Yo---Part II**
>
> $\color{blue}{\text{@m7Yo\ @Further\ Question\ 2:\ Is\ there\ a\ formal\ statement\ regarding\ incentives\ around\ free-riding?}}$
>
> $\rm Response$: We clarify that our refined mechanism provides a mechanism-level intuition for discouraging free-riding. As discussed in the second paragraph of our response to Weakness 4, under our refined payment mechanism, an agent that reports a zero gradient cannot reliably increase its payment gain and will, in fact, always reduce its own reward $R_{i}(f_{i}(\theta))$. Therefore, free-riding is not a utility-improving strategy for a rational agent.
>
> However, providing a formal theoretical characterization of free-riding incentives is highly non-trivial. Doing so would require a substantially different problem formulation and new analytical tools, constituting a distinct research agenda that lies beyond the scope of this work  (see, e.g., [r3]–[r4], which study free-riding but do not address gradient manipulation in coordinated learning and optimization). We appreciate the reviewer's suggestion of this interesting direction for further exploring the potential of our work, and we will pursue it in future research.
>
> As stated in the contribution statements, our theoretical results mainly focus on ensuring both truthfulness and accurate convergence of decentralized learning algorithms in the presence of gradient manipulation by strategic agents. This problem formulation conceptually follows the settings in Dorner et al. (2023) and Chakarov et al. (2024a;b), which study gradient manipulation by strategic agents in server-assisted collaborative learning and do not consider free-riding incentives. (However, we emphasize that our decentralized payment mechanism and convergence guarantees differ substantially from those in Dorner et al. (2023) and Chakarov et al. (2024a;b); see the last paragraph of our response to Question 1 and our response to Question 4 for details.)
>
> [r3] Fraboni Y, Vidal R, Lorenzi M. Free-rider attacks on model aggregation in federated learning. *Int. Conf. Artif. Intell. Stat.*, 2021.
>
> [r4] Karimireddy S P, Guo W, Jordan M I. Mechanisms that incentivize data sharing in federated learning. *In Workshop Fed. Learn.: Recent Adv. New Chall. (with NeurIPS 2022)*, 2022.
>
> $\color{blue}{\text{@m7Yo\ @Further\ Question\ 3:\ How\ does\ the\ updated\ analysis\ in\ Lemma\ 9\ account\ for\ the\ potential\ effects\ of\ $\alpha_{i,t-1}$\ on\ the\ $P_{i,t}$?}}$
>
> $\rm Response$: The payment $P_{i,t-1}$ is computed based on the agent's strategic action $\alpha_{i,t-1}$ at the $(t-1)$th iteration, and $P_{i,t}$ is computed based on the agent's strategic action $\alpha_{i,t}$ at the $t$th iteration. Since the effect of $\alpha_{i,t-1}$ has already been fully captured in $P_{i,t-1}$, it does not need to—and should not—be considered again in $P_{i,t}$.

---

> ### Author Response · Authors · 2025-11-24
> **Second-round Response to Reviewer m7Yo---Part III**
>
> $\color{blue}{\text{@m7Yo\ @Further\ Question\ 4:\ This\ paper's\ main\ contribution\ is\ supposed\ to\ be\ a\ new\ mechanism,\ or\ a\ refined\ analysis}}$
>
> $\color{blue}{\text{ {}  that\ could\ also\ be\ applicable\ to\ previously\ proposed\ mechanisms?}}$
>
> $\rm Response$: Thank you for the comment. We respond to the   questions point by point as follows:
>
> **1. Both mechanism design and convergence analysis are core contributions of the paper.**
>
> Our fully decentralized payment mechanism is, to the best of our knowledge, the **first** payment mechanism implementable in a fully decentralized setting,
> without requiring the assistance of any centralized server or aggregator. This represents a significant advancement, as all existing payment mechanisms require a trusted server to collect information and compute payments. Moreover, compared with the existing payment mechanism in Dorner et al. (2023), our decentralized payment mechanism offers several significant advantages: (1) our mechanism **does not rely on any trusted third party or mutual trust among participants for payment computation**, whereas the mechanism in Dorner et al. (2023) requires a trustworthy centralized server to collect and compute payments; (2) our mechanism **is always budget-balanced**, whereas Dorner et al. (2023) do not provide a general guarantee of budget balance for their mechanism (see explanations in our response to Question 5); and (3) Our mechanism **accommodates general forms of gradient manipulation**, allowing strategic agents to both amplify gradients (i.e., $a_{i,t}\geq 1$) and inject arbitrary noise (i.e., $b_{i,t}\in\mathbb{R}$), whereas the mechanism in Dorner et al. (2023) is limited to noise-based manipulation with no gradient scaling (i.e., $a_{i,t}\equiv1$ and $b_{i,t}\in\mathbb{R}$). Given these advantages, the design of our mechanism constitutes a central contribution of this work.
>
> Furthermore, since our decentralized payment mechanism has a fundamentally different structure from existing server-assisted payment mechanisms in, e.g., Dorner et al. (2023), existing analytical techniques for server-assisted collaborative learning **cannot** be directly applied to our decentralized setting. Consequently, we must develop new analytical techniques to establish both $\varepsilon$-incentive compatibility (or truthfulness) and accurate convergence guarantees for our approach. Therefore, the refined analysis is also a substantive contribution of our work.
>
> (Continued on Part---IV)

---

> ### Author Response · Authors · 2025-11-24
> **Second-round Response to Reviewer m7Yo---Part IV**
>
> (Continued from Part---III)
>
> **2. Clarification on the discussion of Dorner et al. (2023).**
>
> Thank you for the comment. In the revised manuscript, we have updated the statements in the paragraph under Theorem 2 on page 9 to reflect that Dorner et al. (2023) provide a sufficient condition ($W$ growing as $\Omega(T)$) for convergence under general strongly convex functions. However, this is at odds with the boundedness requirement on $W$ in their Theorem 6.1 when $T$ tends to infinity (their convergence error $\mathcal{O}(\frac{1+M+\varepsilon^2}{NT})+\mathcal{O}(\frac{1}{T^2})$ in Theorem 6.1 is strictly larger than $0$ unless $T$ is allowed to approach infinity).
>
> Regarding why Dorner et al. (2023) did not give an approach to reconciling the two conditions and establishing an accurate convergence guarantee, we speculate that one reason is the projection operator introduced in their Theorem 6.1, whose treatment introduces additional inequalities (replacing exact expressions with upper or lower bounds) and, consequently, conservativeness. Another possible reason is that their mechanism is fundamentally different from ours, so the techniques we use in our convergence proof do not directly apply to their setting.
>
> To make the discussions on existing results including Dorner et al. (2023) more accurate, we have changed the wording ''they cannot simultaneously guarantee..."  to ''they do provide..." and replaced ''only when $W$ grows..." with ''they state that ... when $W$ grows...". Please see the revised statement under Theorem 2 on page 9:
>
> **''Existing incentive-based truthfulness results for server-assisted federated learning in, e.g.,  Dorner et al. (2023); Chakarov et al. (2024a;b), do not provide simultaneous guarantees for both $\varepsilon$-incentive compatibility and accurate convergence. Specifically, the convergence analysis in Dorner et al. (2023) requires two conditions:**
>
> **(i) $P(\exists t\leq T: \Pi_{W}(\theta_t^s-\gamma_t\bar{m}_t)\neq \theta_t^s-\gamma_t\bar{m}_t)\in\mathcal{O}(\frac{1}{NT})$ and (ii) the boundedness of $W$ (see Theorem 6.1 in Dorner et al. (2023)), where $\theta_t^s$ is the model parameter computed by the centralized server, $W$ is a projection set, $\gamma_t$ is the stepsize, and $\bar{m}_t$ is the average (manipulated) gradients reported by all agents.**
>
> **Moreover, in the Appendix section ''Discussion on the projection assumptions", they state that Condition (i) can be guaranteed  when $W$ grows at a rate of $\Omega(T)$ for general strongly convex functions, which is at odds with the boundedness requirement on $W$ in Condition (ii) when $T$ tends to infinity (their convergence error $\mathcal{O}(\frac{1+M+\varepsilon^2}{NT})+\mathcal{O}(\frac{1}{T^2})$ in Theorem 6.1 is strictly larger than $0$ unless $T$ is allowed to approach infinity). Therefore, they did not provide a method for ensuring that both conditions hold simultaneously under general strongly convex objectives. A similar issue also exists in Chakarov et al. (2024a) (see Theorem 9 and footnote 2 therein). Although the convergence analysis in  Chakarov et al. (2024b) removes these two conditions, its definition  $G=\sum_{t=1}^T\gamma_{t}\sqrt{\mathcal{C}_t}$ in Theorem 5.1 implies that $\varepsilon=\mathcal{O}(G)$ is finite only when $T$ is finite, indicating that both its $\varepsilon$-incentive compatibility and convergence statements fail to hold in an infinite time horizon."**
>
> $\color{blue}{\text{@m7Yo\ @Further\ Question\ 5:\ Dorner\ et\ al.\ (2023)\ seems\ to\ claim\ budget\ balance.}}$
>
> $\rm Response$: We note that the payment mechanism for multi-round SGD in federated learning in Dorner et al. (2023) is budget-balanced when **all agents are honest** (see the last sentence in Section 6 in Dorner et al. (2023)) or when **all agents adopt the same strategy**, i.e., $\alpha_{i,t}=\alpha_{j,t}$ for all $i,j\in[N]$ and $t\geq 0$ (see the last sentence in Proposition E.2 in Dorner et al. (2023)). In more general settings—such as ours—where strategic agents may choose heterogeneous actions, i.e., $\alpha_{i,t} \ne \alpha_{j,t}$ for some $i, j \in [N]$ and $t \ge 0$, they did not establish  budget balance.
>
> To ensure a more accurate discussion of the results in Dorner et al. (2023), we have added the following footnote to Table I:
>
> **''$^{c}$ Their mechanism is budget-balanced when all agents are honest (see the last sentence in their
> Section 6) or adopt the same strategy, i.e., when $\alpha_{i,t} = \alpha_{j,t}$ for all $i, j \in [N]$ and all $t \ge 0$ (see the  final sentence of their Proposition E.2)."**

---

> ### Author Response · Authors · 2025-11-24
> **Second-round Response to Reviewer m7Yo---Part V**
>
> $\color{blue}{\text{@m7Yo\ @Further\ Question\ 6:\ There\ do\ not\ exist\ any\ strongly\ convex\ and\ globally\ Lipschitz\ functions\ on\ $\mathbb{R}^{n}$,}}$
>
> $\color{blue}{\text{ {} such\ that\ the\ first\ part\ of\ Theorem\ 1\ is\ vacuous\ in\ its\ current\ form.}}$
>
> $\rm Response$: We emphasize that we **do not require** the Lipschitz continuity condition on the objective functions in strongly convex settings (as evidenced by Assumption 1 on page 7 and our convergence analysis in Appendix Section E.1 on page 36 to page 39). To make this point explicit, we have revised the statement of the first part of Theorem 1 in the manuscript as follows:
>
> **''Theorem 1 (Convergence rate). We denote $\theta^{*}$ as a solution to the problem in equation 1. Under our Mechanism 1 and the conditions in Lemma 1, for any $i\in[N]$, $\delta>0$, and $T\geq0$, the following results hold for Algorithm 1 in the presence of strategic behaviors:**
>
> **(i) if $F(\theta)$ is $\mu$-strongly convex (not necessarily Lipschitz continuous), then we have**
>
> $\mathbb{E}[||\theta_{i,T}-\theta^{*}||^2]\leq \mathcal{O}\left(\frac{H^{2}(\sigma^2+\delta^{2})}{\mu(1-\rho)^2(T+1)^{v}}\right);$ **''**

---

> > ### Comment · Reviewer_m7Yo · 2025-11-25
> >
> > I would suggest to rephrase Assumption 1, as the current phrasing does not really make it clear that Lipschitzness is not required for the strongly convex case.
> >
> > Regarding the first point: Wouldn't that simply shift the burden towards honestly reporting parameters $\theta$, which would be similarly hard to guarantee in practice?
> >
> > Regarding the effect of $\alpha_{t-1}$, I am not fully convinced by your argument: As participants gradients and increments depend on $\theta$, a manipulation at time $t-1$ can affect the magnitude of a player's penalty at time $t$. How do you rule that out / account for that?
> >
> > In the proof of Lemma 9, you write that for strongly convex losses, the parameters $\theta$ are always contained in a compact set $\Theta$. What is the precise formal statement here? It seems like for any fixed compact set $\Theta$ this would clearly not be true with probability one when the gradient noise is unbounded (e.g. Gaussian), which seems to be allowed under Assumption 1.
> >
> > Lastly, while the authors do not seem to be stating this explicitly, a brief look at the structure of the mechanism from Dorner et al. (2023) seems to indicate that their mechanism is budget balanced in general (as penalties are redistributed across players).

---

> ### Author Response · Authors · 2025-11-25
> **Third-round Response to Reviewer m7Yo---Part I**
>
> $\color{blue}{\text{@m7Yo\ @Comment\ 1:\ I\ would\ suggest\ to\ rephrase\ Assumption\ 1,\ as\ the\ current\ phrasing\ does\ not\ really\ make}}$
>
> $\color{blue}{\text{ {} it\ clear\ that\ Lipschitzness\ is\ not\ required\ for\ the\ strongly\ convex\ case.}}$
>
> $\rm Response$: Thank you for the comment. In the revised manuscript, we have explicitly indicated that Lipschitz continuity is not required on the objective functions in the strongly convex case. The revised Assumption 1 becomes:
>
> **''Assumption 1. ...... . In addition, for a general convex $f_{i}(\theta)$, we assume that $f_{i}(\theta)$ is $L_{f,i}$-Lipschitz continuous. However, this assumption is not required for a strongly convex $f_{i}(\theta)$.''**
>
> $\color{blue}{\text{@m7Yo\ @Comment\ 2:\ Regarding\ the\ first\ point:\ Wouldn't\ that\ simply\ shift\ the\ burden\ towards\ honestly}}$
>
> $\color{blue}{\text{ {} reporting\ parameters\ $\theta$,\ which\ would\ be\ similarly\ hard\ to\ guarantee\ in\ practice?}}$
>
> $\rm Response$: The answer is negative. As we clearly explain in lines 205–215 of our manuscript, manipulating model parameters **does not** offer any meaningful strategic advantage to an agent. Consequently, a rational agent has no incentive to manipulate the shared model parameters.
>
> Moreover, Dorner et al. (2023) considers manipulation of the form $m_{i,t}=g_{i}(\theta_{t}^{s})+b_{i,t}\xi_{i,t}$ which explicitly relies on the **true model** parameter $\theta_{t}^{s}$ supplied by a **trustworthy centralized server**, rather than a manipulated model parameter $\theta_{i,t}^{\prime}$ (see the first sentence of the second paragraph in Section 6.1 in Dorner et al. (2023) for details). This alignment supports the reasonableness of our approach.
>
> $\color{blue}{\text{@m7Yo\ @Comment\ 3:\ Regarding\ the\ effect\ of\ $\alpha_{i,t-1}$,\ I\ am\ not\ fully\ convinced\ by\ your\ argument:\ As\ participants}}$
>
> $\color{blue}{\text{ {} gradients\ and\ increments\ depend\ on\ $\theta_{i,t}$,\ a\ manipulation\ at\ time\ $t-1$\
> can\ affect\ the\ magnitude\ of\ a\ player's}}$
>
> $\color{blue}{\text{ {} penalty\ at\ time\ $t$.\ How\ do\ you\ rule\ that\ out/account\ for\ that?}}$
>
> $\rm Response$: Not using the past action $\alpha_{i,t-1}$ to influence the current penalty $P_{i,t}$ is **well justified** in the literature, including  Chakarov et al. (2024a) (see the proof of Claim 21 on page 14), Chakarov et al. (2024b) (see the proof of Claim E.3 on page 22), and Dorner et al. (2023) (see the first equation in the proof Proposition E.2 on page 32), which the reviewer has repeatedly cited in support of the reviewer's position.
>
> In fact, in gradient descent algorithms, a different action value $\alpha_{t-1}$ clearly leads to different subsequent states for all times $t,t+1,\cdots,T$. Therefore, continuing to penalize $\alpha_{t-1}$ throughout all iterations $t,t+1,\cdots,T$ is clearly not reasonable.

---

> ### Author Response · Authors · 2025-11-25
> **Third-round Response to Reviewer m7Yo---Part II**
>
> $\color{blue}{\text{@m7Yo\ @Comment\ 4:\ In\ the\ proof\ of\ Lemma\ 9,\ you\ write\ that\ for\ strongly\ convex\ losses,\ the\ parameters\ $\theta$\ are}}$
>
> $\color{blue}{\text{ {} always\ contained\ in\ a\ compact\ set\ $\Theta$.\ What\ is\ the\ precise\ formal\ statement\ here?}}$
>
> $\rm Response$: Following the reviewer's suggestion, we have removed the inaccurate statement of $\theta_{i,t}\in\Theta$ in the proof of Lemma 9. We note that the updated proof **does not** require the Lipschitz continuity assumption on the objective functions in the strongly convex case.
>
> Specifically, $\theta_{i,t}\in\Theta$ was only used to  prove the boundedness of $\mathbb{E}[||g_{i}(\theta_{i,t})||^2]$ in the absence of Lipschitz continuity assumption in the strongly convex case. In the revised manuscript, we have refined the proof by leveraging the inequality $\mathbb{E}[||g_{i}(\theta_{i,t})||^2]= \mathbb{E}[||g_{i}(\theta_{i,t})-\nabla f_{i}(\theta_{i,t})||^2]+\mathbb{E}[||\nabla f_{i}(\theta_{i,t})-\nabla f_{i}(\theta_{i}^{\*})||^2]$. Specifically, using the $H_{i}$-Lipschitz continuity of gradients $\nabla f_{i}(\theta)$ and the bounded gradient-estimate variance $\sigma^2$, we arrive at $\mathbb{E}[||g_i(\theta_{i,t})||^2]\leq\sigma^2+2H_i^2\mathbb{E}[||\theta_{i,t}-\theta^*||^2]+2H_i^2||\theta_i^{\*}-\theta^{\*}||^2$. It is clear that $\sigma^2$ and $||\theta_{i}^{\*}-\theta^{\*}||^2$ are naturally bounded. Moreover, since $\theta_{i,t}$ is  the model parameter generated by the standard decentralized SGD in the absence of strategic manipulation, we have $\mathbb{E}[||\theta_{i,t}-\theta^{\*}||^2]\leq \mathcal{O}(1)$ for any $t\geq 0$ based on existing results in [r1] and [r2] (see, e.g., Proposition 3 in [r1] for details). Combining the natural boundedness of $\sigma^2$ and $2H_{i}^2||\theta_{i}^{\*}-\theta^{\*}||^2$ with the relation $\mathbb{E}[||\theta_{i,t}-\theta^{\*}||^2]\leq \mathcal{O}(1)$ establishes the boundedness of $\mathbb{E}[||g_{i}(\theta_{i,t})||^2]$. It can be seen that our refined proof requires **neither** the condition of $\theta_{i,t}\in\Theta$ **nor** Lipschitz continuity of objective functions.
>
> [r1] Fallah A, Gürbüzbalaban M, Ozdaglar A, et al. Robust distributed accelerated stochastic gradient methods for multi-agent networks. *J. Mach. Learn. Res.*, 2022.
>
> [r2] Pu S, Olshevsky A, Paschalidis I C. A sharp estimate on the transient time of distributed stochastic gradient descent. *IEEE Trans. Autom. Control*, 2021.
>
> $\color{blue}{\text{@m7Yo\ @Comment\ 5:\ Lastly,\ while\ the\ authors\ do\ not\ seem\ to\ be\ stating\ this\ explicitly,\ a\ brief\ look\ at\ the}}$
>
> $\color{blue}{\text{ {} structure\ of\ the\ mechanism\ from\ Dorner\ et al.\ (2023)\ seems\ to\ indicate\ that\ their\ mechanism\ is\ budget\ balanced}}$
>
> $\color{blue}{\text{ {} in\ general\ (as\ penalties\ are\ redistributed\ across\ players).}}$
>
> $\rm Response$: We can mark Dorner et al. (2023) as budget-balanced in our Table I. Our original concern was that in its Proposition E.1 (as well as its formula under Eq. (7) in Section 6.1), the payment is given as $P_i=-\sum_{t=1}^TC_t ||m_t^i-\bar{m}_t||^2$
>
> $+\frac{1}{N-1}\sum_{k \neq i}\sum_{t=0}^TC_t||m_t^k-\bar{m}_t||$. With different starting indices in the two cumulative sums on the right hand side of this expression (one starting at $t=1$ and the other at $t=0$), the budget could become unbalanced at $t=0$ (note that $m_0^k\neq \bar{m}_0$ even if all agents are honest at $t=0$).
>
> We would appreciate it if the reviewer could clarify whether the budget in Dorner et al. (2023) is balanced at $t=0$, from the authors' perspective.
>
> ${}$
>
> ${}$
>
> ${}$
>
>
> We would be most grateful for the opportunity to address any further questions, should you be open to reconsidering your final rating, and we sincerely appreciate your interest in our work.

---

> > ### Comment · Reviewer_m7Yo · 2025-11-26
> >
> > - Regarding honest reporting of the parameters, you write "manipulating model parameters does not provide a clear strategic benefit", but if I understand your mechanism correctly, a player misreporting their parameters to be constant over time while actually updating them could obtain a strong model, while collecting large payments from honest players. I do not think this is a large weakness on its own, as freeriding is also possible in many other previously suggested mechanisms, but strongly disagree with your statement that there is no clear incentive for manipulating shared parameters.
> >
> > - Regarding the effects of $\alpha_{t-1}$: I am not saying that one *should* use the action $\alpha_{t-1}$ to influence the penalty $P_t$. My concern is that due to the explicit use of earlier parameters $\theta_{t},\theta_{t-1}$ in your penalty, $\alpha_{t-1}$ potentially has an indirect effect on $P_t$ that your proof does not seem to consider. As the the gradient magnitude at time $t$ also depends on $\theta_t$, it is plausible that previously proposed methods have the same issue, but that is out of the scope of this review.
> >
> > - Regarding the Lipschitz Continuouity: Proposition 3 in Fallah et al. seems to be concerned with constant (and sufficiently small step sizes). Is it straightforward to extend their proof to your setting with polynomially decaying step sizes?
> >
> > - Regarding the budget balance of Dorner et al (2023), I see what you mean. Given that the last sentence in their proof does not seem to make sense when both sums are not taken over the same domain (the mechanism written down would not actually be budget balanced even when all players play the same strategy), I am inclined to believe that the t=1 vs t=0 inconsistency is a typo.

---

> ### Author Response · Authors · 2025-11-27
> **Fourth-round Response to Reviewer m7Yo**
>
> $\color{blue}{\text{@m7Yo\ {}\ Comment\ on\ player\ misreporting\ constant\ parameters\ to\ collect\ payment.}}$
>
> $\rm Response$: Misreporting its model parameter as a constant while actually updating locally does not provide an agent with a strong model in decentralized learning. This is because, if agent $i$ reports a fake constant model parameter, the decentralized learning dynamics reduce to a standard pinning control (leader-follower consensus) problem. As a result, the models of honest agents that incorporate this fake constant will be pulled to converge toward the reported fake constant value [r1]-[r2].
>
> [r1] Yu W, Chen G, Lu J, et al. Synchronization via pinning control on general complex networks. *SIAM J. Control Optim.*, 2013.
>
> [r2] Wang X F, Chen G. Pinning control of scale-free dynamical networks. *Physica A: Statistical Mechanics and its Applications*, 2022.
>
> Furthermore, local updates alone lead only to convergence to the agent’s local optimum, which is clearly a weak model since it directly contradicts the purpose of collaborative learning—that is, obtaining an optimal model fitted to all agents' data. Such manipulation effectively ignores the contributions of other agents' data to agent $i$’s local model, which contradicts both the free-riding motivation and our problem formulation, as well as those of Dorner et al. (2023) (and even in  more general settings such as [r3]).
>
> Therefore, when agent $i$ reports a fake constant model parameter, both its locally computed model and the neighbors' models it influences (which are pulled to converge toward the fake constant) will fail to constitute a meaningful model, despite any early-stage payment gains. Since the final utility depends on both the quality of the final model and the payment gains, this strategy provides no clear strategic advantage.
>
> **[r3] Dorner F E. Algorithmic collusion: a critical review. *arXiv preprint arXiv:2110.04740*, 2021.**
>
> $\color{blue}{\text{@m7Yo\ {}\ Comment\ on\ indirect\ effect\ of\ $\alpha_{t-1}$\ on\ $P_t$.}}$
>
> $\rm Response$: At time $t$, there is no need to explicitly consider historical actions prior to $t$. In fact, all previous actions influence the current state $\theta_t$, but their effects are already captured in $\theta_t$ itself. Therefore, there is no need to account for them again at the current time $t$. This is why in the classic gradient descent algorithm:
>
> $\theta_{t+1} = \theta_t - \lambda_t \cdot \text{gradient}_t,$
>
> when calculating $\theta_{t+1}$, there is no need to explicitly account for  gradients before time $t$ on the right-hand side of the update rule, since $\theta_t$ already implicitly incorporates the cumulative influence of all previous gradients.
>
> $\color{blue}{\text{@m7Yo\ {}\ Comment\ on\ if\ a\ decaying-stepsize\ can\ be\ used.}}$
>
> $\rm Response$: Convergence of the standard decentralized SGD under a decaying stepsize is a classic result in decentralized optimization. In fact, both of the provided references explicitly discuss convergence under this setting. For example, in [r4] (Fallah et al.), Table 1 presents convergence results specifically for the decaying-stepsize case. Similarly, [r5] focuses exclusively on decaying stepsizes (see, e.g., Lemma 8).
>
> We list the two references again below for your convenience:
>
> [r4] Fallah A, Gürbüzbalaban M, Ozdaglar A, et al. Robust distributed accelerated stochastic gradient methods for multi-agent networks. *J. Mach. Learn. Res.*, 2022.
>
> [r5] Pu S, Olshevsky A, Paschalidis I C. A sharp estimate on the transient time of distributed stochastic gradient descent. *IEEE Trans. Autom. Control*, 2021.
>
> $\color{blue}{\text{@m7Yo\ {}\ Comment\ on\ the\ budget\ balance\ of\ Dorner\ et\ al\ (2023).}}$
>
> $\rm Response$: We have marked Dorner et al. (2023) as budget balanced in the revised version. Thank you very much for the clarification.

---

> > ### Comment · Reviewer_m7Yo · 2025-11-27
> >
> > - The model not converging to a good one does not really matter to an agent that collects large amounts of payments by manipulating its updates. In particular, under your proposed mechanism, it seems like agents that have no interest at all in the final trained model would still be incentivized to participate but report constant parameters to collect payments.
> >
> > - Could you summarize the high-level structure of your proof argument again, please? In my understanding, you
> >    - a) lower bound the difference of payments at time $t$ an agent receives when either being honest or manipulating the gradients at time $t$ (Equation 65)
> >    - b) upper bound the effects of manipulation at time $t$ on the agent's final reward (Equation 68 and below).
> >    - c) Combine these two to conclude that agents are incentivized to be honest.
> >
> >   My issue lies with c) What matters is not that the agent's penalty at time $t$ outweighs the final reward, but rather that the difference of penalties minus received payments caused by manipulation at time $t$ outweighs the gains in terms of reward. This seems to either require an explicit analysis of the sum, or an argument why manipulation at time $t$ does not affect the expected penalty/payment at time $t+1$
> >
> > - "Convergence of the standard decentralized SGD under a decaying stepsize is a classic result in decentralized optimization", Yes, but the statement of Proposition 3 in Fallah et al. you cite is substantially more specific. It is not clear whether/how the technical claim you make in the proof of Lemma 9 would follow from  "Convergence of the standard decentralized SGD under a decaying stepsize" (and if that were the case, why would you cite Proposition 3 rather than the other claims you mentioned in the reply in the proof)?

---

> > > ### Author Response · Authors · 2025-11-30
> > > **Fifth-round Response to Reviewer m7Yo**
> > >
> > > $\color{blue}{\text{@m7Yo\ {}\ Common\ on\ reporting\ a\ constant\ value\ for\ payment\ gains.}}$
> > >
> > > $\rm Response$: This is not true. In the revised manuscript, we   rigorously prove  that even if an agent falsely reports a constant value for its model parameter—which would place it in the most advantageous position to obtain payments from its neighbors under our payment Mechanism 1—its cumulative payment still remains finite (see Lemma 17 in Appendix Section G on page 51 to page 54). Moreover, this cumulative payment depends on external factors such as the network structure and the properties of the  functions  of its neighbors, none of which the agent can accurately predict or control. Consequently, since an agent's net utility is determined by both the quality of the final learned model and the payments it receives, a strategic agent cannot guarantee that its net utility will be maximized by ignoring the learned model's quality and focusing solely on manipulating its reported model parameters to increase payments. Therefore, reporting a constant value provides **NO** clear strategic advantage.
> > >
> > > $\color{blue}{\text{@m7Yo\ {}\ Comment\ on\ the\ structure\ of\ the\ proof\ and\ the\ requirement\ of\ analysis\ of\ cumulative\ payment.}}$
> > >
> > > $\rm Response$: We clarify that the reviewer's summary of the high-level structure of the proof of Lemma 9 (i.e., Lemma 1 in the main text) is correct. However, we emphasize that Lemma 9 only analyzes the action of each agent $i$ at each iteration. The question raised by the reviewer—namely, whether the final reward $R_{i}(f_{i}(\theta_{i,T+1}))$ outweighs the penalty term $-K||\theta_{i,T+1}-\theta^{*}||^2$ minus the cumulative payments $\sum_{t=0}^{T}P_{i,t}(\alpha_{i,t},h_{i,t})$—**has already** been rigorously addressed in Theorem 2. (We note that the reviewer has misinterpreted the reward as penalties minus received payments, a misunderstanding that likely arises from limited familiarity with both the established literature and the basic conceptual logic.)
> > >
> > > Specifically, Theorem 2 proves that the cumulative gain of a strategic agent from persistent manipulation is always bounded by $\varepsilon$ even as the number of iterations tends to infinity, i.e.,
> > >
> > > $\mathbb{E}\left[R_{i}(f_i(\theta_{i,T+1}^\prime))-K||\theta_{i,T+1}^\prime-\theta^*||^2-\sum_{t=0}^TP_{i,t}(\alpha_{i,t},h_{-i,t})\right]$
> > >
> > > $-\mathbb{E}\left[R_{i}(f_i(\theta_{i,T+1}))-K||\theta_{i,T+1}-\theta^{*}||^2-\sum_{t=0}^TP_{i,t}(h_{i,t},h_{-i,t})\right]\leq \varepsilon,$
> > >
> > > where the first term on the left-hand side represents the final net utility of agent $i$ under its persistent manipulation over all iterations, while the second term represents the final net utility of agent $i$ under truthful participation.
> > >
> > > Theorem 2 clearly shows that even if an agent $i$ persistently manipulates at every iteration, its cumulative gain can only exceed that of truthful participation by at most $\varepsilon$. This result holds over an infinite time horizon (as evidenced by our proof of Theorem 2 in Appendix F). We believe that this crucial result has been overlooked by the reviewer.
> > >
> > > $\color{blue}{\text{@m7Yo\ {}\ Comment\ on\ the\ reference\ for\ convergence\ under\ a\ decaying\ stepsize.}}$
> > >
> > > $\rm Response$: In the revised manuscript, we have explicitly clarified that the decaying-stepsize convergence result of standard decentralized SGD can be found in Lemma 8 of [r1].
> > >
> > >
> > > [r1] Pu S, Olshevsky A, Paschalidis I C. A sharp estimate on the transient time of distributed stochastic gradient descent. *IEEE Trans. Autom. Control*, 2021.

---

### Official Review · Reviewer_PjnL · 2025-10-27

**Soundness:** 3
**Presentation:** 3
**Contribution:** 3
**Rating:** 4
**Confidence:** 4

**Summary:**

This paper introduced the incentive mechanism for decentralized learning, so that all nodes send the true stochastic gradient and do not skew it to maximize their reward. Then, in Theorem 1, this paper shows that using the proposed method, the optimal strategy to maximize the reward is sending the true gradient.

**Strengths:**

* The paper is well written and easy to follow. The reason to introduce the incentive into decentralized learning is clear.
* Roughly speaking, in Theorem 1, this paper shows that using the proposed method, the optimal strategy to maximize the reward becomes that nodes send the correct gradient. This result fits the motivation explained tin he introduction.

**Weaknesses:**

* The reviewer feels that introducing the incentive mechanism for decentralized learning is interesting, but how can we guarantee that the nodes follow this mechanism? For instance, the set of actions, Eq. (3), seems to be limited, and there would be other choices. For instance, ignoring the parameters received from neighboring nodes in Eq. (2) or increasing $w_{ii}$ and decreasing $w_{ij}$ would also be possible to increase the reward. Besides this, there is also some degree of freedom in the choice of reward function, $R_i$, and nodes can use different reward functions. How can we guarantee that all nodes use the same reward function? The statement of Theorem 1 only holds when nodes follow this setting. The reviewer feels that there remains a possibility that nodes can work maliciously to maximize the accuracy of their dataset. This is my main concern for this paper.
* It is a bit hard to understand the intuition of Eq. (4). Why do we need to minimize $\sum_t P_{i,t}$? As far as I understand, this paper considered the case where each node tried to minimize its own loss function. Thus, it is natural that Eq. (4) is designed so that Eq. (4) increases as the reward increases. However, it is unclear why Eq. (4) is designed so that Eq. (4) increases as $\sum_t P_{i,t}$ decreases.
* In Theorem 1, the dependence on other parameters, $L_f, H, \sigma, \rho$, is hidden in the convergence rate. Can the authors show the convergence rate in Theorem 1 more precisely?
* I think it is not a critical weakness, but assuming that $f_i$ is Lipschitz continuous is a bit stronger compared with the many decentralized learning papers. It should be clarified.

**Questions:**

See the weakness section.

---

> ### Author Response · Authors · 2025-11-20
> **Response to Reviewer PjnL---Part I**
>
> $\rm\color{orange}{@PjnL\ @Weakness\ 1:}$ We truly thank the reviewer for the suggestions that helped us further improve the proposed mechanism. We answer the questions point by point as follows (note that Eq. (2) corresponds to Algorithm 1 and Eq. (3) corresponds to Eq. (2) in the revised manuscript):
>
> **1. Regarding (maliciously) local learning to maximize the accuracy of its own dataset**
>
> In the revised manuscript, we have included an additional term $-K||\theta_{i,T+1}-\theta^{\*}||^2$ in each agent $i$'s utility in Eq. (3) to represent the benefit for agents to participate in collaborative learning rather than training solely on their own local data. This term is a mathematical abstraction of the performance loss that commonly arises in practical decentralized learning applications. For example, in decentralized medical learning [r1],  hospitals are geographically separated and each serves a distinct patient population, so the data available at any single hospital are inherently limited and biased toward its own caseload. As a result, hospitals must engage in collaborative learning and leverage knowledge learned from models trained at other hospitals to improve their local diagnostic models. Relying solely on local data leads to inferior diagnostic performance, inducing a natural performance penalty for deviating from the cross-hospital shared model, which can be modeled as $-K||\theta_{i,T+1}-\theta^{*}||^2$. Similar phenomena arise in other decentralized settings such as multi-robot collaborative learning [r2] and distributed sensor fusion [r3], where robots operating in different regions observe heterogeneous local environments, and relying only on local observations results in degraded global task performance.
>
> We rigorously establish that, even after incorporating this term, our approach continues to guarantee both truthfulness and the original convergence properties.
>
> [r1] Warnat-Herresthal S, Schultze H, Shastry K L, et al. Swarm learning for decentralized and confidential clinical machine learning. *Nature*, 2021.
>
> [r2] Yu J, Vincent J A, Schwager M. DiNNO: Distributed neural network optimization for multi-robot collaborative learning. *IEEE Robot. Autom. Lett.*, 2022.
>
> [r3] Üney M, Mulgrew B, Clark D E. A cooperative approach to sensor localisation in distributed fusion networks. *IEEE Trans. Signal Process.*, 2015.
>
> **2. Regarding how to ensure that agents follow our decentralized payment mechanism**
>
> First, since our decentralized payment mechanism allows agents to obtain positive payment gains (i.e., when $P_{i,t}^{j}<0$, agent $i$ receives a payment from its neighbor $j$, thereby increasing its utility), agents have a natural incentive to participate in our decentralized payment mechanism.
>
> Furthermore, in the revised manuscript, we have refined our payment mechanism by shifting from gradient-magnitude differences to differences in model-parameter increments. This change enables us to address payment manipulation and ensure agents' honest participation. Specifically, the  payment from agent $i$ to agent $j$  at iteration $t$ is calculated as $P_{i,t}^j=C_t(\Delta_{\theta_i,t}-\Delta_{\theta_j,t})$ with $\Delta_{\theta_i,t}\triangleq||\theta_{i,t+1}-2\theta_{i,t}+\theta_{i,t-1}||^2$ and $\Delta_{\theta_j,t}\triangleq||\theta_{j,t+1}-2\theta_{j,t}+\theta_{j,t-1}||^2$ (see Mechanism 1 on page 6 in the revised manuscript, note that  $\theta_{\iota,t+1}$ for $\iota\in${$i, j$} have been shared at the end of iteration $t$ under Algorithm 1). This modification eliminates the possibility of manipulating the payment mechanism, because all information required to compute the payment is concurrently available to both agents $i$ and $j$ from the updates in Algorithm 1. Specifically, with both agents $i$ and $j$ having access to $\theta_{\iota,t-1}$, $\theta_{\iota,t}$, and $\theta_{\iota,t+1}$ for $\iota\in${$i, j$} under Algorithm 1 (note that $\theta_{\iota,t+1}$ has been shared at the end of iteration $t$), both agents can cross-verify the computed payment value $P_{i,t}^j$. As a result, manipulation of the payment mechanism becomes infeasible and our updated mechanism no longer relies on the assumption of honesty in the payment mechanism.
>
> (Continued on Part---II)

---

> > ### Author Response · Authors · 2025-11-20
> > **Response to Reviewer PjnL---Part II**
> >
> > $\rm\color{orange}{@PjnL\ @Weakness\ 1:}$
> >
> > **3. Regarding other actions beyond the action space in Eq. (2) (e.g., the suggested choices of ignoring neighbors' model parameters or changing consensus weights)**
> >
> > We agree that agents may have other possible action choices. However, as we stated in the paper,  we focus on   untruthful behaviors that can **directly and effectively** benefit strategic agents in decentralized learning and optimization. The suggested   possible actions do not provide a clear strategic benefit. For example, ignoring neighbors' model parameters or modifying the consensus weights (i.e., increasing $w_{ii}$ and decreasing $w_{ij}$) essentially reduces the impact of the information that agent $i$ receives from its neighbors. This reduction weakens the advantages that collaborative learning can provide to agent $i$ (see the examples provided in Point 1 of this response, or the benefits of collaborative learning in increasing the effective data-sample size and thereby reducing gradient-noise variance as discussed in [r4]). Therefore, ignoring neighbors' model parameters or modifying the consensus weights is not a benefit-improving choice for a rational agent.
> >
> > In addition, we would like to mention that in our action space in Eq. (2), the term $b_{i}\xi_{i}$ can capture a general class of agents' strategic behaviors. For example, with different $b_{i}\xi_{i}$, agent $i$'s action space $\mathcal{A}_{i}$ can represent scenarios involving corrupted or noisy gradients, such as non-malicious faults or differential-privacy noise [r5], adversarial/Byzantine attacks [r6], and inexact gradients [r7].
> >
> > [r4] Wu W, Li Z, Zhao Y, et al. Decentralized online learning: Take benefits from others’ data without sharing your own to track global trend. *ACM Trans. Intell. Syst. Technol.*, 2022.
> >
> > [r5] Ding T, Zhu S, He J, Chen C, and Guan X. Differentially private
> > distributed optimization via state and direction perturbation in multiagent
> > systems. *IEEE Trans. Autom. Control* 2022.
> >
> > [r6] Yang Z, Gang A, and Bajwa W U. Adversary-resilient distributed and
> > decentralized statistical inference and machine learning: An overview of recent advances under the Byzantine threat model, *IEEE Signal Process. Mag.*, 2020.
> >
> > [r7] Pu S and Nedić A. Distributed stochastic gradient tracking methods.*Math. Program.*, 2021.
> >
> > **4. Regarding the choice of $R_{i}(f_{i}(\theta))$**
> >
> > We allow different agents to have different reward functions, i.e., $R_{i}\neq R_{j}$. However, we would like to note that modifying $R_{i}$ does not provide any strategic advantage to agent $i$, because the reward function $R_{i}(f_{i}(\theta))$ merely serves as a measurement scale for evaluating agent $i$'s final model performance. Specifically, if agent $i$'s final model $\theta_{i,T+1}^{\prime}$ performs better than $\theta_{i,T+1}$ on its own dataset, then we have $f_{i}(\theta_{i,T+1}^{\prime})<f_{i}(\theta_{i,T+1})$. Any reasonable reward function $R_{i}$ (or any alternative form $R_{i}^{\prime}$) shows this improvement, that is, $R_{i}(f_{i}(\theta_{i,T+1}^{\prime}))>R_{i}(f_{i}(\theta_{i,T+1}))$ (or $R_{i}^{\prime}(f_{i}(\theta_{i,T+1}^{\prime}))>R_{i}^{\prime}(f_{i}(\theta_{i,T+1}))$). Therefore, changing $R_{i}$ only rescales the same underlying performance metric and does not provide agent $i$ with any strategic benefits.
> >
> > To avoid confusion, we have added a sentence on page 5 in the revised manuscript to explicitly state that we allow different agents to have different reward functions.

---

> ### Author Response · Authors · 2025-11-20
> **Response to Reviewer PjnL---Part III**
>
> $\rm\color{orange}{@PjnL\ @Weakness\ 2:}$ Thank you for the comments. We clarify that all existing incentive-based approaches for collaborative learning ensure truthfulness by incorporating a payment or penalty term, which affects an agent's net utility. Evidently, without any payments or constraints, a self-interested agent can engage in unrestricted strategic manipulation to reduce its own loss and increase its rewards, which would significantly distort the collaborative learning process. In Eq. (4), we include the total payment in each agent's net utility (i.e., the agent's final total payoff) to reflect this natural influence of such incentive mechanisms. Namely, a strategic agent seeking to maximize its rewards must also account for the payments incurred by its manipulative actions. This explains why a payment mechanism can enforce truthfulness in collaborative learning. Consequently, for a rational agent, the objective becomes maximizing its final payoff—namely, the net utility defined in Eq. (4).
>
> To clarify this point, we have added a paragraph under Eq. (3) on page 6 in the revised manuscript (note that Eq. (4) becomes Eq. (3) in the revised manuscript):
>
> **''We note that all existing incentive-based approaches for collaborative learning ensure truthfulness by incorporating a payment or penalty term into each agent’s net utility. Without such payments, a self-interested agent can freely manipulate its gradients to reduce its own loss and increase its rewards, thereby distorting the collaborative learning process. Therefore, equation 3 includes the cumulative payments of each agent in its net utility. Accordingly, a rational agent must consider both its rewards and payments when maximizing its net utility."**
>
> $\rm\color{orange}{@PjnL\ @Weakness\ 3:}$ Following the reviewer's suggestion, in the revised manuscript, we have provided a more explicit convergence rate in Theorem 1. The updated Theorem 1 is given as follows:
>
> **''Theorem 1 (Convergence rate). We denote $\theta^*$ as a solution to the problem in equation 1. Under our Mechanism 1 and the conditions in Lemma 1, for any $i\in[N]$, $\delta>0$, and $T\geq0$, the following results hold for Algorithm 1 in the presence of strategic behaviors:**
>
> **(i) if $F(\theta)$ is $\mu$-strongly convex (not necessarily Lipschitz continuous), then we have**
>
> $\mathbb{E}[||\theta_{i,T}-\theta^*||^2]\leq \mathcal{O}\left(\frac{H^2(\sigma^2+\delta^2)}{\mu(1-\rho)^2(T+1)^v}\right)$;
>
> **(ii) if $F(\theta)$ is general convex, then we have**
>
> $\frac{1}{T+1}\sum_{t=0}^T\mathbb{E}[F(\theta_{i,t})-F(\theta^*)]\leq \mathcal{O}\left(\frac{H^2(\sigma^2+L_f^2+\delta^2)}{(1-\rho)^2(T+1)^{1-v}}\right)$,
>
> **where $H$, $\sigma$, and $L_f$ are from Assumption 1 and** $\rho=\max${$|\pi_2|,|\pi_N|$}$<1$ **is from Assumption 2."**
>
> It can be seen that a larger strong convexity coefficient $\mu$ and a smaller $\rho$ (corresponding to a better-connected communication topology) lead to a higher convergence rate. In contrast, larger values of the smoothness coefficient $H$, Lipschitz constant $L_f$, truthfulness parameter $\delta$, and gradient noise variance $\sigma$ lead to a lower convergence rate.
>
> $\rm\color{orange}{@PjnL\ @Weakness\ 4:}$ Thank you for the constructive comment. In the revised manuscript, we have removed the assumption on the $L_f$-Lipschitz continuity of objective functions in the strongly convex case in Assumption 1 on page 7. We retain the Lipschitz continuity assumption in the general convex case. We would like to explain that the Lipschitz continuity assumption in the general convex settings is standard in existing decentralized/collaborative learning literature to ensure convergence, even without truthfulness considerations (see, e.g., [r4]-[r10]). Given that our goal is to ensure both accurate convergence and truthfulness in decentralized learning, the Lipschitz continuity assumption is necessary for our analysis in the general convex setting.
>
> [r4] Xiao L. Dual averaging method for regularized stochastic learning and online optimization, *Adv. Neural Inf. Process. Syst.*, 2009 (see Theorem 2).
>
> [r5] Jakovetić D, et al. Fast distributed gradient methods. *IEEE Trans. Autom. Control*, 2014 (see Assumption 3).
>
> [r6] Ma C, et al. Adding vs. averaging in distributed primal-dual optimization. *Int. Conf. Mach. Learn.*, 2015 (see Theorem 8).
>
> [r7] Koloskova A, et al. Decentralized deep learning with arbitrary communication compression. *Int. Conf. Learn. Represent.*, 2019 (see Assumption 2).
>
> [r8] Li X, et al. On the convergence of FedAvg on non-IID data.*Int. Conf. Learn. Represent.*, 2020 (see Assumption 4).
>
> [r9] Sun T, et al. Adaptive random walk gradient descent for decentralized optimization. *Int. Conf. Learn. Represent.*, 2022 (see Assumption 2).
>
> [r10] Beznosikov A, et al. Decentralized local stochastic extra-gradient for variational inequalities. *Adv. Neural Inf. Process. Syst.*, 2022 (see Assumption 3.1).

---

### Official Review · Reviewer_uTQU · 2025-10-28

**Soundness:** 2
**Presentation:** 2
**Contribution:** 2
**Rating:** 4
**Confidence:** 4

**Summary:**

This paper proposes a fully decentralized payment mechanism that guarantees both truthful behavior and accurate convergence in decentralized stochastic gradient descent algorithms. The theoretical analysis appears sound; however, my main concern is that several key assumptions are unrealistic. I will reevaluate the paper after considering the authors’ rebuttal.

**Strengths:**

This paper proposes a fully decentralized payment mechanism that guarantees both truthful behavior and accurate convergence in decentralized stochastic gradient descent algorithms. The theoretical analysis appears sound; however, my main concern is that several key assumptions are unrealistic.

**Weaknesses:**

My detailed comments are as follows:


1.	My primary concern is that, although agents are allowed to behave strategically, the authors assume they will truthfully follow the proposed payment mechanism. This assumption seems unrealistic — rational agents motivated by self-interest would likely attempt to exploit or manipulate the mechanism to maximize their own benefit. The authors should incorporate additional mechanisms or safeguards to prevent such behavior, rather than relying on an unrealistic assumption of honesty.


2.	In Definition 2 and Lemma 1, the main results rely on the assumption that the neighbors of agent $i$ are truthful. This is impractical, as agent $i$ cannot ensure or verify the honesty of all its neighbors in a decentralized setting. The authors should consider more robust assumptions that better reflect real-world conditions.


3.	In Assumption 1, the authors assume that the loss function is Lipschitz continuous. However, this assumption conflicts with the strongly convex setting, rendering the corresponding results questionable. Moreover, the assumption is overly strong and fails to account for data heterogeneity. The authors should adopt more realistic assumptions that explicitly consider data heterogeneity, as commonly used in the literature.

**Questions:**

N/A

---

> ### Author Response · Authors · 2025-11-20
> **Response to Reviewer uTQU---Part I**
>
> $\rm\color{purple}{@uTQU\ @Weakness\ 1:}$ We thank the reviewer for the insightful comment. In the revised manuscript, we have updated our decentralized payment mechanism to use the shared model parameters directly, rather than transmitting a dedicated gradient magnitude for payment calculation. This change effectively removes the assumption of honest participation in the payment mechanism. Specifically, the  payment from agent $i$ to agent $j$ at  iteration $t$ is calculated as $P_{i,t}^j=C_t(\Delta_{\theta_i,t}-\Delta_{\theta_j,t})$ with $\Delta_{\theta_i,t}\triangleq||\theta_{i,t+1}-2\theta_{i,t}+\theta_{i,t-1}||^2$ and $\Delta_{\theta_{j},t}\triangleq||\theta_{j,t+1}-2\theta_{j,t}+\theta_{j,t-1}||^2$ (see Mechanism 1 on page 6 in the revised manuscript). This modification eliminates the possibility of manipulating the payment mechanism, because all information required to compute the payment is concurrently available to both agents $i$ and $j$ from the updates in Algorithm 1. Specifically, with both agents $i$ and $j$ having access to $\theta_{\iota,t-1}$, $\theta_{\iota,t}$, and $\theta_{\iota,t+1}$ for $\iota\in${$i, j$} under Algorithm 1 (note that $\theta_{\iota,t+1}$ has been shared at the end of iteration $t$), both agents can cross-verify the computed payment value $P_{i,t}^j$. As a result, manipulation of the payment mechanism becomes infeasible and our updated mechanism no longer relies on the assumption of honesty in the payment mechanism. Under the refined payment mechanism, we establish that the truthfulness guarantee and convergence accuracy continue to hold.
>
> To clarify this point, we have added a paragraph of discussion on page 7 in the revised manuscript:
>
> **''In Mechanism 1, with both agents $i$ and $j$ having access to $\theta_{\iota,t-1}$, $\theta_{\iota,t}$, and $\theta_{\iota,t+1}$ for** $\iota\in${$i, j$} **from the update of Algorithm 1 (note that $\theta_{\iota,t+1}$ has been shared at the end of iteration $t$), the two  agents can cross-verify the computed payment value, making the mechanism robust to unilateral manipulation. This represents a significant advance compared with the payment mechanism in Angeli \& Manfredi (2023) for server-assisted collaborative optimization, which requires all agents to truthfully report their local objective-function values for payment calculation—thereby creating a risk that strategic agents may manipulate the algorithmic update and the payment mechanism separately.''**

---

> ### Author Response · Authors · 2025-11-20
> **Response to Reviewer uTQU---Part II**
>
> $\rm\color{purple}{@uTQU\ @Weakness\ 2:}$ Thank you for the comment. We would like to clarify that $\varepsilon$-incentive compatibility (also referred to as $\varepsilon$-Bayesian incentive compatibility) in Definition 2 is a standard and commonly used notion in the existing incentive-compatibility-related literature on collaborative computation (see, e.g., [r1]–[r8]). In fact, the notion suggested by the reviewer (also referred to as $\varepsilon$-dominant incentive compatibility) has been shown to be **infeasible** in a simple two-armed bandit game [r5] (see Lemma 1 in [r5]). Furthermore, the recent truthfulness result in [r6] for collaborative mean estimation has established a hardness result: no nontrivial mechanism can guarantee a dominant-strategy equilibrium (or dominant incentive compatibility) in which truthful reporting is always the best response regardless of the other agents’ strategies (see Section 5 in [r6]).
>
> These infeasibility and hardness results effectively rule out the possibility of designing a dominant-strategy incentive-compatible collaborative-computation mechanism that allows at least one agent to benefit from participating, which is why we believe that the use of Definition 2 is reasonable. It is worth noting that, in contrast to existing results within this incentive compatibility framework—which rely on a trusted party—our work is the first to be implementable in a fully decentralized setting without assistance from any server. Moreover, our approach achieves both $\varepsilon$-Bayesian incentive compatibility and exact convergence, which is a challenging problem even in server-assisted collaborative optimization and learning. Therefore, we believe that our results are significant.
>
> We strongly agree that the notion of incentive compatibility should be clearly named and stated. In the revised manuscript, we have added a sentence under Definition 2 on page 6 to explicitly state that we consider $\varepsilon$-Bayesian-incentive compatibility and it is a standard and commonly used notion in the incentive-compatibility literature.
>
> All existing incentive-compatibility results for collaborative learning and optimization use the same Bayesian incentive compatibility as ours, with some of the typical ones listed below.
>
> [r1] Deng X, et al. A game-theoretic analysis of the empirical revenue maximization algorithm with endogenous sampling. *Adv. Neural Inf. Process. Syst.*, 2020 (see Section ''Approximate Bayesian incentive-compatibility").
>
> [r2] Yin S, et al. Online allocation and learning in the presence of strategic agents. *Adv. Neural Inf. Process. Syst.*, 2022 (see Definition 2).
>
> [r3] Chakarov D, et al. Incentivizing truthful collaboration in heterogeneous federated learning. *OPT: Optimization for Machine Learning*, 2024 (see Definition 4).
>
> [r4] Kakade S M, et al. Optimal dynamic mechanism design and the virtual-pivot mechanism. *Oper. Res.*, 2013 (see Definition 2.2).
>
> [r5] Mansour Y, et al. Bayesian incentive-compatible bandit exploration. *Oper. Res.*, 2020 (see Definition 1).
>
> [r6] Clinton A, et al. Collaborative mean estimation among heterogeneous strategic agents: Individual rationality, fairness, and truthful contribution. *Int. Conf. Mach. Learn.*, 2025 (see Section 3.3).
>
> [r7] Dorner F E, et al. Incentivizing honesty among competitors in collaborative learning and optimization. *Adv. Neural Inf. Process. Syst.*, 2023 (see the statements of Theorem 5.1 and Theorem 5.2).
>
> [r8] Zhao Y, et al. Truthful incentive mechanism for federated learning with crowdsourced data labeling. *IEEE Conf. Comput. Commun.*, 2023 (see Definition 1).
>
> $\rm\color{purple}{@uTQU\ @Weakness\ 3:}$ Thank you for the comment. In the revised manuscript, we have removed the assumption on the $L_{f}$-Lipschitz continuity of objective functions in the strongly convex case. Furthermore, following the reviewer's suggestion, we have also rewritten Assumption 1 in a distributed (local) manner to better adapt to the data heterogeneity across agents. The updated Assumption 1 in the revised manuscript is given as follows:
>
> **''Assumption 1. For any $i\in[N]$, $R_{i}(f_i(\theta))$ is $L_{R,i}$-Lipschitz continuous *w.r.t.* $\theta$ and $g_i(\theta)$ is $H_{i}$-Lipschitz continuous. Moreover, the stochastic gradient estimate $g_{i}(\theta)$ is unbiased and has bounded variance $\sigma_{i}^2$, i.e., $\mathbb{E}[g_i(\theta)] = \nabla f_i(\theta)$ and $\mathbb{E}[||g_i(\theta) - \nabla f_{i}(\theta)||^2] \leq \sigma_{i}^2$. In addition, for a general convex $f_i(\theta)$, we assume that $f_{i}(\theta)$
> is $L_{f,i}$-Lipschitz continuous. However, this assumption is not required for a strongly
> convex $f_i(\theta)$. For the sake of notational simplicity, we denote** $L_R=\max_{i\in[N]}${$L_{R,i}$}, $H=\max_{i\in[N]}${$H_i$}, $L_f=\max_{i\in[N]}${$L_{f,i}$}, **and** $\sigma=\max_{i\in[N]}${$\sigma_i$}**."**
>
> All results and proofs have been updated accordingly in the revised manuscript based on the new Assumption 1.

---

> > ### Comment · Reviewer_uTQU · 2025-11-26
> >
> > Thank you for the authors’ response.
> >
> > I am still not fully convinced why the proposed method can ensure that a strategic agent will truthfully follow the payment mechanism. The authors mention that two agents can cross-verify the computed payment value; however, a strategic agent may report the correct value during cross-verification but still record an incorrect value locally and refuse to follow the payment mechanism in practice. The paper should further clarify how the proposed payment mechanism prevents such behavior. This point remains unclear.
> >
> > Moreover, I am still confused about the assumption that all neighbors of agent $i$ are truthful. The paper aims to study and mitigate strategic behavior, yet it relies on an unrealistic assumption that excludes the possibility of strategic neighbors. If some neighbors of agent $i$ behave strategically, does the algorithm lose all theoretical guarantees?

---

> > > ### Author Response · Authors · 2025-11-27
> > > **Further Response to Reviewer uTQu---part II**
> > >
> > > $\rm\color{purple}{@uTQu\ {}\ Comment\ on\ all\ neighbors\ being\ truthful:}$
> > >
> > > $\rm Response$: Thank you for the comment. Our results do not require neighbor agents to be truthful. We apologize for our earlier unclear rebuttal, which stemmed from a misunderstanding of the comment.
> > >
> > > More specifically, our Definition 2 and theoretical results follow the classic Bayesian incentive compatibility (BIC) framework. In this framework, the analysis of each agent’s optimal action relies on other agents' actual (truthful) information, but this information reflects their **private types**, not their **strategic reports** in the language of mechanism design. Consequently, neighboring agents may still misreport by manipulating their individual types (see, e.g., [r1] for a detailed formal explanation). Thus, although Definition 2 and Lemma 1 use neighboring agents' actual type information in the analysis, we do not require agents to truthfully report (share) this information. Therefore, our definitions and theoretical analysis remain valid even when all agents behave untruthfully.
> > >
> > > [r1] D.C. Parkes. Online mechanisms (book chapter). *Algorithmic Game Theory*, pages 411–439, 2007.
> > >
> > > We acknowledge that the BIC framework is not as strong as the dominant-strategy incentive compatibility (DSIC) framework, in which no actual type information is used in the definition. We emphasize that we intentionally do not use DSIC for two main reasons: 1. the BIC-based mechanism is **the only feasible choice** for decentralized learning with incomplete information; 2. the BIC-based mechanism **is more practical** than DSIC-based mechanisms in real-world decentralized applications. More specifically,
> > >
> > > **1. BIC is the only feasible choice for decentralized learning with incomplete information.**
> > >
> > > Decentralized learning is inherently an incomplete-information paradigm, where each agent observes only its own type and cannot access the types of other agents. In this context, foundational works such as [r2] and [r3] show that **only dictatorial mechanisms**—i.e., mechanisms with a centralized authority capable of accessing all agents' actual information—**can satisfy DSIC**. Consequently, DSIC is fundamentally **incompatible** with decentralized learning settings, where no centralized server exists to aggregate actual information from all agents.
> > >
> > > Recognizing that DSIC is too restrictive for most practical incomplete-information scenarios, the seminal work [r4] introduced the weaker BIC concept, demonstrating that BIC-based mechanisms can still achieve desirable objectives (e.g., budget balance or social-welfare maximization) when agents possess only probabilistic beliefs about others' actual information. These historical results clearly illustrate that BIC is not merely an alternative to DSIC—it is the **only feasible** solution concept for decentralized learning, where no agent has access to the full set of actual information.
> > >
> > > [r2] Gibbard A. Manipulation of voting schemes: a general result. *Econometrica*, 1973.
> > >
> > > [r3] Satterthwaite M A. Strategy-proofness and Arrow's conditions: Existence and correspondence theorems for voting procedures and social welfare functions. *J. Econ. Theory*, 1975.
> > >
> > > [r4] Roberts J. Incentives in planning procedures for the provision of public goods. *Rev. Econ. Stud.*, 1979.
> > >
> > > **2. The consensus in the economic theory community is that BIC offers greater flexibility and tractability for mechanism design than DSIC [r5].**
> > >
> > > As discussed in Point 1, the stringent limitations of DSIC make mechanism design infeasible in most incomplete-information environments. In contrast, the BIC framework provides mechanism designers with far greater flexibility to achieve key objectives such as budget balance or maximizing global social welfare. For example, [r6] shows that budget balance can be achieved under BIC but is **impossible** under DSIC. In auction design, [r7] demonstrates that both optimal social welfare and optimal revenue can be mathematically characterized within the BIC framework, whereas such characterizations are **infeasible** under DSIC.
> > >
> > > These examples underscore why **nearly all** modern mechanism-design literature adopts BIC as the standard framework for coordinated optimization and learning.
> > >
> > > [r5] Williams S R. A characterization of efficient, Bayesian incentive compatible mechanisms. *Econ. Theory*, 1999.
> > >
> > > [r6] d'Aspremont C, Gérard-Varet L A. Incentives and incomplete information. *J. Public Econ.*, 1979.
> > >
> > > [r7] Myerson R B. Optimal auction design. *Math. Oper. Res.*, 1981.

---

> ### Author Response · Authors · 2025-11-27
> **Further Response to Reviewer uTQu---Part I**
>
> $\rm\color{purple}{@uTQu\ {}\ Comment\ on\ manipulating\ recorded\ values:}$
>
> $\rm Response$: We thank the reviewer for the thoughtful response. We clarify that in Mechanism 1 the payments are **executed  immediately** at each iteration, and the recorded values were used solely to define notation that facilitated  subsequent convergence and truthfulness analysis in the paper. These recorded values have **no role** in the actual implementation. In particular, **at every iteration**, each agent $i$  is required to **actually** transfer or receive the cross-verified payment to or from its neighbor $j$, completely independent of any recorded quantities—hence manipulating these recorded values would be entirely meaningless. To avoid confusion, we have **removed** the ''record step" and ''recorded value" in the revised Mechanism 1.

---

### Official Review · Reviewer_fpuV · 2025-11-01

**Soundness:** 3
**Presentation:** 3
**Contribution:** 3
**Rating:** 6
**Confidence:** 4

**Summary:**

This paper addresses a core problem in fully decentralized learning: how to incentivize truthful information sharing among selfish or strategic agents. Specifically, the authors consider a scenario where agents can manipulate their gradients (e.g., by scaling them) to bias the model. They propose a novel payment mechanism where agents with larger gradient norms must pay agents with smaller norms. The authors provide theoretical proofs showing that by carefully designing the decay rate of the payment coefficient ($C_t$), the optimal strategy for agents converges to "full truthfulness" over time. They also prove that DSGD converges even in the presence of this (diminishing) strategic manipulation.

**Strengths:**

* To the reviewer's knowledge, this is the first work to propose a fully decentralized payment mechanism to address truthfulness in this setting, which is a significant contribution.
* The theoretical results, particularly the the convergence (despite manipulation) and finite cumulative gain, provide strong theoretical support for the proposed method.
* The experimental results align with the theory, demonstrating that the presence of the payment mechanism effectively mitigates the drop in test accuracy caused by strategic manipulation.
* The paper is well-written, clear, and provides a thorough background on the problem.

**Weaknesses:**

1.  The proposed payment mechanism may inadvertently incentivize free-riding.
    The reviewer understands that this paper focuses on the "Truthfulness" problem—preventing *active* participants from manipulating gradients for personal gain. However, it does not address (and may worsen) a related and important incentive problem: free-riding.
    Consider a free-rider agent $i$ that submits a zero-gradient (or a gradient with a very small norm), i.e., $m_{i,t} = 0$. According to Mechanism 1, if this agent's honest neighbor $j$ computes a valid, non-zero gradient, then $||m_{j,t}|| > ||m_{i,t}||$. This forces the honest agent $j$ to *pay* the free-riding agent $i$. Consequently, the mechanism does not penalize this "passive" non-participation and, in fact, provides a direct financial incentive for agents to submit low-norm gradients.

2.  The payment mechanism appears vulnerable due to its simplicity.
    The mechanism's reliance *only* on the reported gradient norm makes it susceptible to strategic decoupling. The paper's core “implicit” assumption is that agents will honestly report the norm $||m_{i,t}||$ for the payment calculation. A strategic agent can separate its update from its payment. For example, an agent $i$ could use a highly manipulated gradient $m_{i,t}$ in its local update (Eq. 2) to influence its $\theta_{i,t+1}$ (which is then shared). Simultaneously, to avoid payment, it could falsely report a "reasonable" norm $||m'_{i,t}||$ (e.g., a norm similar to its neighbors' or its own true gradient) to the payment mechanism. In this scenario, the agent gains the benefit of manipulation without incurring the cost.

**Questions:**

1.  Clarity on Convergence Rate Comparison:
    The main convergence results are presented in Big-O notation, which obscures some constants and makes direct comparison difficult. Theorem 1 shows a convergence rate of $\mathcal{O}(T^{-(1-v)})$ for the non-convex case, where $v \in (1/2, 2/3)$. This rate (between $\mathcal{O}(T^{-1/3})$ and $\mathcal{O}(T^{-1/2})$) appears slower than the $\mathcal{O}(1/\sqrt{T})$ rate achievable by standard DSGD (e.g., Koloskova et al., 2020). For a clearer comparison, could the authors provide a convergence rate in a form that is more directly comparable to standard DSGD rates, such as Theorem 2 in Koloskova et al. [1]?

2.  Impact of Data Heterogeneity on Honest Agents:
    The paper's experiments rightly suggest that high data heterogeneity *increases* the incentive for gradient manipulation. However, this raises a question about the mechanism's fairness. In a highly non-IID setting, an *honest* agent's true gradient norm $||g_{i,t}||$ could naturally be an order of magnitude larger or smaller than its neighbors' norms, simply due to different data distributions.
    Does Mechanism 1 risk misinterpreting this "natural difference" as "strategic manipulation"? Could this lead to the mechanism unfairly penalizing honest agents whose data distributions are simply unique or outliers compared to their neighbors?

### References

[1] Koloskova et al. A Unified Theory of Decentralized SGD with Changing Topology and Local Updates. ICML 2020.

---

> ### Author Response · Authors · 2025-11-20
> **Response to Reviewer fpuV---Part I**
>
> $\rm\color{red}{@fpuV\ @Weakness\ 1:}$ We thank the reviewer for the time spent reviewing our manuscript and the insightful comments. In the revised version, we have refined our payment mechanism to use differences in model-parameter increments instead of differences in gradient-magnitudes, enabling us to address free-riding and payment manipulation. At the same time, we establish that the truthfulness guarantee and convergence accuracy continue to hold.
>
> Specifically, in the revised version, we have updated the payment from agent $i$ to agent $j$ at iteration $t$ as $P_{i,t}^j=C_t(\Delta_{\theta_i,t}-\Delta_{\theta_j,t})$ with $\Delta_{\theta_i,t}\triangleq||\theta_{i,t+1}-2\theta_{i,t}+\theta_{i,t-1}||^2$ and $\Delta_{\theta_j,t}\triangleq||\theta_{j,t+1}-2\theta_{j,t}+\theta_{j,t-1}||^2$. Because $\theta_{\iota,t-1}$, $\theta_{\iota,t}$, $\theta_{\iota,t+1}$ for $\iota\in${$i, j$} are available to both agents $i$ and $j$ under Algorithm 1 (note that $\theta_{\iota,t+1}$ has been shared at the end of iteration $t$), the two agents can cross-verify the computed payment value, making the mechanism robust to unilateral manipulation. Moreover, since agent $i$'s model-parameter increment $\Delta_{\theta_i,t}$ is determined by both consensus error (see dynamics of $\Delta_{\theta_i,t}$ in Eq. (54)) and the (sign-indefinite) local gradients $m_{i,t}$ and $m_{i,t-1}$, even if agent $i$ shares a zero gradient, the resulting value of $\Delta_{\theta_i,t}$ is not necessarily smaller than that of its neighbors. Therefore, agent $i$ cannot reliably increase its payment gains by sharing zero gradients. Meanwhile, free-riding behavior (or using $a<1$) invariably degrades agent $i$'s own reward $R_i(f_i(\theta_{i,T}))$, as it weakens the influence of agent $i$'s data in collaborative learning and leads to a worse final model for the agent itself. Hence, free-riding does not constitute a utility-improving choice for a rational agent.
>
> We added a paragraph on page 7 to clarify this (please also see the refined Mechanism 1, updated Lemma 1/Theorem 1 on pages 6-8):
>
> **''Our payment mechanism can effectively discourage agents from free-riding. Specifically, the model-parameter increment $||\theta_{i,t+1}-2\theta_{i,t}+\theta_{i,t-1}||$ of agent $i$ depends on both the consensus errors (see the dynamics of $\Delta_{\theta_i,t}$ in equation (54)) and the (sign-indefinite) local gradients. Consequently, even if agent $i$ uses a zero (or low) gradient, there is no guarantee that  $||\theta_{i,t+1}-2\theta_{i,t}+\theta_{i,t-1}||$ will be smaller than that of its neighbor $j$, meaning that leveraging zero gradients does not reliably increase agent $i$'s payment gains. In contrast, free-riding behavior (or using low gradients) invariably degrades agent $i$’s own reward $R_{i}(f_i(\theta_{i,T}))$, as it weakens the influence of agent $i$'s data in collaborative learning and ultimately leads to a worse final model for the agent itself. Hence, free-riding is not a utility-improving choice for a rational agent.''**
>
> $\rm\color{red}{@fpuV\ @Weakness\ 2:}$ Thanks for the insightful comments. As stated above, we have refined our payment mechanism to use differences in model-parameter increments instead of differences in gradient magnitude, enabling us to address free-riding and payment manipulation.
>
> Specifically, in the revised version, we have updated the payment from  agent  $i$ to agent $j$ at iteration $t$ as $P_{i,t}^j=C_t(\Delta_{\theta_i,t}-\Delta_{\theta_j,t})$ with $\Delta_{\theta_i,t}\triangleq||\theta_{i,t+1}-2\theta_{i,t}+\theta_{i,t-1}||^2$ and $\Delta_{\theta_{j},t}\triangleq||\theta_{j,t+1}-2\theta_{j,t}+\theta_{j,t-1}||^2$. Because $\theta_{\iota,t-1}$, $\theta_{\iota,t}$, $\theta_{\iota,t+1}$ for $\iota\in${$i, j$} are available to both agents $i$ and $j$ under Algorithm 1 (note that $\theta_{\iota,t+1}$ has been shared at the end of iteration $t$), the two agents can cross-verify the computed payment value, eliminating the possibility of payment manipulation. As a result, our updated mechanism no longer relies on the assumption of honesty in the payment mechanism.
>
> To clarify this point, we have added discussions on page 7 in the revised manuscript:
>
> **''In Mechanism 1, with both agents $i$ and $j$ having access to $\theta_{\iota,t-1}$, $\theta_{\iota,t}$, and $\theta_{\iota,t+1}$ for** $\iota\in${$i, j$} **from the update of Algorithm 1 (note that $\theta_{\iota,t+1}$ has been shared at the end of iteration $t$), the two  agents can cross-verify the computed payment value, making the mechanism robust to unilateral manipulation. This represents a significant advance compared with the payment mechanism in Angeli \& Manfredi (2023) for server-assisted collaborative optimization, which requires all agents to truthfully report their local objective-function values for payment calculation—thereby creating a risk that strategic agents may manipulate the algorithmic update and the payment mechanism separately."**

---

> ### Author Response · Authors · 2025-11-20
> **Response to Reviewer fpuV---Part II**
>
> $\rm\color{red}{@fpuV\ @Question\ 1:}$ Thank you for the comment. In the revised manuscript, we have provided a more explicit expression of convergence rates in Theorem 1. The updated Theorem 1 is given as follows:
>
> **''Theorem 1 (Convergence rate). We denote $\theta^*$ as a solution to the problem in equation 1. Under our Mechanism 1 and the conditions in Lemma 1, for any $i\in[N]$, $\delta>0$, and $T\geq0$, the following results hold for Algorithm 1 in the presence of strategic behaviors:**
>
> **(i) if $F(\theta)$ is $\mu$-strongly convex (not necessarily Lipschitz continuous), then we have**
>
> $\mathbb{E}[||\theta_{i,T}-\theta^{*}||^2]\leq \mathcal{O}\left(\frac{H^{2}(\sigma^{2}+\delta^2)}{\mu(1-\rho)^2(T+1)^{v}}\right)$;
>
> **(ii) if $F(\theta)$ is general convex, then we have**
>
> $\frac{1}{T+1}\sum_{t=0}^{T}\mathbb{E}[F(\theta_{i,t})-F(\theta^{*})]\leq \mathcal{O}\left(\frac{H^{2}(\sigma^{2}+L_{f}^2+\delta^2)}{(1-\rho)^2(T+1)^{1-v}}\right)$,
>
> **where $H$, $\sigma$, and $L_{f}$ are from Assumption 1 and** $\rho=\max${$|\pi_2|,|\pi_N|$}$<1$ **is from Assumption 2."**
>
> It can be seen that a larger strong convexity coefficient $\mu$ and a smaller $\rho$ (corresponding to a better-connected communication topology) lead to a higher convergence rate. In contrast, larger values of the smoothness coefficient $H$, Lipschitz constant $L_{f}$, truthfulness parameter $\delta$, and gradient noise variance $\sigma$ lead to a lower convergence rate. In addition, according to the condition $\frac{1}{2}<v<\frac{2}{3}$ in Lemma 1, we can choose $v=0.5+\varsigma$ with an arbitrarily small $\varsigma>0$ to yield a convergence rate arbitrarily close to $\mathcal{O}(1/\sqrt{T})$ in both the general convex and strongly convex settings.
>
> Furthermore, in a form similar to [1], in the revised manuscript, we have also provided the computational complexity   in Corollary 2 in Appendix E.3 on page 44. The results prove that the computational complexity of our approach is arbitrarily close to $\mathcal{O}(n\varrho^{-2})$ in finding a $\varrho$-solution in both the strongly convex and general convex settings, where $n$ denotes the dimension of the model parameter $\theta_{i,t}$.
>
> For general convex objective functions, our convergence result is similar to that of [1], which established a convergence rate of $\mathcal{O}(1/\sqrt{T})$ and a computational complexity of $\mathcal{O}(n\varrho^{-2})$ in the absence of strategic manipulation. For strongly convex objective functions, our convergence rate is lower than that of [1], where a convergence rate of $\mathcal{O}(1/T)$ and a computational complexity of $\mathcal{O}(n\varrho^{-1})$ were obtained. We consider the reduction in convergence speed compared with [1] to be a necessary cost for achieving $\varepsilon$-incentive compatibility in fully decentralized optimization/learning. In fact, compared with existing truthfulness results for collaborative learning in, e.g., [r1]–[r4], which sacrifice convergence accuracy to ensure truthfulness, we consider a degradation in convergence speed to be a comparatively mild cost.
>
> To clarify these points, we have added a paragraph under Corollary 2 in Appendix E.3 on page 45 in the revised manuscript. For the reviewer's convenience, we reproduce the added text below:
>
> **''Corollary 2 proves that in general convex settings, Algorithm 1 with our decentralized payment mechanism achieves a convergence rate of $\mathcal{O}(1/\sqrt{T})$ and a computational complexity of $\mathcal{O}(n\varrho^{-2})$ similar to those attained by the standard decentralized SGD algorithm in Koloskova et al. (2020c), even under the constraint of truthfulness. Although the results of convergence rate and computational complexity in Corollary 2 are inferior to those achieved by Koloskova et al. (2020c) in strongly convex settings, we would like to emphasize that this degradation is a comparatively mild cost for ensuring both accurate convergence and $\varepsilon$-incentive compatibility (truthfulness) in fully decentralized optimization and learning. This is different from existing truthfulness results in, e.g., Han et al. (2015); Hale \& Egerstedt (2015); Dorner et al. (2023); Chakarov et al. (2024a;b); Chen et al. (2025), all of which sacrifice convergence accuracy to guarantee truthfulness."**
>
> [r1] Zhang L, Zhu T, Xiong P, et al. A robust game-theoretical federated learning framework with joint differential privacy. *IEEE Trans. Knowl. Data Eng.*, 2022.
>
> [r2] Dorner F E, Konstantinov N, Pashaliev G, et al. Incentivizing honesty among competitors in collaborative learning and optimization. *Adv. Neural Inf. Process. Syst.*, 2023.
>
> [r3] Chakarov D, Tsoy N, Minchev K, et al. Incentivizing truthful collaboration in heterogeneous federated learning. *OPT: Optimization for Machine Learning*, 2024.
>
> [r4] Chen Z, Egerstedt M, Wang Y. Ensuring truthfulness in distributed aggregative optimization. *IEEE Trans. Autom. Control*, 2025.

---

> > ### Author Response · Authors · 2025-11-20
> > **Response to Reviewer fpuV---Part III**
> >
> > $\rm\color{red}{@fpuV\ @Question\ 2:}$ Thank you for the comment. In the revised manuscript, we have updated our decentralized payment mechanism to use differences in model-parameter increments, rather than gradient-magnitude differences for payment calculation. This change effectively mitigates the risk of misinterpreting the natural difference of gradient heterogeneity among agents as strategic manipulation.
> >
> > Specifically, we have  updated the payment from  agent  $i$ to agent $j$ at iteration $t$ as $P_{i,t}^{j}=C_{t}(\Delta_{\theta_{i},t}-\Delta_{\theta_{j},t})$ with $\Delta_{\theta_{i},t}\triangleq||\theta_{i,t+1}-2\theta_{i,t}+\theta_{i,t-1}||^2$ and $\Delta_{\theta_{j},t}\triangleq||\theta_{j,t+1}-2\theta_{j,t}+\theta_{j,t-1}||^2$. Each agent $i$'s model-parameter increment $\Delta_{\theta_{i},t}$ is determined by the consensus operation, the stepsize choice, and  the (manipulated) local gradient updates. Among these factors, the consensus operation and diminishing stepsizes effectively smoothen the natural differences arising from heterogeneous gradients among agents (leaving only abnormally large model-parameter increments to signal strategic manipulation), and thereby avoiding the fairness issue mentioned by the reviewer.

---

### Meta-Review · Area_Chair_szhk · 2025-12-18

**Summary:**

The reviewers agree that this paper is the first one to address truthfulness in the decentralized setting. They appreciate the theoretical analysis and motivation of the work. However, all reviews indicate that the attack model and the payment mechanism are impractical or too simple, since the nodes might not follow the mechanism and exploit it. After receiving the initial reviews, the authors did a huge amount of work to improve the paper. Nevertheless, those reviewers (Reviewer uTQU and m7Yo) who participated in the discussion were still hesitant to change their opinion. Thus, I recommend rejection and suggest the authors use the discussion to improve their payment mechanism design.

**Reviewer Concerns:**

According to "Official Comment by Reviewer uTQU," Reviewer uTQU is not convinced by the changes: "I am still not fully convinced why the proposed method can ensure that a strategic agent will truthfully follow the payment mechanism. [...]" and "[...] Moreover, I am still confused about the assumption that all neighbors of agent are truthful." Moreover, Reviewer m7Yo in the last comments disagrees that there is no clear incentive for manipulating shared parameters and indicates that there are still unsolved concerns. Thus, at least two reviewers were not satisfied, making me believe that the other may concur with them.

**Reviewer Scores:**

I believe that Reviewers uTQU and m7Yo wouldn't change their score since they are still unconvinced, which is clear from their follow-up comments. I'm not sure whether Reviewer PjnL would increase the score since Reviewer PjnL outlined almost the same weakness as Reviewers uTQU and m7Yo.

---

### Decision · Program_Chairs · 2026-01-26

Reject